# Learning Successor Features the Simple Way

**Raymond Chua**[*][§]

**Arna Ghosh**[§]

**Christos Kaplanis**[¶]

**Blake A. Richards**[†][‡][§]

**Doina Precup**[‡][§][¶]

## Abstract

In Deep Reinforcement Learning (RL), it is a challenge to learn representations that do not exhibit catastrophic forgetting or interference in non-stationary environments. Successor Features (SFs) offer a potential solution to this challenge. However, canonical techniques for learning SFs from pixel-level observations often lead to representation collapse, wherein representations degenerate and fail to capture meaningful variations in the data. More recent methods for learning SFs can avoid representation collapse, but they often involve complex losses and multiple learning phases, reducing their efficiency. We introduce a novel, simple method for learning SFs directly from pixels. Our approach uses a combination of a Temporal-difference (TD) loss and a reward prediction loss, which together capture the basic mathematical definition of SFs. We show that our approach matches or outperforms existing SF learning techniques in both 2D (Minigrid), 3D (Miniworld) mazes and Mujoco, for both single and continual learning scenarios. As well, our technique is efficient, and can reach higher levels of performance in less time than other approaches. Our work provides a new, streamlined technique for learning SFs directly from pixel observations, with no pretraining required[1].

## 1 Introduction

Deep reinforcement learning (RL) [Sutton and Barto, 2018] is important to modern artificial intelligence (AI), but standard approaches to deep RL can struggle when deployed for continual learning [Parisi et al., 2019, Hadsell et al., 2020, Khetarpal et al., 2022]. When either the reward function or the transition dynamics of the environment changes, standard deep RL techniques, such as deep Q-learning, will either struggle to adapt to the changes or they will exhibit catastrophic forgetting [Kirkpatrick et al., 2017, Kaplanis et al., 2018]. Given that the real-world is often non-stationary, better techniques for deep RL in continual learning are a major goal in AI research [Rusu et al., 2016, Rolnick et al., 2019, Powers et al., 2022, Abbas et al., 2023, Anand and Precup, 2024].

One potential solution that researchers have explored is the use of Successor Features (SFs). Successor Features, the function approximation variant of Successor Representations (SRs) [Barreto et al., 2017], decompose the value function into a separate reward function and transition dynamics representation [Dayan, 1993]. In doing so, they make it easier to adapt to changes in the environment, because the network can relearn either the reward function or the transition dynamics separately [Borsa et al., 2018, Barreto et al., 2018, 2020, Hansen et al., 2019, Lehnert and Littman, 2019, Liu and Abbeel, 2021]. Furthermore, there are theoretical guarantees that SFs can improve generalization in multi-task

---

[*]Correspondence to: ray.r.chua@gmail.com

[†]Dept of Neurology & Neurosurgery, and Montreal Neurological Institute of McGill University.

[‡]Co-senior Authorship. CIFAR Learning in Machines and Brains.

[§]School of Computer Science, McGill University & Mila

[¶]Google Deepmind

[1]Code: `https://github.com/raymondchua/simple_successor_features`

38th Conference on Neural Information Processing Systems (NeurIPS 2024).

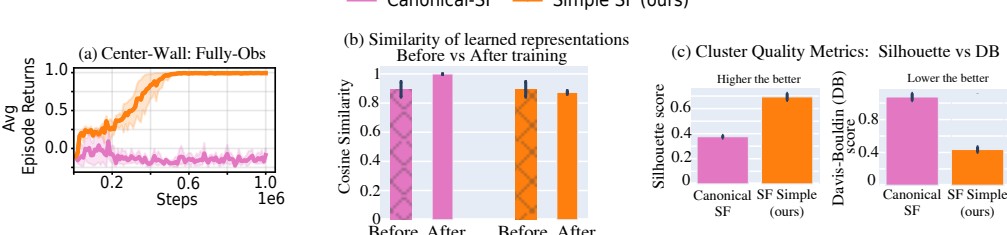

Figure 1: (**a**) Results from a single task within a 2D two-room environment, illustrating the suboptimal performance of the canonical Successor Features (SF) learning rule (Eq. 4) due to representation collapse. (**b**) In the canonical SF approach, the average cosine similarity between pairs of SFs converges towards a value of 1, demonstrating representation collapse occurs. (**c**) The canonical SF learning rule does not develop distinct clusters in its representations, as evidenced by lower silhouette scores and higher Davies-Bouldin scores, which again indicates representation collapse. A mathematical proof can be found in section 3.4.

settings [Barreto et al., 2017]. SFs are therefore a promising candidate for deep RL in non-stationary settings.

However, learning SFs is non-trivial. The most straightforward solution, which is to use a temporal-difference (TD) error on subsequent observations [Barreto et al., 2018], can lead to representational collapse, where the artificial neural network maps all inputs to the same point in a high-dimensional representation space. This phenomenon is commonly observed in various deep learning pipelines that end up learning similar or identical latent representations for very different inputs [Aghajanyan et al., 2020]. In RL, representation collapse can lead to different states or state-action pairs being mapped to similar representations, leading to suboptimal policy decisions or inaccurate estimation of values.

To solve this problem, a variety of solutions have been proposed. One solution is to use an additional reconstruction loss [Kulkarni et al., 2016, Zhang et al., 2017, Machado et al., 2020] to force the network to maintain information about the inputs in its representations. Another solution is to use extensive pretraining coupled with additional loss terms to encourage high-entropy representations [Hansen et al., 2019, Liu and Abbeel, 2021]. More recently, an alternative solution using loss terms to promote orthogonal representations has been put forward [Mahadevan and Maggioni, 2007, Machado et al., 2017b]. Finally, an unconventional approach integrates Q-learning and reward prediction losses with the SF-TD loss, enhancing the learning process by providing additional supervisory signals that improve the robustness and effectiveness of the successor features [Janz et al., 2019]. This method allows the network to simultaneously learn the basis features, successor features, and task encoding vector, with the hope that the learned variables will satisfy their respective constraints.

Though these solutions prevent representational collapse, they can impair learning, introduce additional training phases, or add expensive covariance calculations to the loss function [Touati et al., 2022]. Ideally, there would be a way to learn deep SFs directly during task engagement with a simple, easy to calculate loss function.

Here, we introduce a simple technique for learning SFs directly during task engagement. We designed a neural network architecture specifically to achieve this training objective. Our approach leverages the mathematical definition of SFs and constructs a loss function with two terms: one that learns the value function with a TD-error, and another that enforces representations that make the rewards linearly predictable. By mathematical definition, this loss is minimized when the system has learned a set of SFs. We show that training with this loss during task engagement, facilitated by our neural network architecture, leads to the learning of deep SFs as well as, or better than, other approaches. It does so with no pretraining required and very minimal computational overhead. As well, we show that our technique improves continual reinforcement learning in dynamic environments, in both 2D grid worlds and 3D mazes. Altogether, our simple deep SF learning approach is an effective way to achieve the benefits of deep SFs without any of the drawbacks.

## 2 Related work

Our work builds on an extensive literature on decomposing the value function dating back to the 1990s [Dayan, 1993]. More recent work on learning deep SFs falls broadly into three categories

of solutions. The first are solutions that use a reconstruction term in the loss function in order to avoid representational collapse [Kulkarni et al., 2016, Zhang et al., 2017, Machado et al., 2020]. This general approach is effective at avoiding collapse, but it can lead to impaired performance on the actual RL task, as we show below. The next set of solutions rely on hand-crafted features [Lehnert et al., 2017, Barreto et al., 2018, Borsa et al., 2018, Madarasz and Behrens, 2019, Machado et al., 2021, Emukpere et al., 2021, Nemecek and Parr, 2021, Brantley et al., 2021, McLeod et al., 2022, Alegre et al., 2022, Reinke and Alameda-Pineda, 2021] or hand-crafted task knowledge [Hansen et al., 2019, Filos et al., 2021, Liu and Abbeel, 2021, Carvalho et al., 2023a]. In these cases, the networks can learn and generalize well, but hand-crafted solutions cannot scale-up to real-world applications. Another category of solutions uses pretraining of the features in the deep neural network before any engagement with the actual RL task [Fujimoto et al., 2021, Abdolshah et al., 2021, Touati et al., 2022, Carvalho et al., 2023b]. Such solutions are not as applicable for continual RL because they introduce the need to engage in new pretraining when the environment changes, which assumes some form of oracle knowledge of the environment. Finally, there are solutions that rely on additional loss terms to encourage orthogonal representations, since SRs are built off of purely orthogonal tabular inputs [Touati et al., 2022, Farebrother et al., 2023]. These techniques can improve SF learning, but they require computationally expensive calculations of orthogonality in the basis features.

Among these prior approaches, the work most closely related to ours is the application of multiple losses to jointly learn the SFs, a task-encoding vector, and Q-values [Ma et al., 2020]. However, there are several key differences: (1) Our approach does not require the agent to be provided with a goal—it is learned through interaction with the environment; (2) We provide direct evidence that our method works with pixel inputs; (3) We demonstrate that our approach eliminates the need for an SF loss; and (4) By removing the SF loss, we reduce the number of hyperparameters required, thereby simplifying the model.

In our results below, we compare our method to these classes of solutions described, namely reconstruction solutions [Machado et al., 2020], pretraining solutions [Liu and Abbeel, 2021], and orthogonality solutions [Touati et al., 2022].

## 3 Preliminaries

### 3.1 Reinforcement Learning

The RL setting is formalized as a Markov Decision Process defined by a tuple $(S, A, p, r, \gamma)$, where $\mathcal{S}$ is the set of states, $\mathcal{A}$ is the set of actions, $r : S \to \mathbb{R}$ is the reward function, $p : \mathcal{S} \times \mathcal{A} \to [0, 1]$ is the transition probability function and $\gamma \in [0, 1)$ is the discount factor which is being to used to balance the importance of immediate and future rewards [Sutton and Barto, 2018].

At each time step $t$, the agent observes state $S_t \in \mathcal{S}$ and takes an action $A_t \in \mathcal{A}$ sampled from a policy $\pi : \mathcal{S} \times \mathcal{A} \to [0, 1]$, resulting in to a transition of next state $S_{t+1}$ with probability $p(S_{t+1} \mid S_t, A_t)$ and the reward $R_{t+1}$.

### 3.2 Successor Features

SFs are defined via a decomposition of the state-action value function (i.e. the expected return), $Q$, into the reward function and a representation of expected features occupancy for each state $S_t$ and action $A_t$ of time step $t$:

$$Q(S_t, A_t, \boldsymbol{w}) = \psi(S_t, A_t, \boldsymbol{w})^\top \boldsymbol{w} \tag{1}$$

where $\psi \in \mathbb{R}^n$ are the SFs that capture expected feature occupancy and $\boldsymbol{w} \in \mathbb{R}^n$ is a vector of the task encoding, which can be considered a representation of the reward function [Borsa et al., 2018].

Canonically, the SFs for a state-action pair $(s, a)$ under a policy $\pi$ are defined as:

$$\psi^\pi(s, a) \equiv \mathrm{E}^\pi \left[ \sum_{i=t}^{\infty} \gamma^{i-t} \phi_{i+1} \mid S_t = s, A_t = a \right] \tag{2}$$

where $\phi \in \mathbb{R}^n$ is a set of basis features, and $\pi$ is the policy [Barreto et al., 2017].

However, as shown by Borsa et al. [2018], we can treat the task encoding vector $\boldsymbol{w}$ as a way to encode policy $\pi$. This results in *Universal SFs*, $\psi(s, a, \boldsymbol{w})$, on which we base our work.

The task encoding vector $\boldsymbol{w}$ can also be related directly to the rewards themselves via the underlying basis features ($\phi$):

$$R_{t+1} = \phi(S_{t+1})^\top \boldsymbol{w} \tag{3}$$

## 3.3 Canonical Approach to Learning Successor Features and its Limitations

The canonical approach for learning the basis features $\phi$ and successor features $\psi$ for each state $S_t$ and action $A_t$ of time step $t$, with respect to policy $\pi$, are achieved by optimizing the following SF Temporal-Difference loss:

$$L_{\phi,\psi} = \frac{1}{2} \left\| \phi(S_{t+1}) + \gamma\psi(S_{t+1}, a, \boldsymbol{w})) - \psi(S_t, A_t, \boldsymbol{w}) \right\|^2 \tag{4}$$

where action $a \sim \pi(S_{t+1})$. The basis features $\phi$ are typically defined as the normalized output of an encoder, which the SFs $\psi$ learn from concurrently (see Figure 2 for an example).

However, when the basis features, $\phi$, must be learned from high-dimensional, complex observations such as pixels, optimizing Eq. 4 may result in the basis features, $\phi$, converging to a constant vector. This outcome occurs because it can minimize the loss, as noted by Machado et al. [2020], which we will also prove mathematically below.

## 3.4 Proof by Contradiction: Representation Collapse in Successor Features

Consider the basis features function $\phi(\cdot)$ and the Successor Features $\psi(\cdot)$, omitting the inputs for clarity. The canonical SF-TD loss (Eq. 4) is defined as:

$$L_{\phi,\psi} = \frac{1}{2} \left\| \phi(\cdot) + \gamma\psi(\cdot) - \psi(\cdot) \right\|^2 \tag{5}$$

Using *proof by contradiction*, we aim to show that when both $\phi(\cdot)$ and $\psi(\cdot)$ are constants across all states $S$, specifically when $\phi(\cdot) = c_1$ and $\psi(\cdot) = c_2$ with $c_1 = (1 - \gamma)c_2$, the system satisfies the zero-loss conditions, leading to representation collapse.

We start with the assumption that if $\phi(\cdot) = c_1, \psi(\cdot) = c_2$, then $L_{\phi,\psi} \neq 0 \; \forall c_1, c_2 \in \mathbb{R}$.

Substituting $\phi(\cdot) = c_1$ and $\psi(\cdot) = c_2$ into the loss function:

$$L_{\phi,\psi} = \frac{1}{2} \left\| c_1 + \gamma c_2 - c_2 \right\|^2 \tag{6}$$

It is trivial to observe that if $c_1 = (1 - \gamma)c_2$, the expression for $L_{\phi,\psi}$ is as follows:

$$\begin{aligned} L_{\phi,\psi} &= \frac{1}{2} \left\| (1 - \gamma)c_2 + \gamma c_2 - c_2 \right\|^2 \\ &= \frac{1}{2} \left\| 0 \right\|^2 \\ &= 0 \end{aligned} \tag{7}$$

This contradicts our assumption that $L_{\phi,\psi} \neq 0$ for a particular relationship between $c_1$ and $c_2$. $\quad\square$

Thus, we have shown that there exist constants $c_1, c_2$ such that when $\phi(\cdot) = c_1$ and $\psi(\cdot) = c_2$ with $c_1 = (1 - \gamma)c_2$, the system **does** satisfy the zero-loss conditions, resulting in degenerate solutions for $L_{\phi,\psi}$, i.e. causing representation collapse. In this collapsed state, $\phi(\cdot)$ loses its ability to distinguish between different states effectively, causing the model to lose critical discriminative information and thus impairing its generalization capabilities.

Additionally, we also show empirically in Figure 1(a-c) of the presence of representation collapse when learning using Eq. 4. In this work, our method aims to mitigate these issues with a novel, simple approach for learning SFs directly from pixels.

# 4 Proposed Method

The key insight from the proof above (section 3.4) is that preventing representation collapse requires avoiding the scenario where the basis features $\phi$ become a constant vector for all states $S$, which would minimize the loss without contributing to meaningful learning. Below, we will describe the steps taken in our approach to mitigate these issues causing representation collapse.

We note that when the representations $\psi$ form a set of SFs, Eq. (1) is satisfied for some $w$ that also satisfies Eq. (3). Therefore, the approach we take to learn SFs is simply to ensure that over the course of the learning $\psi$ and $w$ come to satisfy both of these equations, which can be achieved by using the following loss functions:

$$L_w = \frac{1}{2}\left\| R_{t+1} - \overline{\phi}(S_{t+1})^\top w \right\|^2 \quad (8)$$

$$L_\psi = \frac{1}{2}\left\| \hat{y} - \psi(S_t, A_t, w)^\top w \right\|^2 \quad (9)$$

where $\overline{\phi}(S_{t+1})$ is treated as a constant in Eq. 8 using a stop-gradient operator, and $\hat{y}$ is the bootstrapped target:

$$\hat{y} = R_{t+1} + \gamma \max_{a'} \psi(S_{t+1}, a', w)^\top w \quad (10)$$

Here, $w$ is only altered by $L_w$, whereas SF $\psi$ and the basis features $\phi$ are learned via $L_\psi$.

Specifically, our proposed approach can *overcome representation collapse by treating the basis features $\phi$ as the L2 normalized output from the encoder of the SF $\psi$ network* (Figure 2), because unlike in Eq. 4, Eq. 8 and Eq. 9 are not minimized by setting $\phi$ to a constant value, given that $\hat{y}$ and $R_{t+1}$ are *not constants for all states $S$*. Hence, there is nothing encouraging the network to converge to a constant vector, naturally avoiding representational collapse.

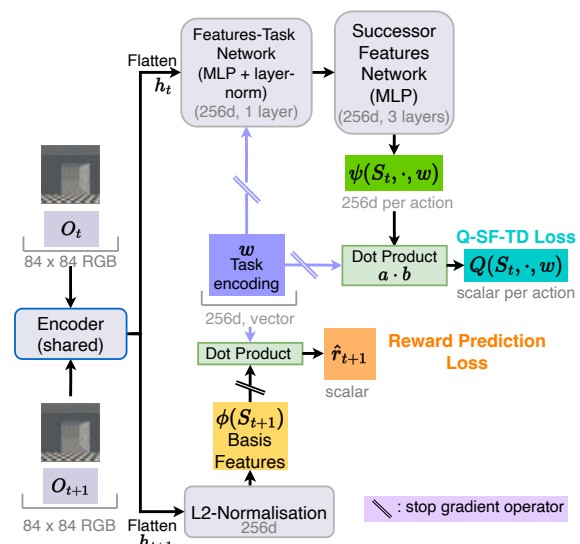

Figure 2: Our proposed model for learning SFs. Starting from the top, the representations of state $S_t$ are learned using the shared encoder, resulting in $h_t$. The basis features $\phi(S_{t+1})$ are the normalized output of the encoder using state $S_{t+1}$. The task-encoding vector $w$ is learned through the reward prediction loss (Eq. 8). Concatenated with $w$, the basis features and successor features are learned through computing the Q-values with $w$ and minimizing the *Q-SF-TD* loss function (Eq. 9). A schematic for continuous actions and previous approaches can be found in Appendix G and H respectively.

When the basis features $\phi$ are needed to learn the task encoding vector $w$ through the reward prediction loss (Eq. 8), we apply a stop-gradient operator to treat the basis features $\phi$ as fixed. As we will demonstrate in section 7 "Analysis of Efficiency and Efficacy", this inclusion of a stop-gradient operator is crucial. Without it, learning both the basis features $\phi$ and the task encoding vector $w$ concurrently can lead to learning instability.

Next, we will clarify how our approach relates to learning SFs, as they are defined mathematically. Given the straightforward nature of our approach, we refer to the SFs learned as *"Simple SFs."*

## 4.1 Bridging Simple SFs and Universal Successor Features

In Proposition 1 (Appendix C), we show that our approach ultimately produces true SFs, equivalent to the SFs learned using Eq. 4. Proposition 1 does this by proving that minimizing our losses (Eq. 8 & Eq 9) also minimizes the canonical SF loss used in Universal Successor Features (Eq. 4). Furthermore, Proposition 1 supports the proof above (Section 3.4) that our approach minimizes these losses in a manner such that setting the basis features $\phi$ to a constant is not a solution. Once again, if $\psi = c_2$ and $\phi = c_1 = (1 - \gamma)c_2$ then Eq. 8 & Eq. 9 are not minimized, due to the fact that $\hat{y}$ and $R_{t+1}$ in Eq. 10 are not constants for all states $S$.

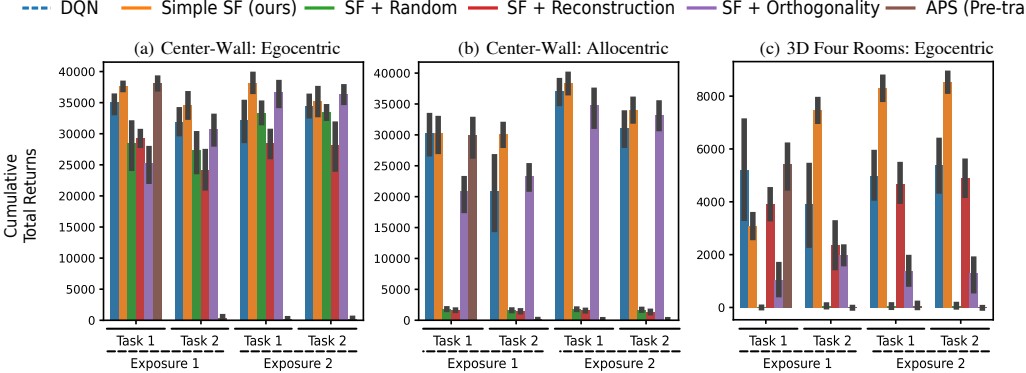

Figure 3: Continual Reinforcement Learning Evaluation with pixel observations in 2D Minigrid and 3D Four Rooms environment. **Replay buffer resets at each task transitions** to simulate drastic distribution shifts: Agents face two sequential tasks (Task 1 & Task 2), each repeated twice (Exposure 1 & Exposure 2). **(a-c):** The total cumulative returns accumulated during training. Overall, our agent, Simple SF (orange), shows notable superiority and exhibited better transfer in later tasks over both DQN (blue) and agents with added constraints. Importantly, constraints like reconstruction and orthogonality on basis features can impede learning. The plots for moving average episode returns are available in Appendix J.6 for additional insights.

## 5 Learning Successor Features the Simple Way

The architecture for our network is shown in Figure 2, which is broadly inspired by Liu and Abbeel [2021]. Pixel-level observations, $S_t$, are fed into a convolutional encoder that outputs a latent representation $h(S_t)$, which is used both to construct the basis features and the SFs. To construct the basis features, $\phi(S_t)$, we simply normalize the latent representations $h$ (via $L2$ normalization, following Machado et al. [2020]). To calculate the representations $\psi(S_t, A_t, \boldsymbol{w})$, the latent representation is combined with the task encoding vector, $\boldsymbol{w}$, and fed into a multilayer perceptron that generates one set of representations for each possible action, $A_t$. These representations are then combined with the task encoding via a dot product operation to estimate the $Q$-value function, $Q(S_t, A_t, \boldsymbol{w}) = \psi(S_t, A_t, \boldsymbol{w})^\top \boldsymbol{w}$. The policy is then simply an $\epsilon$-greedy policy based on the $Q$-value function.

To learn the basis features ($\phi$) and representations ($\psi$), we minimize the losses in Eq. 8 and Eq. 9 using minibatch samples of experience tuples $(S_t, A_t, R_{t+1}, S_{t+1}, \boldsymbol{w})$, collected while interacting with the environment and sampled from a replay buffer which is similar to Mnih et al. [2015]. Critically, only the task-encoding vector $\boldsymbol{w}$ is learned by optimizing Eq. 8, so a stop gradient operator is applied to the basis features $\phi(S_t)$ (see Figure 2). The successor features, $\psi$, in the bootstrap target, $\hat{y}$, actually come from a target network, $\overline{\psi}$, which is updated periodically by using the actual network, a common approach in deep RL [Mnih et al., 2015]. The successor features $\psi$, and all of the upstream network parameters $\theta$, are learned by minimizing Eq. 9. The full algorithm used for training our network is given in Algorithm 1 in Appendix B.

## 6 Experimental results

The environments used in our studies are $10 \times 10$ 2D grid worlds, 3D Four Rooms environments (Figure 9 in Appendix D) and Mujoco. All studies were conducted exclusively using pixel observations, as the primary motivation for this paper is to address representation collapse when learning with pixel observations.

The grid worlds offer both egocentric (partially) and allocentric (fully observable) scenarios while the 3D Four Rooms environments provide exclusively egocentric observations. The rationale behind selecting these environments was threefold: first, to evaluate the agent's learning capabilities across varying levels of environmental visibility, second, to examine its ability to interpret spatial relation-

ships and distances, and third, to provide a set of tasks where the transition dynamics are easy to quantify for constructing SRs that can serve as a comparison to evaluate the SFs with.

For a more complex setting, we considered the Mujoco environment because it allows for direct manipulation of the reward function and domains switching, such as moving from half-cheetah to walker, given that they both have the same action dimensions.

To evaluate the generalization capabilities of the learned SFs, our studies focus on continual learning setting. In the 2D grid worlds and 3D Four Rooms environment, agents are exposed to two cycles of two distinct tasks. These tasks involves changes in reward locations (as shown in Figure 9b & Figure 9d) and/or changes in environment dynamics (as shown in Figure 9a & Figure 9e). Additionally, we explored two different scenarios to better simulate real-world conditions. The first scenario involves resetting the replay buffer at each task transition, which emulates drastic distribution shifts typically encountered in real-world applications. The second scenario maintains the replay buffer across task transitions, allowing us to assess the agent's learning continuity in a more stable data setting.

In the Mujoco environment, agents are exposed to one cycle of two distinct task as in this setting, we primarily wish to evaluate how well the agents can adapt to new tasks and mitigating interference.

In all experiments, we make comparisons with several baselines, namely, a Double Deep Q-Network (DQN) agent [Van Hasselt et al., 2016] and agents learning SFs ($\psi$) with constraints on their basis features ($\phi$), including reconstruction loss [Machado et al., 2020], orthogonal loss [Touati et al., 2022], and unlearnable random features [Touati et al., 2022]. Additionally, we compare with an agent that learns SFs through a non-task-engaged pre-training regime [Liu and Abbeel, 2021]. The mathematical definitions of the constraints can be found in Appendix F. To ensure the robustness of our results, all experiments are conducted across 5 different random seeds.

## 6.1 2D Grid world

The 2D Gridworld environments were developed based on 2D Minigrid [Chevalier-Boisvert et al., 2023]. We created two different layouts of the 2D Gridworld environment, namely Center-Wall (Figure 9a) and Inverted-LWalls (Figure 9b). In order to align the setting more closely with the canonical Gridworld environment as described by Sutton and Barto [2018], we altered the reward function such that it returns a reward of +1 when the agent successfully reaches the goal state and 0 otherwise. For the 2D Gridworld environments, the agents were trained for one million steps per task.

Figure 3a presents the cumulative returns for the Center-Wall environment with egocentric observations, while Figure 3b shows the results for allocentric observations.

The results show that our agent learns as well as, if not better than, the baseline models. Furthermore, when analysing the cumulative total returns during training, our model, SF Simple, exhibited better transfer that the baseline models. Particularly, SFs that are learned with constraints on the basis features clearly struggle to learn effectively, either due to the additional computational overhead or because representations that fulfill those constraints do not lead to effective policy learning.

## 6.2 3D Four Rooms environment

We developed the 3D Four Rooms environments (Figure 9d) using Miniworld [Chevalier-Boisvert et al., 2023]. In this environment, the state and action spaces are continuous. In the first task, the agent receives a reward of +1 when it reaches the green box and a reward of -1 when it reaches the yellow box and this alternates for the second task. The agents were trained for five million steps per task. Similarly, the results in Figure 3c show that our agent is able to learn effectively using egocentric pixel observations in a 3D environment.

## 6.3 Mujoco

In order to demonstrates our model's capabilities with continuous actions, we consider the Mujoco environment. We followed the established protocol in Yarats et al. [2021] for effective learning with pixels observations in this environment. We started in the half-cheetah domain, rewarding agents for running forward in Task 1. For Task 2, we introduced scenarios with running backwards, running faster, and switching to the walker domain. The results are presented in Figure 4.

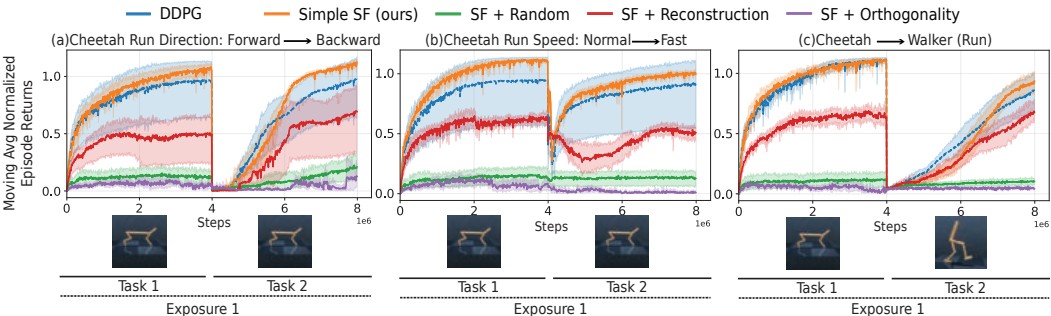

Figure 4: Continual Reinforcement Learning results using pixel observations in *Mujoco* environment across 5 random seeds. **Replay buffer resets at each task transitions** to simulate drastic distribution shifts. we started with the half-cheetah domain in Task 1 where agents were rewarded for running forward. We then introduced three different scenarios in Task 2: **(a)** agents were rewarded for running backwards, **(b)** running faster, and, in the most drastic change, **(c)** switching from the half-cheetah to the walker domain with a forward running task. To ensure comparability across these diverse scenarios, we normalized the returns, considering that each task has different maximum attainable returns per episode. We did not evaluate APS (Pre-train) here because it struggles in the Continual RL setting, even in simpler environments such as the 2D Minigrid and 3D Miniworld.

Across all scenarios, our model not only maintained high performance but consistently outperformed all baselines in both Task 1 and Task 2, highlighting its superior adaptability and effectiveness in complex environments. This contrasted sharply with other SF-related baseline models, which struggled to adapt under these conditions.

## 7 Analysis of Efficacy and Efficiency

### 7.1 Comparison to Successor Representations

Can our SF-learning technique, like traditional SRs, effectively capture the transition dynamics of the environment [Stachenfeld et al., 2017]? To investigate, we first sought a quantitative measure to compare SFs to SRs. To do this, we trained a simple non-linear decoder to assess which model's SFs can be most effectively decode into SRs. We conducted this evaluation using both allocentric and egocentric observations within the center-wall environment. The results, depicted in Figure 5, shows that our model demonstrate consistently high accuracy (lower errors) across both settings. This contrasts sharply with SFs developed using reconstruction constraints or random basis features, which, while effective in egocentric settings, perform poorly in allocentric settings where feature sparsity is greater.

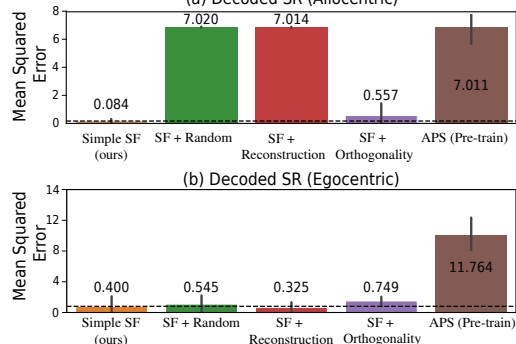

Figure 5: Decoding performance comparison of models' SFs into SRs using a non-linear decoder in the Center-Wall environment. Ground truth SRs are generated analytically using Eq. 21, described in Appendix N. Lower Mean Squared Error values on the y-axis indicate better performance.

We next utilized 2D visualizations with geospatial color mapping to differentiate environmental locations, aiming to see if similar SFs that are proximate in neural space are proximate in physical space. Using UMAP [McInnes et al., 2018] for dimension reduction, our results (Figure 6) suggest that our simple approach captures environmental statistics comparably to other models, but with less overhead for calculating the loss. Moreover, our technique consistently forms organized spatial clusters across partially, fully, and egocentric observational settings.

Additionally, we performed a correlation analysis in 2D Gridworld environments, comparing each spatial position and head direction against analytically computed SRs [Dayan, 1993], further confirm-

(a)

Center-Wall
(Fully-Obs.)

(b)

Four Rooms
(Egocentric)

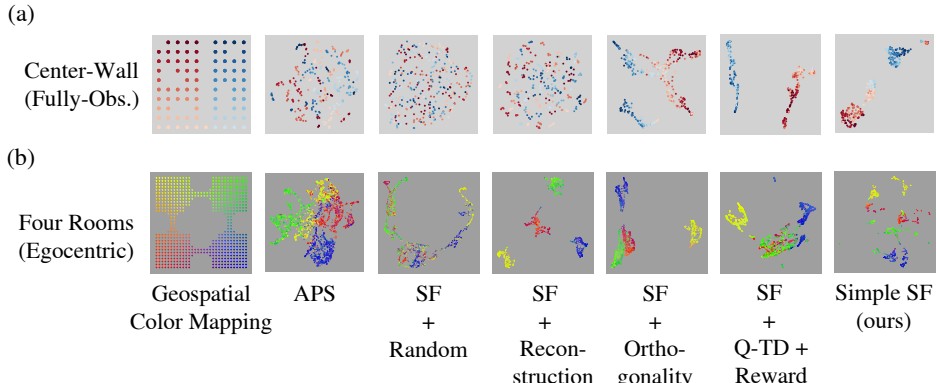

| Geospatial Color Mapping | APS | SF + Random | SF + Reconstruction | SF + Orthogonality | SF + Q-TD + Reward | Simple SF (ours) |

Figure 6: 2D visualization of Successor Features in **(a)** the fully-observable Center-Wall environment and **(b)** the 3D Four Rooms environment. Each row represents different models' visualizations post-training, starting with geospatial color mapping of the layout in the first column, followed by comparisons of SF-based models. Clustering indicates the capture of environmental statistics. Despite this, well-clustered SF models, especially those with orthogonality constraints, may not always translate to effective policy learning, as seen in Figure 3. In allocentric scenarios, SFs with reconstruction constraints struggle with minimal pixel variations, unlike in the distinct pixel changes in the Four Rooms environment. For more visualizations, see Appendix M.

ing the robustness and adaptability of our model's SFs in various observational contexts (Table 6 and Table 7 in Appendix N).

## 7.2 Is Stop Gradient critical for learning?

Previous methods that concurrently learn the basis features, $\phi$, and the task-encoding vector $w$, often face challenges with learning efficiency and stability, particularly in environments characterized by sparse rewards. This issue is illustrated in Figure 10 in Appendix D of Ma et al. [2020], where optimizing the reward prediction loss (Eq. 8) can inadvertently drive the basis features towards zero ($\phi \rightarrow \vec{0}$), causing significant representational collapse. Representational collapse not only reduces the discriminative capabilities of $\phi$ but also undermines the agent's ability to differentiate between distinct states, thus severely impacting the overall learning process.

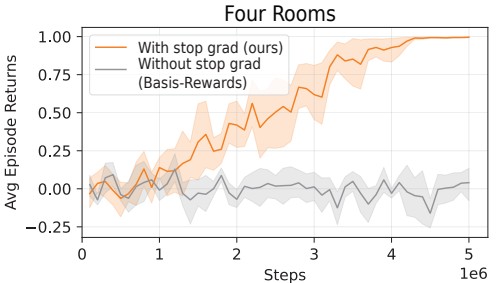

Figure 7: Efficacy of the Stop Gradient Operator in the Four Rooms Environment. Agents without a stop gradient operator exhibit degraded learning.

As depicted in Figure 2, our solution involves the strategic use of a stop gradient operator applied to the basis features $\phi$. This operator prevents the gradient from the reward prediction loss from updating basis features $\phi$, effectively decoupling the learning of $\phi$ from $w$, thus ensuring that it retains its critical discriminative statistics, allowing for effective learning as demonstrated in Figure 7.

## 7.3 Robustness to Stochasticity within the environment

How robust are the SFs learned using our approach as the environment dynamics become noisier? To explore this question and verify the robustness of our technique, we also created a slippery variant of the Four Rooms environment (Figure 9e). Specifically, in the top right and bottom left rooms, the agent experiences a "slippery" dynamic: chosen actions have a certain probability of being replaced with random, alternative actions. This design mimics the effects of a low-friction or slippery surface, creating a scenario where the agent's intended movements might lead to unpredictable outcomes.

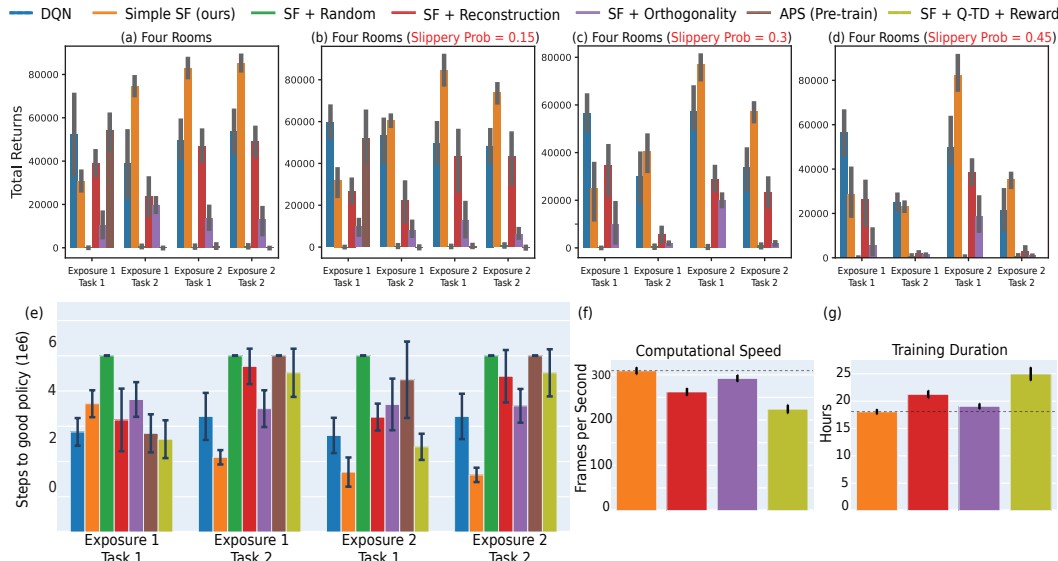

Figure 8: Efficiency analysis using 3D Slippery Four Rooms environment. **(a-d)**: Robustness analysis to increasing levels of stochasticity. **(e)** Bar plot showing efficiency in learning, measured as steps to achieve a policy that produces a reasonable level of performance, with low values indicating higher efficiency. **(f)** Bar plot showing frames per second achieved by the agent during gradient computation, back-propagation, and interaction with the environment. These metrics provide insights into the computational efficiency and the real-time interaction capabilities of the agent across different tasks or conditions. **(g)** Bar plot showing the total training duration for completing two exposures of two tasks, demonstrating overall time efficiency. Collectively, these plots reveal that our agent not only learns tasks effectively but also excels in computational efficiency.

The results in Figure 8a-d demonstrate that our approach is robust to increasing levels of stochasicity. Notably, when the stochasicity is high (slippery probability $>= 0.3$), all other SF methods fail to learn effectively in the second task, whereas our approach continues to perform well.

### 7.4 Efficiency analysis

How do alternative SF learning methods with extra loss functions, like orthogonality constraints, stack up against our approach in terms of efficiency? We analyzed the number of steps each method takes to learn an effective policy, using a performance threshold defined by the shortest expected episode length from the last 10 episodes. A shorter episode length indicates better performance, as it signifies quicker goal achievement. We noted the timestep when each model first met or exceeded this threshold. Our results, shown in Figure 8e, demonstrate that our method outperforms all baselines in learning efficiency. Additionally, our method leverages simpler compute blocks and loss functions, enhancing computational speed and reducing training duration, as shown by faster frame processing rates (Figure 8f) and shorter overall training times (Figure 8g). Therefore, our approach is more efficient than the baseline methods for learning SFs.

## 8 Discussion

In this work, we developed a method for learning SFs from pixel-level observations without pretraining or complex auxiliary losses. By applying the mathematical principles of SFs, our system effectively learns during task engagement using standard losses based on typical training returns and rewards. This simplicity and efficiency are key advantages of our approach.

Our experiments demonstrate that our method learns SFs effectively under various conditions and surpasses baseline models in continual RL scenarios. It effectively captures environmental transition dynamics and correlates well with analytically computed Successor Representations (SRs), offering a streamlined, efficient strategy for integrating SFs into RL models. Future work could build on this to create more sophisticated models that leverage SFs for enhanced flexibility in RL.

# 9 Limitations and Broader Impact

The algorithms we developed were evaluated predominantly in simulated environments, which may not capture the diverse complexity of real-world scenarios. A key assumption in our approach is that pixel observations are of good quality. This assumption is critical as poor image quality could substantially degrade the performance and applicability of our algorithms.

The use of Successor Features in learning algorithms, as demonstrated in this work, offers significant advantages, particularly in mitigating catastrophic interference. This capability is crucial for the development of machine learning systems that require continuous learning, such as in dynamic environments. For instance, autonomous vehicles operating in ever-changing conditions can retain learned knowledge while adapting to new information, enhancing their safety and reliability.

However, the enhanced capabilities of these systems also raise concerns. The ability of machine learning models to continuously adapt and learn can lead to challenges in predictability and control, potentially making outcomes less transparent. As systems become more autonomous and capable of adapting over time, there's a risk that errors or biases in the learning process could propagate more extensively before detection, especially if oversight does not keep pace with the rate of learning.

# 10 Acknowledgements

Raymond Chua was supported by the DeepMind Graduate Award and UNIQUE Excellence Scholarship (PhD). We extend our gratitude to the FRQNT Strategic Clusters Program (2020-RS4-265502 - Centre UNIQUE - Quebec Neuro-AI Research Center).

Arna Ghosh was supported by the Vanier Canada Graduate scholarship and Healthy Brains, Healthy Lives Doctoral Fellowship.

Blake A. Richards was also supported by NSERC (Discovery Grant RGPIN-2020- 05105, RGPIN-2018-04821; Discovery Accelerator Supplement: RGPAS-2020-00031; Arthur B. McDonald Fellowship: 566355-2022), Healthy Brains, Healthy Lives (New Investigator Award: 2b-NISU-8), and CIFAR (Canada AI Chair; Learning in Machine and Brains Fellowship).

This research was further enabled by computational resources provided by Calcul Québec [2] and the Digital Research Alliance of Canada [3], along with the computational resources support from NVIDIA Corporation.

We are grateful to Gheorghe Comanici, Pranshu Malviya, Xing Han Lu, Isabeau Prémont-Schwarz and the anonymous reviewers whose insightful comments and suggestions significantly enhanced the quality of this manuscript. Additionally, our discussions with members of the LiNC lab [4], Mila [5], and early collaborators from Microsoft Research (MSR) have been invaluable in shaping this research. Special thanks to Ida Momennejad, Geoff Gordon and Mehdi Fatemi from MSR for their substantial insights and contributions during the initial phases of this work.

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

# A  Appendix

This supplementary section provides detailed insights and additional information that supports the findings and methodology discussed in the main paper. Below is a brief overview of what each section contains:

# B  Pseudocode Implementation

---
**Algorithm 1** Learning Simple Successor Features Online
---
1: Initialize task encoding vector $\boldsymbol{w}$
2: Initialize SF $\psi_\theta$ network, SF $\overline{\psi_\theta}$ target network
3: **for** $t := 1$, T **do**
4:     Receive observation $S_t$ from environment
5:     $A_t \leftarrow \epsilon$-greedy using $Q(S_t, \cdot \mid \boldsymbol{w})$
6:     Send $A_t$ to receive $S_{t+1}$ and $R_{t+1}$ from environment
7:     $a' \in \underset{b}{\arg\max}\, \overline{\psi_\theta}(S_{t+1}, b, \boldsymbol{w})^\top \boldsymbol{w}$
8:     $\hat{y} = R_{t+1} + \gamma\overline{\psi_\theta}(S_{t+1}, a', \boldsymbol{w})^\top \boldsymbol{w}$
9:     $\phi \leftarrow$ L2 normalized output from the encoder of SF $\psi$ network
10:    $loss_{\psi_\theta} = (\psi_\theta(S_t, A_t, \boldsymbol{w})^\top \boldsymbol{w} - \hat{y})^2$
11:    $loss_w = (\phi^\top \boldsymbol{w} - R_{t+1})^2$
12:    Gradient descent on $\psi_\theta$ and $\boldsymbol{w}$
13: **end for**
---

# C  Proofs and Theoretical Justifications

In this section, we provide a proof sketch to show that minimizing the Q-SF-TD loss (Eq. 9) will also result in minimizing the canonical universal SF-TD loss [Borsa et al., 2018] for learning the SFs ($\psi(\cdot) \in \mathbb{R}^n$). For the sake of brevity, we consider a tabular RL setting, where state $s$ is the current state, $s'$ is the next state, $a$ is the current action, and $r$ is the reward of the transition tuple $(s, a, s', r)$ and as per defined in the main text, $\boldsymbol{w} \in \mathbb{R}^n$ is the task encoding vector and $\phi(\cdot) \in \mathbb{R}^n$ is the set of basis features.

Let $L_{\mathrm{SF}}$ be the canonical universal SF-TD loss [Borsa et al., 2018]:

$$L_{\mathrm{SF}} = \frac{1}{2}\left\| \phi(s') + \gamma\overline{\psi}(s', a', \boldsymbol{w})) - \psi(s, a, \boldsymbol{w}) \right\|^2 \tag{11}$$

where $a' = \underset{b}{\arg\max}\, Q(s', b, \boldsymbol{w}) = \underset{b}{\arg\max}\, \psi(s', b)^\top \boldsymbol{w}$ and $\gamma$ is the discount factor. We treat $\overline{\psi}(s', a', \boldsymbol{w})$ as part of the bootstrapped target: $\hat{y}_{\mathrm{SF}} = \phi(s') + \gamma\overline{\psi}(s', a', \boldsymbol{w})$, which results in semi-gradient methods [Sutton and Barto, 2018]. Subsequently, the gradient $\nabla_\psi$ for $L_{\mathrm{SF}}$ (Eq. 4) is defined

as:
$$\nabla_\psi L_{\text{SF}} = -\left(\phi(s') + \gamma\overline{\psi}(s', a', \boldsymbol{w}) - \psi(s, a, \boldsymbol{w})\right) \tag{12}$$

Next, as previously discussed in section 3, the Q-SF-TD loss $L_\psi$ which we used to learn the successor features ($\psi$) is defined as:
$$L_\psi = \frac{1}{2}\left\|\hat{y} - \psi(s, a, \boldsymbol{w})^\top \boldsymbol{w}\right\|^2 \tag{13}$$

where $\hat{y} = r + \gamma \max_{a'} \overline{\psi}(s', a', \boldsymbol{w})^\top \boldsymbol{w}$ is the bootstrapped target.

(Note: Eq. 13 and the bootstrapped target $\hat{y}$ is the same as Eq. 9 and Eq. 10 respectively, presented in Section 3 of the main text)

Following the same reasoning in Eq. 4, the gradient $\nabla_\psi$ for $L_\psi$ is defined as:

$$\nabla_\psi L_\psi = -\left(r + \gamma\overline{\psi}(s', a', \boldsymbol{w})^\top \boldsymbol{w} - \psi(s, a, \boldsymbol{w})^\top \boldsymbol{w}\right)\boldsymbol{w} \tag{14}$$

**Proposition 1** *Optimizing $\nabla_\psi L_\psi \simeq \boldsymbol{w}^\top \nabla_\psi L_{SF}\boldsymbol{w}$, where $L_{SF}$ is the canonical loss for universal successor features [Borsa et al., 2018].*

*Proof.* Now, assuming that for any given state $s$, the reward $r$ for state $s$ can be linearly decomposed into the dot product of the basis features $\phi(s)$ and the task encoding vector $\boldsymbol{w}$, as suggested by Sutton [1988], Dayan [1993], it follows that there exists an optimal set of basis features $\phi^*(s)$. This optimal set ensures that the reward $r$ can be accurately represented as the dot product of $\phi^*(\cdot)$ and the task encoding vector $\boldsymbol{w}$:
$$r = \phi^*(s')^\top \boldsymbol{w} \tag{15}$$

where $s'$ is the next state.

Thereafter, let us recall that the reward prediction loss $L_w$ is defined as:
$$L_w = \frac{1}{2}\left\|r - \phi(s')^\top \boldsymbol{w}\right\|^2 \tag{16}$$

(Note: This equation is the same as Eq. 8 presented in Section 3 of the main text.)

Substituting the assumption that we made in Eq. 15 into the reward prediction loss (Eq. 16),
$$\begin{aligned}
L_w &= \frac{1}{2}\left(r - \phi(s')^\top \boldsymbol{w}\right)^2 \\
&= \frac{1}{2}\left(\phi^*(s')^\top \boldsymbol{w} - \phi(s')^\top \boldsymbol{w}\right)^2 && \text{(Subst. } r = \phi^*(s')^\top \boldsymbol{w} \text{ following Eq. 15)} \\
&= \frac{1}{2}\left((\phi^*(s') - \phi(s'))^\top \boldsymbol{w}\right)^2 \\
&= \frac{1}{2}\left(\epsilon(s')^\top \boldsymbol{w}\right)^2 \tag{17}
\end{aligned}$$

Where $\epsilon(s')$ is the difference between $\phi^*(s')$ and $\phi(s')$. Furthermore, if $L_w \simeq 0$, then $\epsilon(s')^\top \boldsymbol{w} = \boldsymbol{w}^\top \epsilon(s') \simeq 0$.

Shifting our focus back to the gradient $\nabla_\psi L_\psi$ of our Q-SF-TD loss function (Eq. 14),
$$\begin{aligned}
\nabla_\psi L_\psi &= -\left(r + \gamma\overline{\psi}(s', a', \boldsymbol{w})^\top \boldsymbol{w} - \psi(s, a, \boldsymbol{w})^\top \boldsymbol{w}\right)\boldsymbol{w} && \text{(Eq. 14)} \\
&= -\boldsymbol{w}^\top\left(\phi^*(s') + \gamma\overline{\psi}(s', a', \boldsymbol{w}) - \psi(s, a, \boldsymbol{w})\right)\boldsymbol{w} && \text{(Subst. } r = \phi^*(s')^\top \boldsymbol{w} \text{ following Eq. 15)} \\
&= -\boldsymbol{w}^\top\left(\phi(s') + \epsilon(s') + \gamma\overline{\psi}(s', a', \boldsymbol{w}) - \psi(s, a, \boldsymbol{w})\right)\boldsymbol{w} && \text{(Subst. } \phi^*(s') = \phi(s') + \epsilon(s') \text{ from Eq. 17)} \\
&= -\boldsymbol{w}^\top\left(-\nabla_\psi L_{\text{SF}} + \epsilon(s')\right)\boldsymbol{w} && \text{(Subst. definition from Eq. 12)} \\
&= \boldsymbol{w}^\top \nabla_\psi L_{\text{SF}}\boldsymbol{w} - \boldsymbol{w}^\top \epsilon(s')\boldsymbol{w} \\
&= \boldsymbol{w}^\top \nabla_\psi L_{\text{SF}}\boldsymbol{w} - 2\sqrt{L_w}\boldsymbol{w} && \text{(Subst.} \boldsymbol{w}^\top \epsilon(s') = 2\sqrt{L_w} \text{ from Eq. 17)} \\
&\simeq \boldsymbol{w}^\top \nabla_\psi L_{\text{SF}}\boldsymbol{w} && \square \tag{18}
\end{aligned}$$

In conclusion, this proof demonstrates that the gradients $\nabla_\psi L_\psi$ computed using our proposed Q-SF-TD loss function (Eq. 13) effectively project the gradients $\nabla_\psi L_{\text{SF}}$ from the canonical universal SF-TD loss function (Eq. 12) onto the task encoding vector $\boldsymbol{w}$. This indicates that our loss function maintains the essential characteristics of the canonical form while aligning closely with the specific direction of the task encoding vector $\boldsymbol{w}$.

# D  Environments

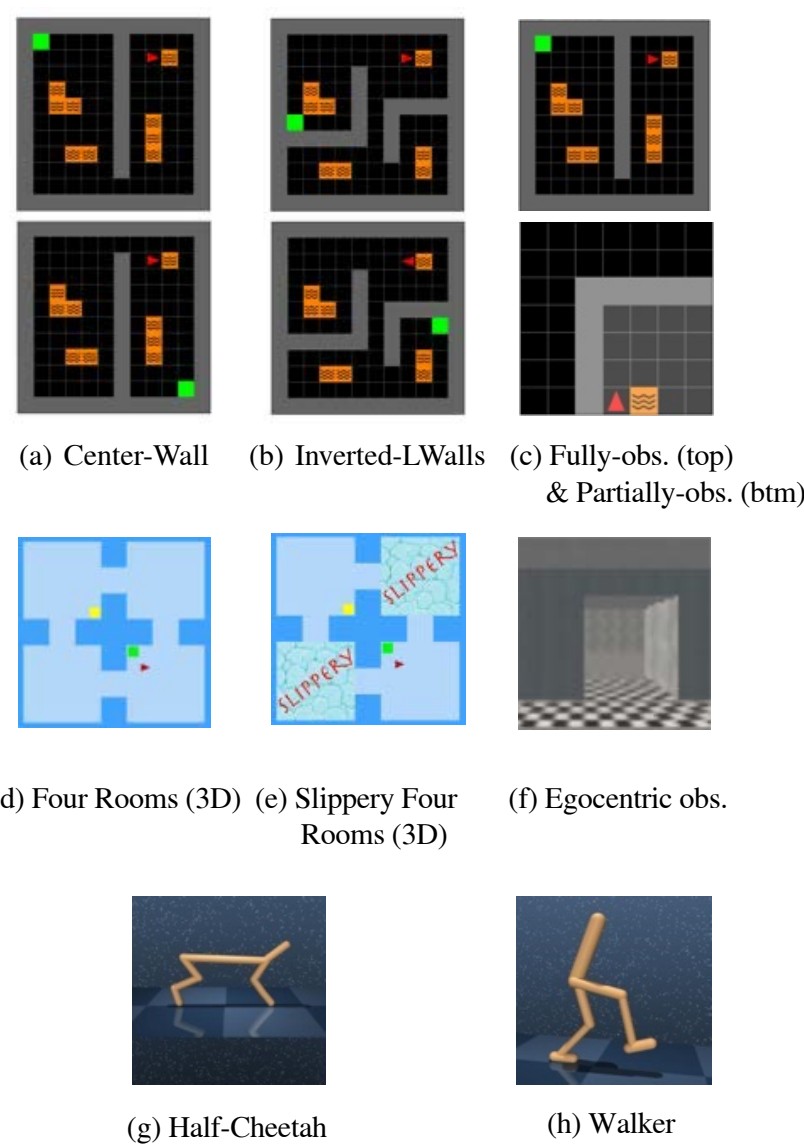

 (a) Center-Wall (b) Inverted-LWalls (c) Fully-obs. (top)
& Partially-obs. (btm)

 (d) Four Rooms (3D) (e) Slippery Four
Rooms (3D) (f) Egocentric obs.

 (g) Half-Cheetah (h) Walker

Figure 9: Overview of 2D Minigrid, 3D Four Rooms environments and Mujoco used in our studies with changing dynamics and rewards. Both 2D Minigrid and 3D Four Rooms environments utilize discrete actions while Mujoco utilizes continuous actions. All studies in this paper were done using only pixel observations.

# E  Experimental details

In this section, we provide more details about the environments used in our experiments.

## E.1  2D Gridworld Environments.

The specific parameters defining the 2D Gridworld environments are detailed in Table 1.

Table 1: 2D Minigrid Environment Specific Parameters

| PARAMETER | VALUE |
| --- | --- |
| GRID SIZE | $10 \times 10$ |
| OBSERVATION TYPE | FULLY-OBSERVABLE & PARTIALLY-OBSERVABLE |
| FRAME STACKING | NO |
| RGB OR GREYSCALING | RGB |
| NUM TRAINING FRAMES PER TASK | 1 MILLION FRAMES |
| NUM EXPOSURE | 2 |
| NUM TASK PER EXPOSURE | 2 |
| NUM FRAMES PER EPOCH PER TASK | 10K |
| BATCH SIZE | 256 |
| $\epsilon$ DECAY | 20K FRAMES |
| ACTION REPEAT | NO |
| ACTION DIMENSION | 3 |
| OBSERVATION SIZE | $84 \times 84 \times 3$ |
| MAX FRAMES PER EPISODE | 400 |
| TASK LEARNING RATE | 0.0001 |

### E.1.1  Center-Wall environment

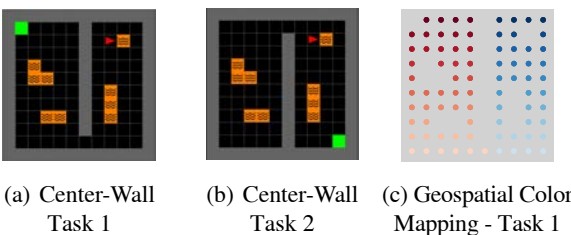

(a) Center-Wall Task 1  (b) Center-Wall Task 2  (c) Geospatial Color Mapping - Task 1

Figure 10: Center-Wall environment and Geospatial Color Mapping

In the Center-Wall environment, a vertical wall splits the area into two distinct regions. Task 1 features a passage from the left to the right side at the bottom, with the goal state located in the top left corner Figure 10a). In Task 2, the layout is modified: the passage is moved to the top, while the goal state is relocated to the bottom right corner Figure 10b). These changes are strategically implemented to evaluate the agents' ability to adapt to simultaneous alterations in both the environmental structure and the goal location. To aid in visual analysis, we use a geospatial color mapping initially developed for Task 1 (Figure 10c). This mapping effectively illustrates the spatial positioning within the environment and is particularly useful in the 2D visualization of the Successor Features and DQN Representations, providing a clearer understanding of how agents interpret and navigate the modified environment (Figures 6, 29 and 30).

### E.1.2  Inverted-Lwalls environment

In the Inverted-Lwalls environment, we placed two L-shaped walls within the environment, one on the left and the other on the right, creating a unique layout. This design results in a single, central path acting as a bottleneck, which the agent must navigate to reach the goal states. Specifically, to access the goal state located on the left side of the environment, the agent is required to traverse this central path while facing north (Figure 11a). Conversely, to reach the goal state situated on the right,

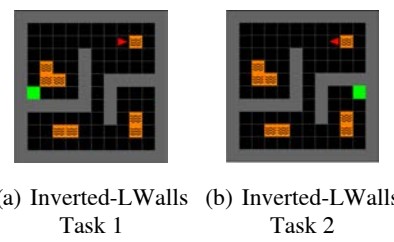

(a) Inverted-LWalls
Task 1

(b) Inverted-LWalls
Task 2

Figure 11: Inverted-Lwalls environment

the agent must navigate the same path but facing south (Figure 11b). This layout ensures that the agent consistently encounters and must maneuver this bottleneck area, regardless of the goal state's location.

## E.2    3D Miniworld Environments.

The actions in this environment consists of moving Forward and Backwards, turning Left and Right. The specific parameters defining the 3D Miniworld environments are detailed in Table 2.

Table 2: 3D Miniworld Four Rooms Environment Specific Parameters

| PARAMETER | VALUE |
|---|---|
| OBSERVATION TYPE | EGOCENTRIC |
| FRAME STACKING | NO |
| RGB OR GREYSCALING | RGB |
| NUM TRAINING FRAMES PER TASK | 5 MILLION FRAMES |
| NUM EXPOSURE | 2 |
| NUM TASK PER EXPOSURE | 2 |
| NUM FRAMES PER EPOCH PER TASK | 100K |
| BATCH SIZE | 32 |
| $\epsilon$ DECAY | 1 MILLION FRAMES |
| ACTION REPEAT | NO |
| ACTION DIMENSION | 4 |
| OBSERVATION SIZE | $84 \times 84$ |
| MAX FRAMES PER EPISODE | 4000 |
| TASK LEARNING RATE | 0.001 |
| SLIPPERY PROBABILITY | $\{0.15, 0.3, 0.45, 0.6\}$ |

### E.2.1    Four Rooms environment

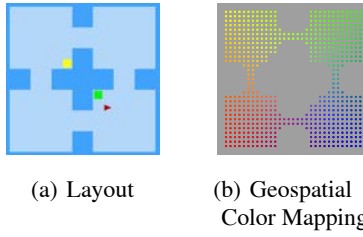

(a) Layout

(b) Geospatial
Color Mapping

Figure 12: Four Rooms (3D)

The Four Rooms environment consists of four identical square rooms arranged in a 2x2 grid, with passages connecting the rooms and allowing an agent to move between the rooms (Figure 12a). Each room in our 3D environment is designed with unique textures, a deliberate choice to reduce the complexity associated with localization ambiguities often encountered in more uniform settings. This variation in textures aids the agent in distinguishing between rooms based solely on visual cues,

thereby simulating more realistic navigation scenarios. This setup also allows us to observe how visual diversity impacts the agent's ability to infer its location and navigate to specific goals, providing insights into the interplay between environmental features and SFs learning in a 3D spatial context. Depending on the task, the agent receives a reward of either +1 or -1 when it reaches the yellow or green box. Similar to the Center-Wall environment, we also create a geospatial color mapping for the 2D visualization of the Successor Features and DQN Representations (Figure 12b).

### E.2.2 Slippery Four Rooms environment

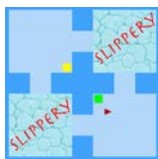

Figure 13: Slippery Four Rooms (3D) layout

In the slippery variant of the Four Rooms environment, our goal is to rigorously test the robustness of agents in learning SFs under challenging conditions. Specifically, in the top right and bottom left rooms of this setup, the agent experiences a 'slippery' dynamic: chosen actions have a certain probability of being replaced with random, alternative actions. This design mimics the effects of a low-friction or slippery surface, creating a scenario where the agent's intended movements might lead to unpredictable outcomes. Such a setup is instrumental in assessing the agent's adaptability and the robustness of SF learning in the face of environmental unpredictability. This variant not only challenges the agent to adapt to unexpected changes but also provides valuable insights into the flexibility and resilience of the SFs when navigating environments where control and predictability are compromised.

### E.3 Mujoco

In this work, we only utilised pixels inputs from Mujoco since our focus is on learning SFs directly from pixel observations. For domains, we chose both walker and half-cheetah. We broadly follow the same setup as Yarats et al. [2021], and included their model as a baseline, which we denote as "DDPG" in our results (Figure 4).

The codebase from their model is provided in the Unsupervised Reinforcement Learning (URL) Benchmark repository[Laskin et al., 2021][6], which we further described in the APS Agent in section F. The specific parameters we used for training in the Mujoco environment are detailed in Table 3.

Table 3: Mujoco Environment Specific Parameters

| PARAMETER | VALUE |
|---|---|
| FRAME STACKING | YES |
| RGB OR GREYSCALING | RGB |
| NUM TRAINING FRAMES PER TASK | 2 MILLION FRAMES |
| NUM EXPOSURE | 1 |
| NUM TASK PER EXPOSURE | 2 |
| ACTION REPEAT | 2 |
| BATCH SIZE | 256 |
| FEATURE DIM | 128 |
| HIDDEN DIM | 1024 |
| OBSERVATION SIZE | $84 \times 84$ |
| MAX FRAMES PER EPISODE | 4000 |
| SF DIM | 64 |
| TASK LEARNING RATE | 0.00001 |
| TASK UPDATE FREQUENCY | 10 |

---

[6]https://github.com/rll-research/url_benchmark

# F   Agents

In this section, we describe how we create our agent as well as the ones we used for comparisons. In addition, we provide the mathematical definitions of the constraints used on the basis features. For all agents, we swept the learning rates for both the SF network and the task encoding (specific for all SFs agents) using a gridsearch. The values ranged from 1e-1 to 1e-6, and the process was repeated using 5 random seeds in both 2D Gridworld and 3D Four Rooms environments. The same was also applied to the Double DQN agent [Van Hasselt et al., 2016] and we took extra care to ensure that the architecture and its number of parameters were as similar as possible to our model. Detailed hyperparameters for learning SFs and the task encoding $w$ for our agent are outlined in Tables 4 and 5.

## F.1   APS Agent

In our study, we take inspiration from the neural network architecture from Liu and Abbeel [2021] from the Unsupervised Reinforcement Learning (URL) Benchmark repository[Laskin et al., 2021][7], which utilizes PyTorch [Paszke et al., 2019]. This repository was chosen for its robust implementation and served as the foundation for all SF-variant agents, including ours. Within the URL Benchmark, the encoder follows the Deep Deterministic Policy Gradient (DDPG) network architecture [Lillicrap et al., 2015]. Notably, there is a discrepancy in the network architecture hyperparameters between the APS paper [Liu and Abbeel, 2021]) and the URL Benchmark repository. Given the practical implications of these differences, our implementation aligns with the hyperparameters specified in the URL Benchmark.

In line with the URL Benchmark's methodology, we initially employed the least squares method to determine the optimal task encoding $w$. However, we observed that this analytical approach was excessively sensitive in our experimental context, particularly due to its reliance on the mini-batch samples. This sensitivity was especially pronounced in environments with sparse rewards, like those in our study, suggesting that the least squares method might be less suited for such settings. This challenge was not present in the original APS framework [Liu and Abbeel, 2021], which was structured around distinct pre-training and fine-tuning phases. In contrast, our research focuses exclusively on continuous online learning, introducing unique challenges and dynamics not addressed in the APS paper [Liu and Abbeel, 2021].

## F.2   Reconstruction constraints

At each time step $t$, the basis features $\phi(S_t)$ are generated from the current state $S_t$ using an encoder. Together with the action $A_t$, these features are fed into a reconstruction decoder to predict the next state $\hat{S}_{t+1}$. Both the encoder and decoder are optimized using the reconstruction loss:

$$L_{recon} = ||S_{t+1} - \hat{S}_{t+1}||^2 \tag{19}$$

where $S_{t+1}$ is the ground truth of the next state. The same set of basis features $\phi$ is also utilized in optimizing the Reward Prediction Loss (Eq. 8) and the Q-SF-TD Loss (Eq. 9).

## F.3   Orthogonality constraints

At each time step $t$, the basis features $\phi$ are generated from the current state $S_t$ using an encoder. Besides being utilized to optimize the Reward Prediction Loss (Eq. 8) and the Q-SF-TD Loss (Eq. 9), the basis features $\phi$ are also optimized with the orthogonality loss [Koren, 2003, Mahadevan and Maggioni, 2007, Machado et al., 2017b,a]:

$$L_{ort} = \mathbb{E}_{(S_t, S_{t+1}) \sim \mathcal{D}} \left[ \|\phi(S_t) - \phi(S_{t+1})\|^2 \right] + \lambda \mathbb{E}^2_{\substack{s \sim \mathcal{D} \\ s' \sim \mathcal{D}}} \left[ \left( \phi(s)^\top \phi(s') \right)^2 - \|\phi(s)\|^2 - \|\phi(s')\|^2 \right] \tag{20}$$

---

[7] https://github.com/rll-research/url_benchmark

Table 4: Simple SF Hyperparameters

| PARAMETER | VALUE |
|---|---|
| OPTIMIZER | ADAM [KINGMA AND BA, 2014] |
| DISCOUNT($\gamma$) | 0.99 |
| REPLAY BUFFER SIZE | 100K |
| DOUBLE Q | YES [VAN HASSELT ET AL., 2016] |
| TARGET NETWORK: UPDATE PERIOD | 1000 |
| TARGET SMOOTHING COEFFICIENT | 0.01 |
| MULTI-STEP RETURN LENGTH | 10 |
| MIN REPLAY SIZE FOR SAMPLING | 5000 |
| FRAMESTACKING | NO |
| REPLAY PERIOD EVERY | 16 FRAMES |
| EXPLORATION | $\epsilon$-GREEDY |
| LEARNING RATE | 0.0001 |
| RESET BUFFER WHEN TASK SWITCHES | NO |
| ENCODER CHANNELS | {32, 32, 32, 32} |
| ENCODER KERNEL SIZE | {3, 3, 3, 3} |
| ENCODER STRIDE | {2, 1, 1, 1} |
| ENCODER NON-LINEARITY | ReLU |
| BASIS FEATURES $\phi$ | L2-NORMALIZE (OUTPUT OF ENCODER) |
| FEATURE $\phi$ DIMENSION | 256 |
| FEATURES-TASK NETWORK HIDDEN UNITS | 256 |
| FEATURES-TASK NETWORK NORMALIZATION | LAYER-NORM |
| FEATURES-TASK NETWORK NON-LINEARITY | TANH |
| SF $\psi$ DIMENSION | 256 |
| SF $\psi$ NETWORK HIDDEN UNITS | {256, 256, SF $\psi$ DIM X ACTION DIM} |
| SF $\psi$ NETWORK NON-LINEARITY | ReLU |

Table 5: Task $w$ encoding Hyperparameters

| PARAMETER | VALUE |
|---|---|
| TASK $w$ DIMENSION | 256 |
| TASK $w$ LEARNING RATE | ENVIRONMENT-DEPENDENT (SEE TABLE 1 & 2) |
| TASK $w$ OPTIMIZER | ADAM [KINGMA AND BA, 2014] |

where states $s$ and $s'$ are two different states sampled from the replay buffer $\mathcal{D}$. The first term encourages the basis features $\phi(S_t)$ and $\phi(S_{t+1})$ to be similar and the second term promotes orthogonality by ensuring that the basis features of the different states $\phi(s)$ and $\phi(s')$ are distinct and decorrelated. Following Touati et al. [2022], we set the regularization factor $\lambda = 1$.

### F.4 Random constraints

In this agent, the basis features $\phi$ are constrained to be unlearnable random features, which are defined during initialization. The SFs $\psi$ are subsequently learned on top of these predefined basis features. To guarantee that the basis features $\phi$ remain unlearnable throughout the training process, a stop gradient operator is employed.

### F.5 Learning SFs through integrating all losses

This agent learns Successor Features using a complex learning strategy that integrates three distinct losses: the SF-TD loss (Eq. 4), the reward prediction loss (Eq. 8) and the Q-SF-TD loss (Eq. 9). This multifaceted approach, proposed by Janz et al. [2019] aims to ensure that the learnt SFs satisfy all desired constraints.

# G  Our Architecture for Continuous Control

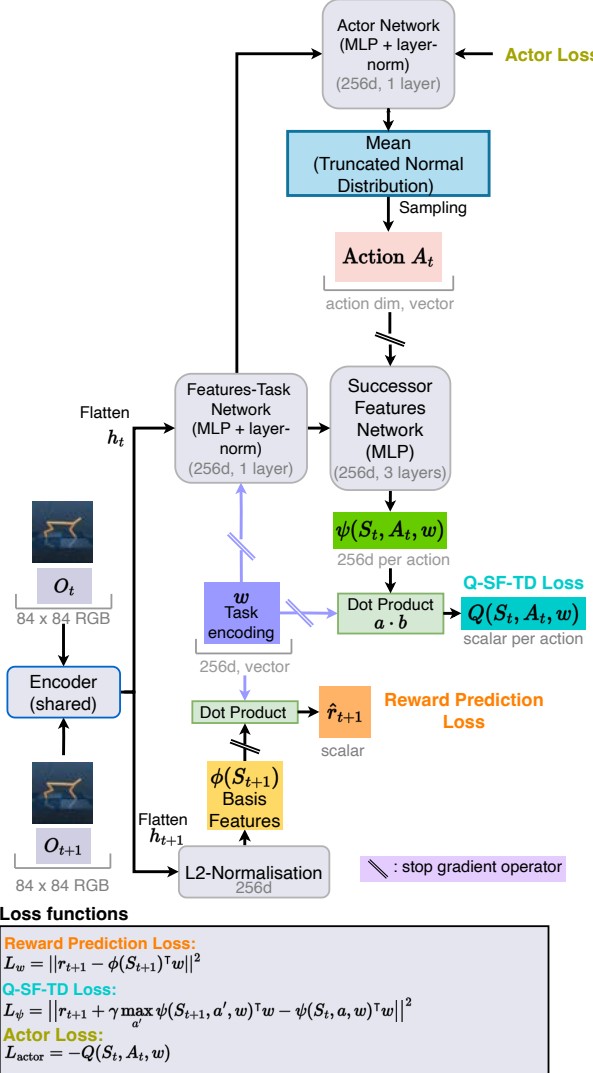

Figure 14: Our model adapted for continuous action spaces, based on the Actor-Critic architecture commonly used in DDPG [Lillicrap et al., 2015] and implemented following the URL benchmark [Laskin et al., 2021]. This design modifies our original architecture to accommodate continuous action environments, enabling the model to handle a broader range of control tasks. The model incorporates a linear decomposition of Successor Features $\psi$ and the task encoding vector $w$ to compute Q-value, following Eq. 1. Following the DDPG implementation in URL benchmark, actions are sampled from a truncated normal distribution, and LayerNorms are applied to normalize inputs to a unit distribution.

# H    Models of Previous Approaches

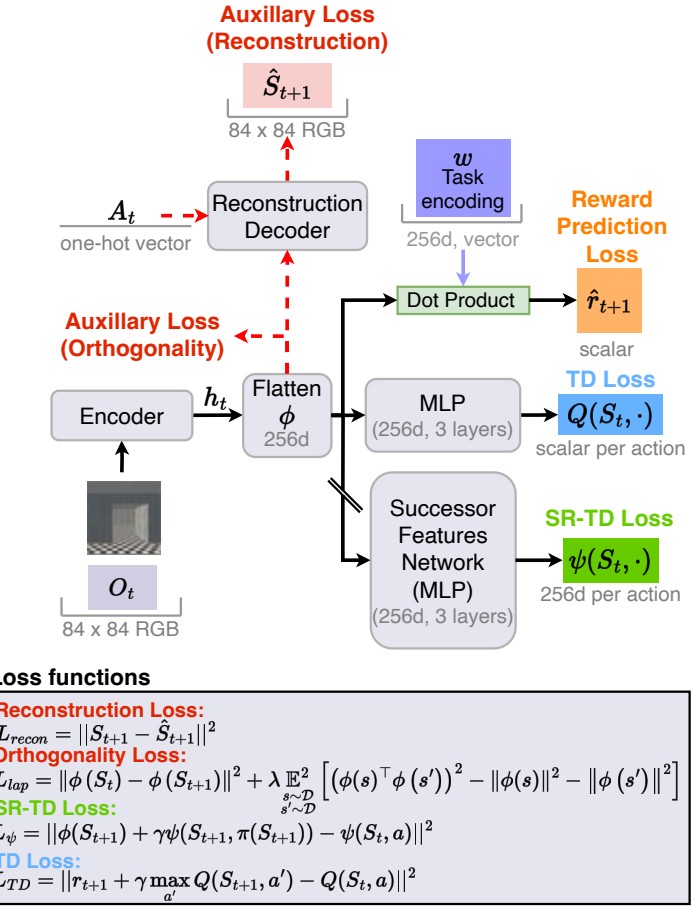

**Loss functions**

**Reconstruction Loss:**
$L_{recon} = ||S_{t+1} - \hat{S}_{t+1}||^2$
**Orthogonality Loss:**
$L_{lap} = \|\phi(S_t) - \phi(S_{t+1})\|^2 + \lambda \mathop{\mathbb{E}}_{\substack{s \sim \mathcal{D} \\ s' \sim \mathcal{D}}}^2 \left[ \left( \phi(s)^\top \phi(s') \right)^2 - \|\phi(s)\|^2 - \|\phi(s')\|^2 \right]$
**SR-TD Loss:**
$L_\psi = ||\phi(S_{t+1}) + \gamma \psi(S_{t+1}, \pi(S_{t+1})) - \psi(S_t, a)||^2$
**TD Loss:**
$L_{TD} = ||r_{t+1} + \gamma \max_{a'} Q(S_{t+1}, a') - Q(S_t, a)||^2$

Figure 15: In order to prevent representation collapse in the basis features $\phi$, previous methods on learning SFs from pixel observations often relied on an additional loss, such as reconstructing the state of the next time step $\hat{S}_{t+1}$ after executing action $A_t$ [Machado et al., 2020]. Recent approaches in learning SFs include encouraging orthogonal representations in the basis features [Touati et al., 2022]. A stop gradient operator is also used to prevent the SFs from updating the basis features $\phi$ when optimizing the SF-TD loss. [Kulkarni et al., 2016]

# I   Impact of Learning Rate Variations on Task Encoding Vector

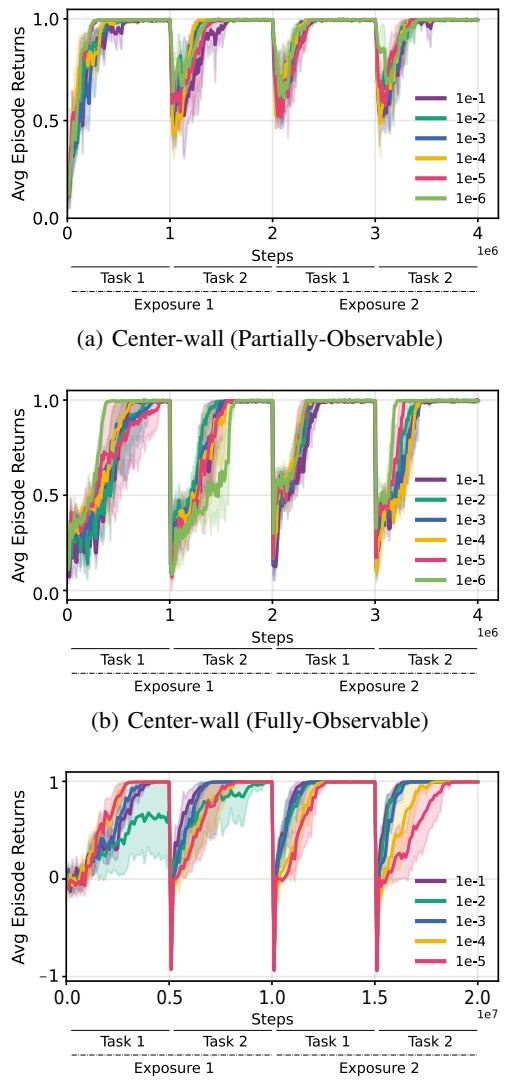

(a) Center-wall (Partially-Observable)

(b) Center-wall (Fully-Observable)

(c) 3D Four Rooms

Figure 16: Comparison of learning rates for the task encoding vector in grid worlds and 3D Four Rooms environments. Generally, a lower learning rate is required for the task encoding vector, despite its use of a simple reward prediction loss (Eq. 8), compared to the SF network, which needs more steps to converge due to its involvement in capturing complex environmental dynamics.

# J Further Experimental Results

In this section, we present expanded illustrations of the results initially introduced in the main paper. These larger visual figures provide a clearer and more detailed view to enhance the reader's understanding of our findings. Additionally, we include additional supplementary experimental results that were not featured in the main paper due to space limitations.

## J.1 Single task results for 2D Minigrid and 3D Four Room environment

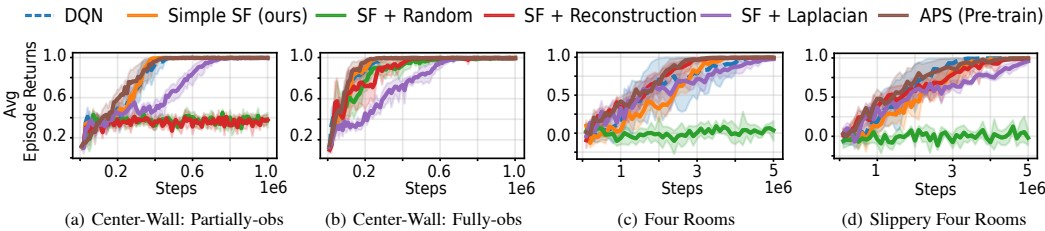

Figure 17: Performance of agents trained on a single task in both 2D Minigrid and 3D Four Rooms environments across 5 random seeds. The Y-axis represents the moving average of the average episode returns. Our model, Simple SF (orange), performs comparably to DQN (blue), even though it learns two functions—Successor Features (SFs) and the task encoding vector—while DQN only learns a single function, the Q-value.

## J.2 Continual RL results for Inverted-LWalls environment

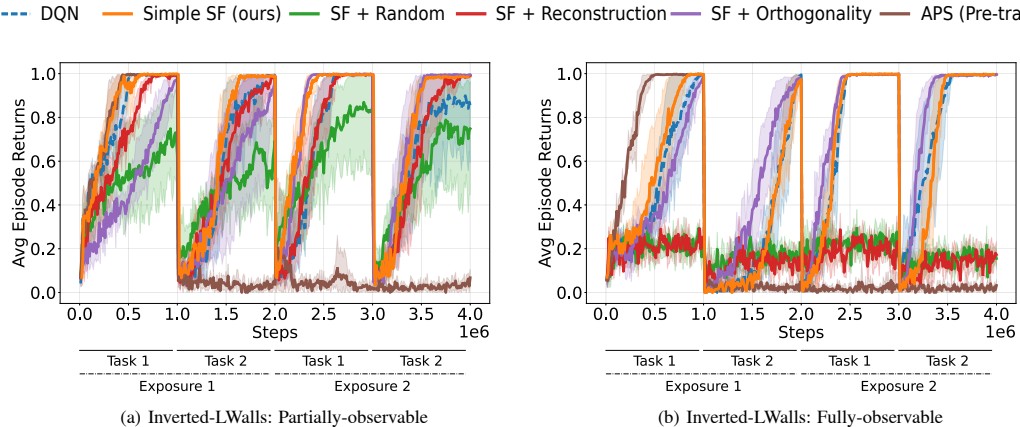

Figure 18: Evaluation in a Continual Reinforcement Learning setting across 5 random seeds, **without replay buffer resets at each task transition** in the Inverted L-Walls environment. Here, the goal location alternates between the left and right sides with each task change, while the environment dynamics remain constant. **(a)** In the partially-observable scenario, our agent demonstrates a faster re-learning ability for new tasks compared to other agents. **(b)** In the fully-observable scenario, while our agent shows performance comparable to the DQN agent, it is slightly outperformed by the agent employing SFs with orthogonality constraints on its basis features. Notably, despite the superior performance of this latter agent in later tasks during Exposure 2, it initially faces difficulties in developing an effective policy, attributed to the added complexity of adhering to orthogonality constraints.

## J.3 Continual RL results for Center-Wall environment

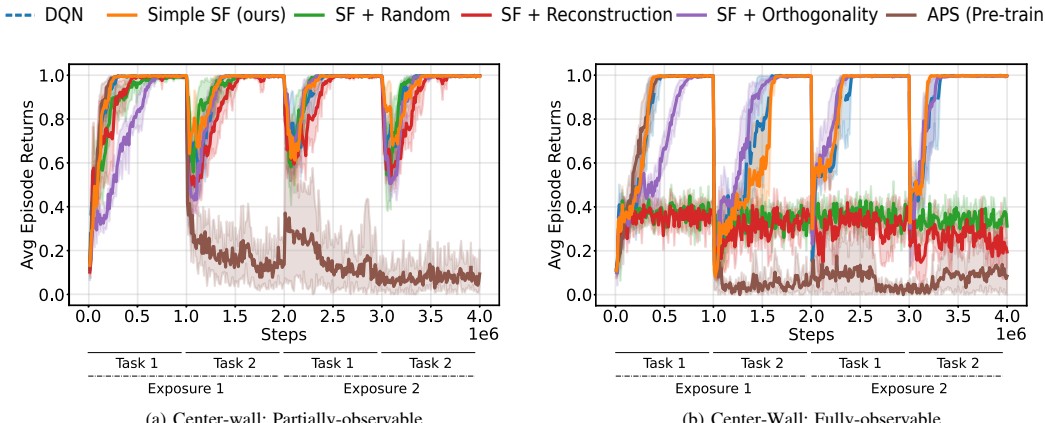

(a) Center-wall: Partially-observable          (b) Center-Wall: Fully-observable

Figure 19: Evaluation in a Continual Reinforcement Learning setting across 5 random seeds, **without replay buffer resets at each task transition** in the Center-Wall environment. In this setup, both the goal location and environment dynamics change with each task switch. **(a)** In the partially-observable scenario, our agent demonstrates performance comparable to that of other agents. **(b)** In the fully-observable scenario, our agent outperformed all others, with the agent employing SFs with orthogonality constraints on its basis features coming in as a close second. Notably, while this latter agent shows improved performance in later tasks of Exposure 2, it initially encounters difficulties in developing an effective policy, which can be attributed to the added complexity of adhering to orthogonality constraints.

## J.4 Continual RL results for Four Rooms environment

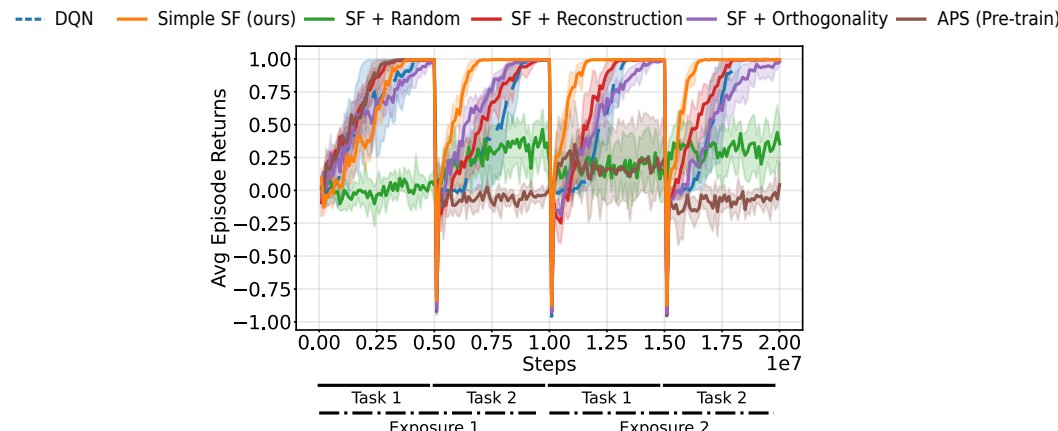

Figure 20: Evaluation in a Continual Reinforcement Learning setting across 5 random seeds, **without replay buffer resets at each task transition** in the 3D Four Rooms environment.

## J.5 Continual RL results for Slippery Four Rooms environment

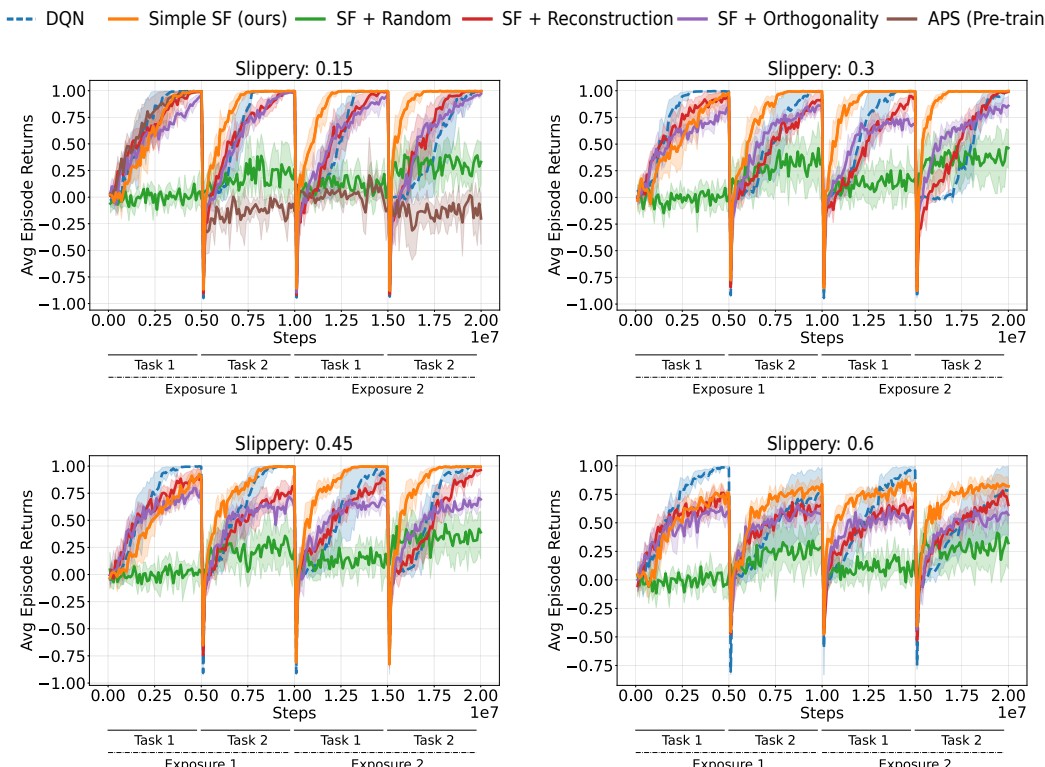

Figure 21: Evaluation in a slippery Four-Rooms environment with varied slipperiness probabilities, **without replay buffer resets at each task transition**. This environment features slippery conditions in the top-right and bottom-left rooms for both tasks, Task 1 and Task 2. Both tasks have differing reward structures: In Task 1, rewards are set at +1 for the green box and -1 for the yellow box; in Task 2, this reward scheme is reversed (green box: -1, yellow box: +1). The diagram illustrates the layout of the environment can be found in Figure 9. Note: The APS Pre-trained agent was tested only at a slippery probability of 0.15; higher probabilities were not evaluated due to performance decline beyond Task 1 of Exposure 1 when the slippery probability is 0.15.

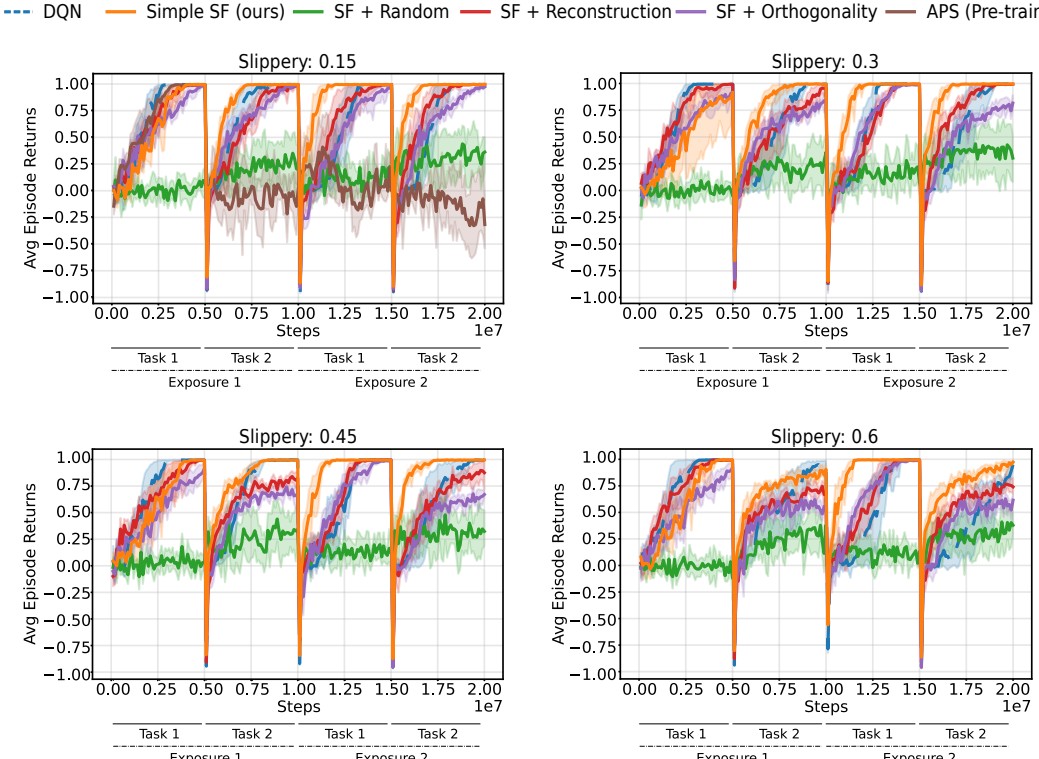

Figure 22: Evaluation in a Continual Reinforcement Learning setting across 5 random seeds in the Four-Rooms Environment with environmental changes, **without replay buffer resets at each task transition**. Task 1 adheres to the canonical Four Rooms environment dynamics, while Task 2 employs the slippery variant, where chosen actions are altered based on the slippery probability to simulate environmental changes. Throughout both tasks, reward associations remain consistent: +1 for the green box and -1 for the yellow box. The layout of this environment is depicted in Figure 9. Note: The APS Pre-trained agent was tested only at a slippery probability of 0.15; higher probabilities were not evaluated due to performance decline beyond Task 1 of Exposure 1 when the slippery probability is 0.15.

## J.6 Continual RL results for 2D Minigrid and 3D Four Rooms environment with Replay resets

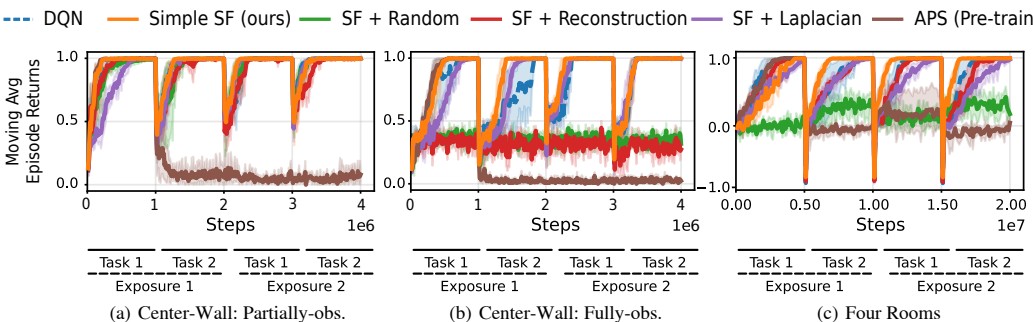

(a) Center-Wall: Partially-obs.    (b) Center-Wall: Fully-obs.    (c) Four Rooms

Figure 23: Continual Reinforcement Learning Evaluation with pixel observations in 2D Minigrid and 3D Four Rooms enviroment. **Replay buffer resets at each task transitions** to simulate drastic distribution shifts: Agents face two sequential tasks (Task 1 & Task 2), each repeated twice (Exposure 1 & Exposure 2). Moving average episode returns using most recent episodes in both egocentric and allocentric 2D Minigrid environments and egocentric 3D Four Rooms environment.

# K   Experimental results of SF + Q-TD + Reward vs SF Simple (Ours)

In this section, we present the experimental results of our agent (SF Simple) and the agent which optimizes the three losses (SF + Q-TD + Reward) simultaneously. For more information about this agent, see section F.5.

## K.1   Continual RL results for Inverted-LWalls environment

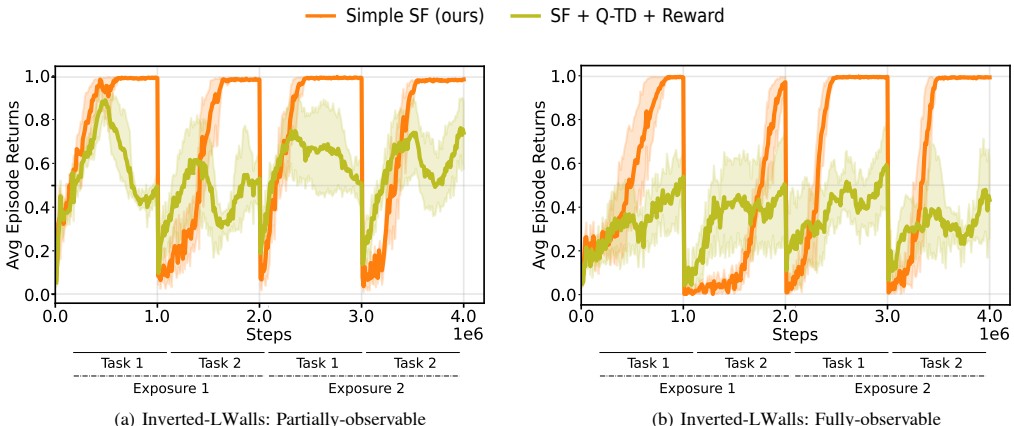

(a) Inverted-LWalls: Partially-observable      (b) Inverted-LWalls: Fully-observable

Figure 24: Evaluation in a Continual Reinforcement Learning setting across 5 random seeds, **with replay buffer resets at each task transition** in the Inverted L-Walls environment. Here, the goal location alternates between the left and right sides with each task change, while the environment dynamics remain constant. **(a & b)** In both partially-observable and fully-observable scenario, the agent, which optimizes three losses simultaneously (SF + Q-TD + Reward), experiences learning instabilities due to the higher complexity involved in managing all constraints.

## K.2    Continual RL results for Center-Wall environment

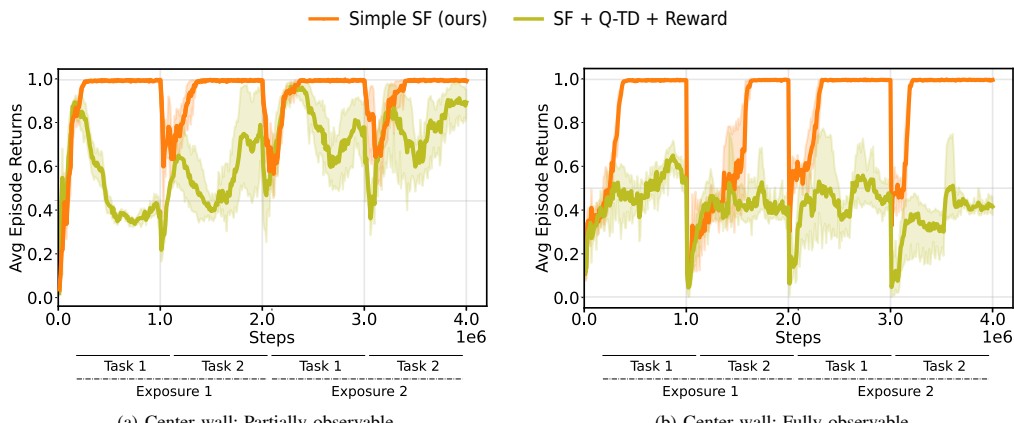

(a) Center-wall: Partially-observable

(b) Center-wall: Fully-observable

Figure 25: Evaluation in a Continual Reinforcement Learning setting across 5 random seeds, **with replay buffer resets at each task transition** in the Center-Wall environment. In this setup, both the goal location and environment dynamics change with each task switch. **(a & b)** In both partially-observable and fully-observable scenario, the agent, which optimizes three losses simultaneously (SF + Q-TD + Reward), experiences learning instabilities due to the higher complexity involved in managing all constraints.

### K.3 Continual RL results for Four Rooms environment

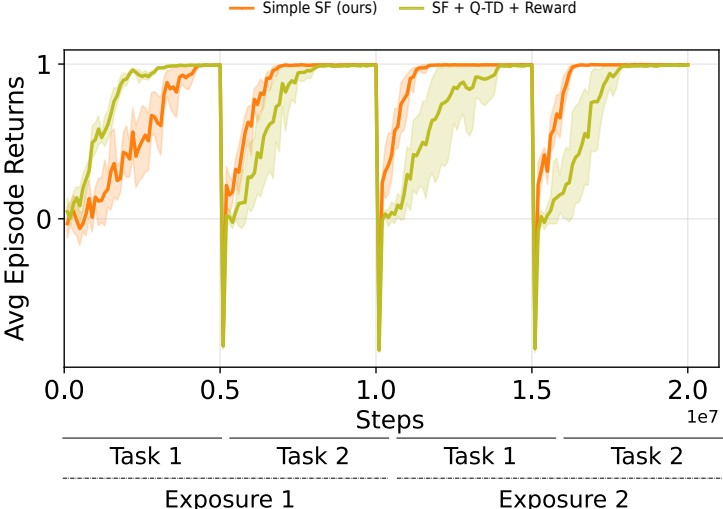

Figure 26: Evaluation in a Continual Reinforcement Learning setting across 5 random seeds, **with replay buffer resets at each task transition** in the 3D Four Rooms environment. Simultaneous optimization of three losses (SF + Q-TD + Reward) slows the learning process, as the agent requires more time to learn an effective policy.

### K.4 Continual RL results for Slippery Four Rooms environment

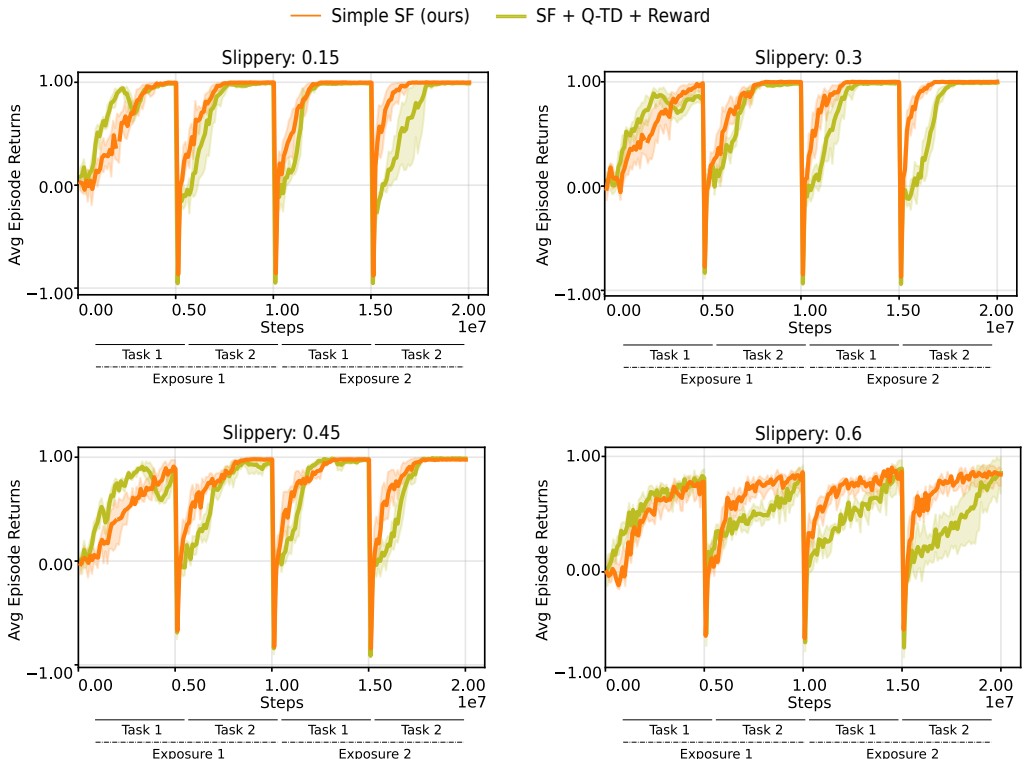

Figure 27: Evaluation in a Continual Reinforcement Learning setting across 5 random seeds, **with replay buffer resets at each task transition** in the 3D Four Rooms environment. This environment features slippery conditions in the top-right and bottom-left rooms for both tasks, Task 1 and Task 2. Both tasks have differing reward structures: In Task 1, rewards are set at +1 for the green box and -1 for the yellow box; in Task 2, this reward scheme is reversed (green box: -1, yellow box: +1). The diagram illustrates the layout of the environment can be found in Figure 9. As observed, simultaneous optimization of three losses (SF + Q-TD + Reward) significantly impedes the agent's ability to learn effectively in a stochastic environment.

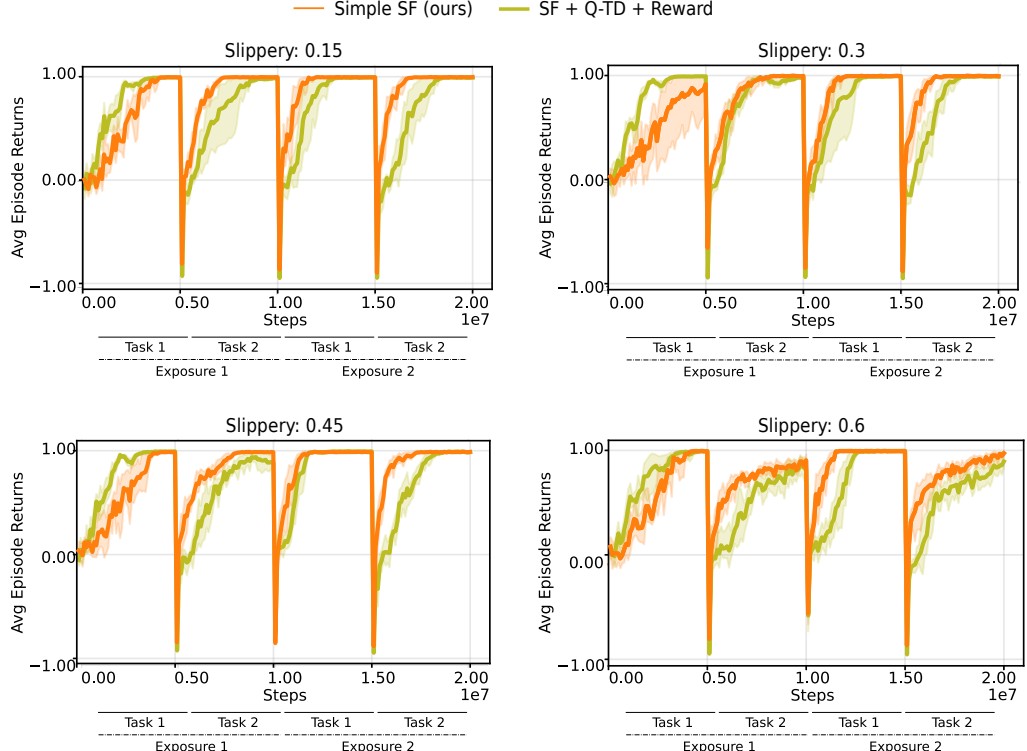

Figure 28: Evaluation in a Continual Reinforcement Learning setting across 5 random seeds, **with replay buffer resets at each task transition** in the 3D Four Rooms environment. Task 1 adheres to the canonical Four Rooms environment dynamics, while Task 2 employs the slippery variant, where chosen actions are altered based on the slippery probability to simulate environmental changes. Throughout both tasks, reward associations remain consistent: +1 for the green box and -1 for the yellow box. The layout of this environment is depicted in Figure 9. Once again, simultaneous optimization of three losses (SF + Q-TD + Reward) significantly impedes the agent's ability to learn effectively.

## L   Implementation Details

For our experimental setup, we utilized Python 3 [Van Rossum and Drake, 2009] as the primary programming language. The agent creation and computational components were developed using Jax [Bradbury et al., 2018, Godwin* et al., 2020], while Haiku [Hennigan et al., 2020] was employed for implementing the neural network components. For data visualization, we used Matplotlib [Hunter, 2007] and Seaborn [Waskom, 2021] to generate line plots. Additionally, we utilized Plotly[8] for creating the violin plots and heat maps used in our correlation analysis, as well as the 2D visualizations of the SFs and DQN Representations. We utilized Scikit-learn [Pedregosa et al., 2011] in our correlation analysis studies as well as the open-source Uniform Manifold Approximation and Projection (UMAP) tool [McInnes et al., 2018] to generate the 2D embeddings of the SFs. The configuration and management of our experiments were facilitated by Hydra [Yadan, 2019] and Weights & Biases [Biewald, 2020]. All experiments, particularly those in the continual learning setting, were conducted using Nvidia V100 GPUs and completed within a maximum of one day. The code used in the study will be released in the near future, following an internal review process.

---

[8]Plotly Technologies Inc. Collaborative data science. Montréal, QC, 2015. https://plot.ly.

# M Visualizations of Successor Features

Given that Successor Features (SFs) are action-dependent, and considering the space constraints in the main paper, our visualizations here are more comprehensive. In the main paper, we primarily showcased visualizations for the *forward* action due to these limitations. However, in this section, we expand our focus to include visual representations for a variety of actions, providing a more holistic view of the SFs' behavior and their influence across different action scenarios. This expanded visualization not only enhances our understanding of the SFs' multidimensional nature but also offers deeper insights into the agent's decision-making process and its interaction with the environment.

## M.1 Center-wall Environment (Fully-observable)

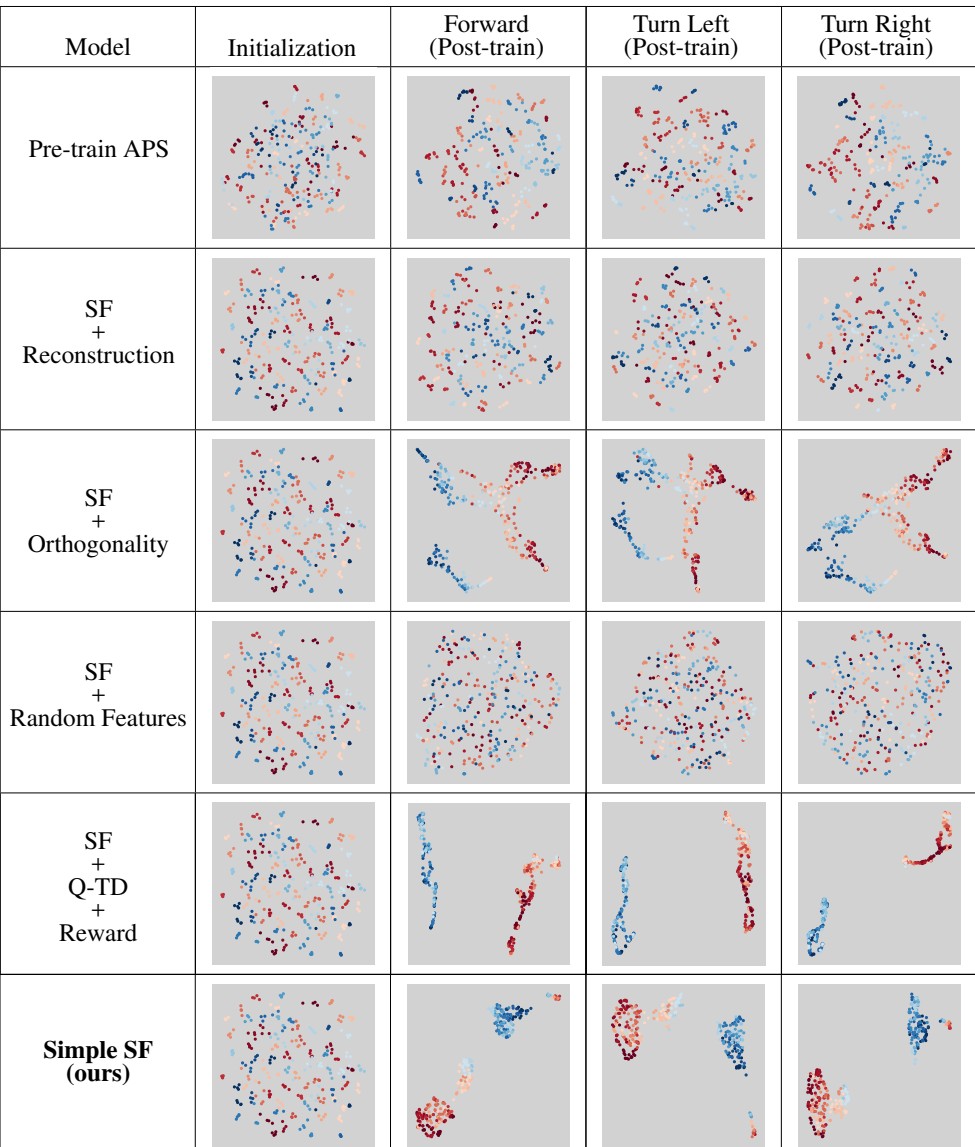

Figure 29: 2D Geospatial Color-Mapped Visualizations of initial and action-based Successor Features in the *Fully-Observable* Center-Wall Environment. This figure displays the successor features of various RL agents, each panel representing a different agent and action.The first column illustrates the initial state of successor features before training, using geospatial color mapping for clear visualization. Subsequent columns correspond to successor features developed for specific actions: Forward, Turn Left, and Turn Right, also visualized using geospatial color mapping. In this scenario, only the agent learning SFs with orthogonality constraints as well as our agent (Simple SF) learned well-clustered representations after training. It's crucial to recognize, however, that while clustered representations may suggest effective learning, they do not automatically equate to successful policy development. These visualizations highlight the varied encoding strategies of agents in response to full observability and different actions.

## M.2 Center-wall Environment (Partially-observable)

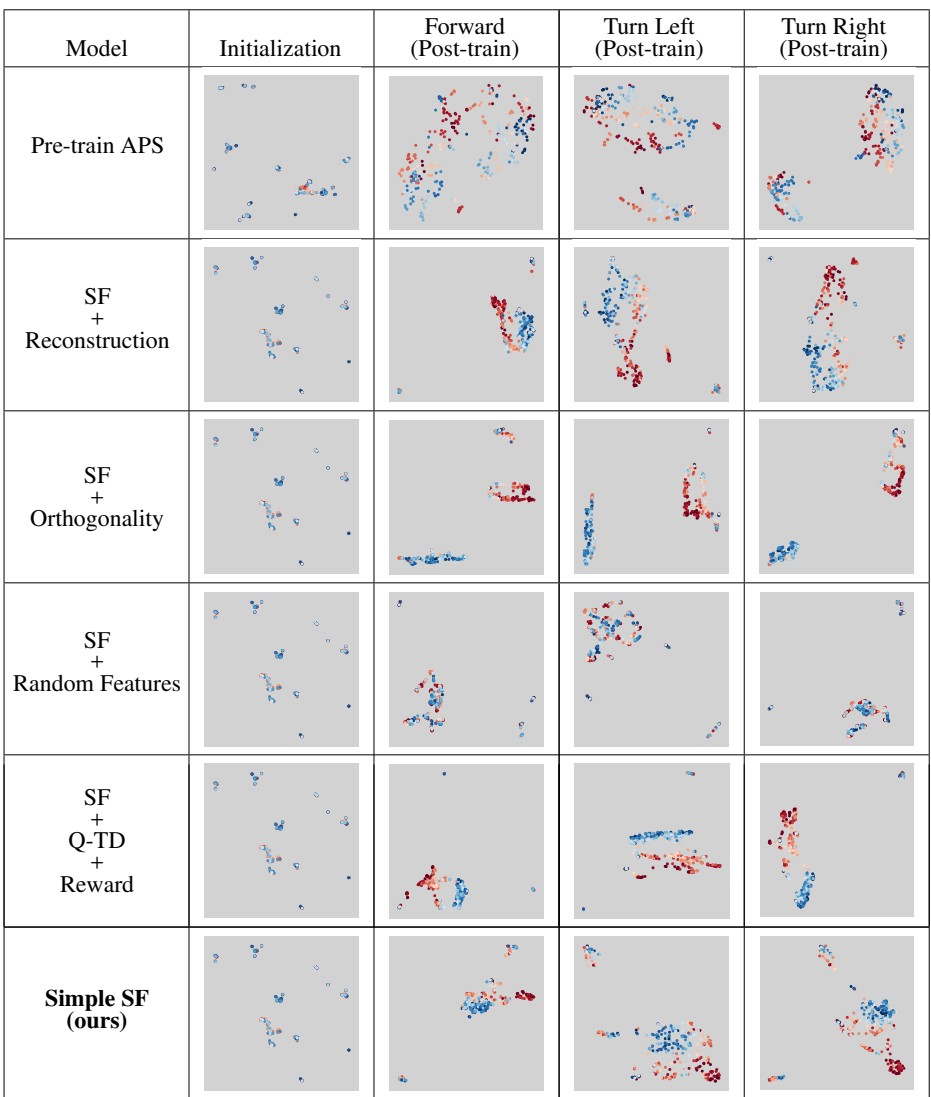

Figure 30: 2D Geospatial Color-Mapped Visualizations of initial and action-based Successor Features in the *Partially-Observable* Center-Wall Environment. This figure displays the successor features of various RL agents, each panel representing a different agent and action. The first column illustrates the initial state of successor features before training, using geospatial color mapping for clear visualization. Subsequent columns correspond to successor features developed for specific actions: Forward, Turn Left, and Turn Right, also visualized using geospatial color mapping. Some agents demonstrate well-clustered representations after training, which typically correlates with improved performance compared to agents with more dispersed or noisy features. It's crucial to recognize, however, that while clustered, color-mapped representations may suggest effective learning, they do not automatically equate to successful policy development. These visualizations highlight the varied encoding strategies of agents in response to partial observability and different actions.

## M.3 Four Rooms Environment

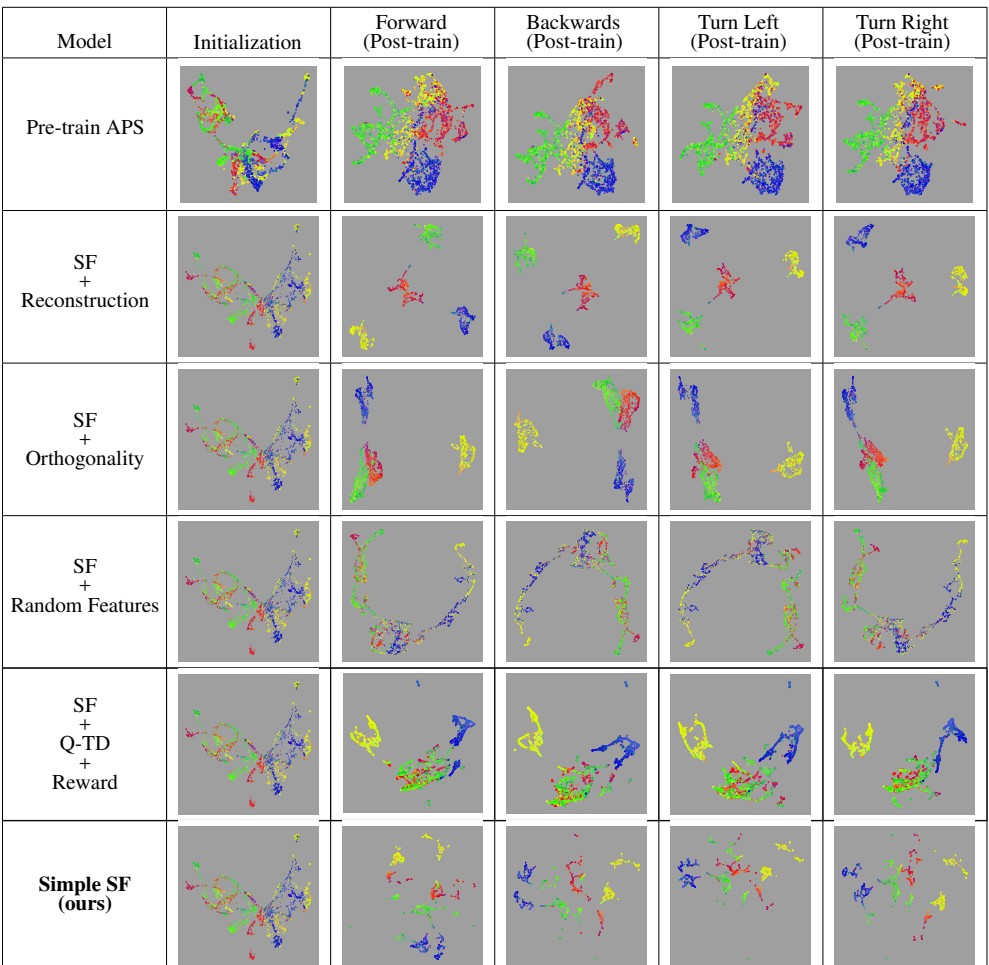

Figure 31: 2D Geospatial Color-Mapped Visualizations of initial and action-based Successor Features in the 3D Four Rooms Environment. Agents operate with solely egocentric observations. Each panel represents the successor features of a different RL agent and action. The first column, using red, green, blue, and yellow to distinguish the four rooms, shows the initial state of successor features pre-training. Subsequent columns depict features for specific actions: Move Forward, Move Backwards, Turn Left, and Turn Right. Except for SF learned using unlearnable random basis features, most agents exhibit well-clustered representations post-training. However, it's important to note that such clustered, color-mapped representations, while indicative of effective learning, do not necessarily translate into successful policy development.

# N  Correlation Analysis

Considering that the SRs are not normally distributed [Stachenfeld et al., 2017], we conduct our correlation analysis in the Grid world environments (Figure 9a and b) using the Spearman's rank correlation. The SRs were analytically computed using the transition matrix $T$ where $T(s' \mid s, a)$ denotes the probability of transitioning from state $s$ to state $s'$ given an action $a \sim \pi(\cdot \mid s)$:

$$\text{SR} = (I - \gamma T)^{-1} \tag{21}$$

where $0 \leq \gamma < 1$ is the discount factor and $I$ is the identity matrix. The same policy $\pi$ was used to generate the transition matrix $T$ and to adjust the final correlations. These adjustments account for less frequently chosen actions and for positions and head directions less likely to be encountered by the agent, as outlined in the main text. Statistics regarding positions and head directions were collected using policy $\pi$.

In the remaining part of this section, we provide additional detailed violin plots to depict the correlation dynamics in both the Center-wall and Inverted-LWalls environments, covering scenarios that are both partially-observable and fully-observable. These plots are segmented into different stages: before training, after training, and the differences post-training. This segmentation offers a comprehensive view of the agents' learning progression over time. Specifically for the Inverted-LWalls environment, a table is included to provide a summary of mean and standard deviation statistics for these correlations, thus offering a clear quantitative perspective of our findings. Additionally, we present heatmaps that showcase the correlation at each spatial position in the environment for various SF agents. These heatmaps further enrich our analysis by visually representing the spatial distribution of correlation values, highlighting how different agents adapt to the environment.

## N.1 Center-wall Environment (Partially-observable)

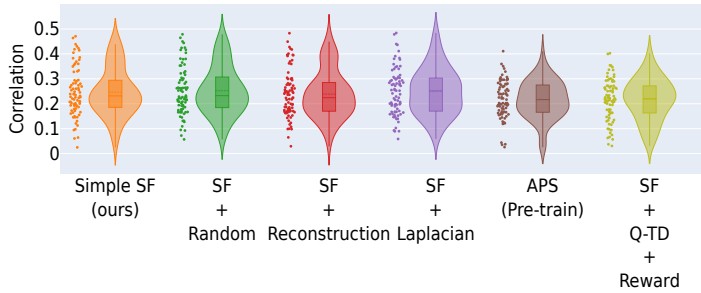

(a) Before Training

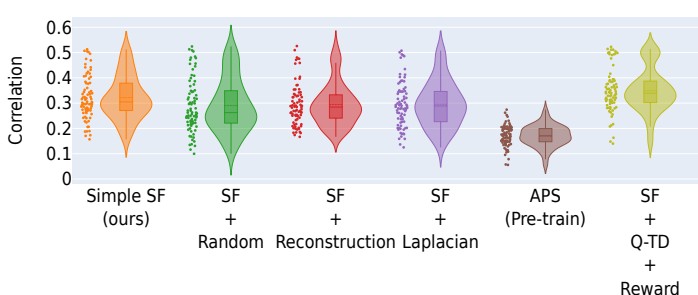

(b) After Training

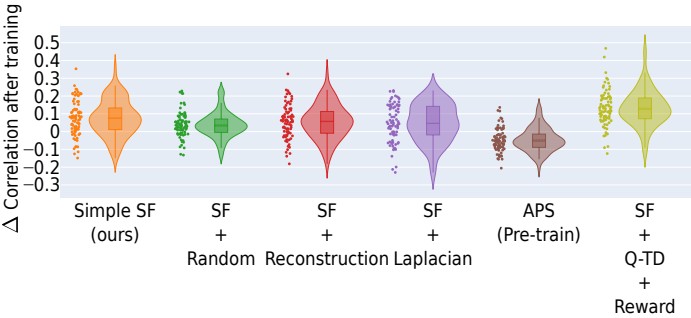

(c) Difference (Before vs After)

Figure 32: Correlation analysis between learned Successor Features and analytically computed Successor Representation for all positions in the Center-Wall Environment under the *Partially-observable* scenario.

## N.2   Center-wall Environment (Fully-observable)

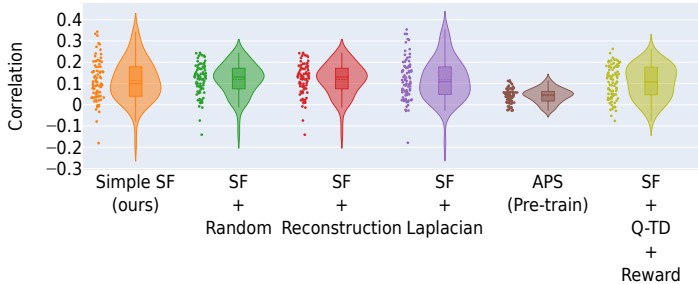

(a) Before Training

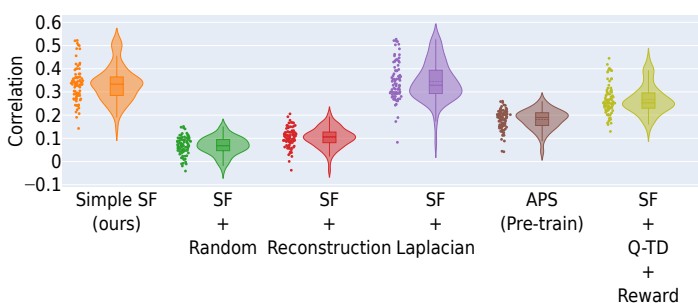

(b) After Training

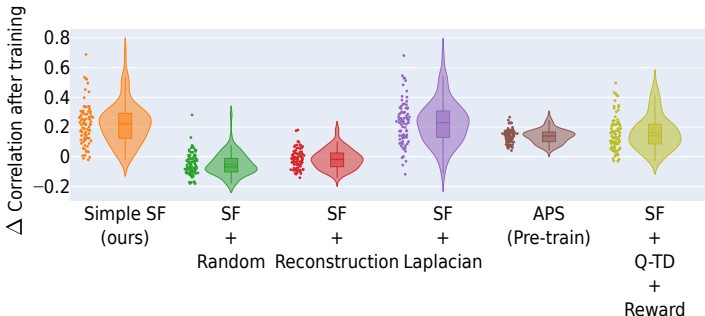

(c) Difference (Before vs After)

Figure 33: Correlation Analysis between Successor Features and Successor Representation for all positions in the Center-Wall Environment (Fully-observable).

## N.3  Inverted-LWalls Environment (Partially-observable)

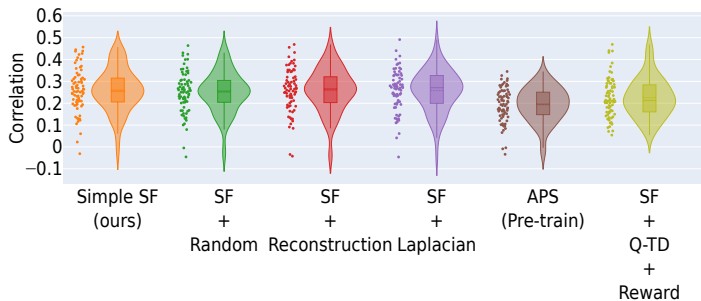

(a) Before Training

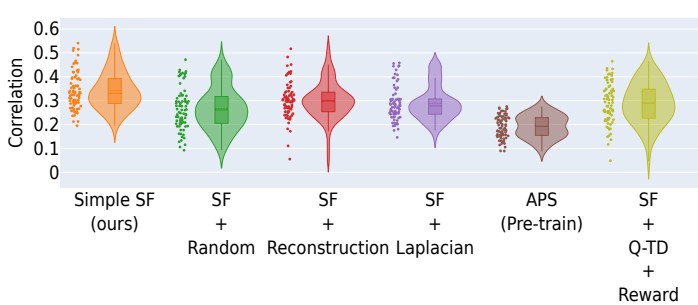

(b) After Training

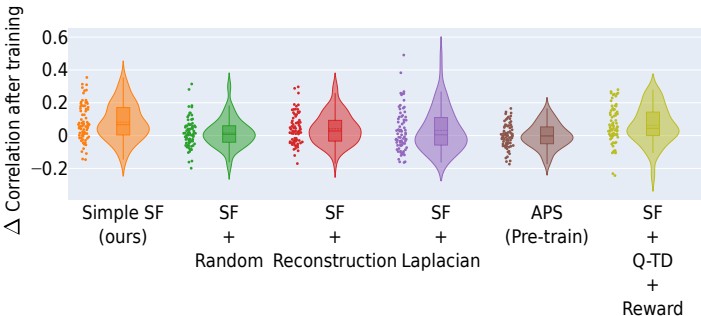

(c) Difference (Before vs After)

Figure 34: Correlation Analysis between Successor Features and Successor Representation for all positions in the Inverted-LWalls-Grid Environment (Partially-observable).

### N.4 Inverted-LWalls Environment (Fully-observable)

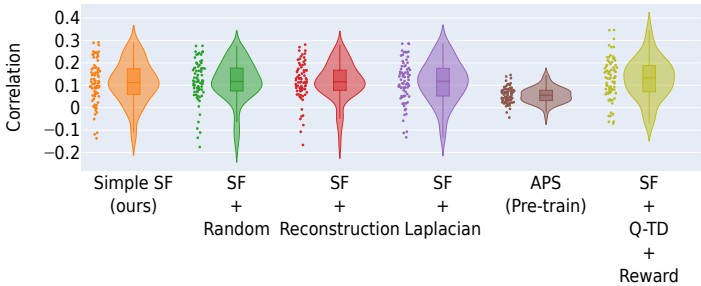

(a) Before Training

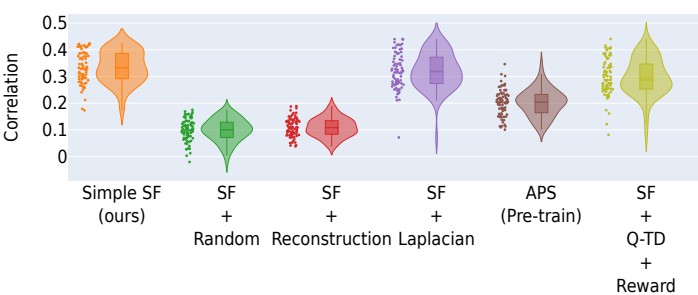

(b) After Training

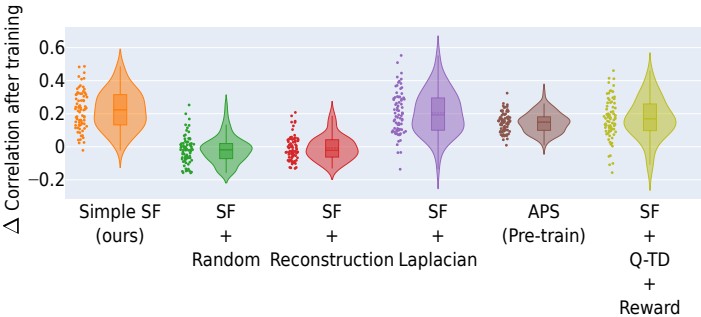

(c) Difference (Before vs After)

Figure 35: Correlation Analysis between Successor Features and Successor Representation for all positions in the Inverted-LWalls-Grid Environment (Fully-observable).

## N.5 Summary Statistics of the Correlation Analysis

Table 6: Correlation Analysis against analytically computed Successor Representation in the Center-Wall Environment with mean and standard deviation of the correlations. The data are categorized into three stages: before training, after training, and the observed differences post-training. The left column: Partially-observable scenarios in which our agent shows the highest correlation and greatest improvement post-training. The right column: Fully-observable scenarios where our agent and the agent with orthogonality constraints on basis features exhibit high correlation and significant post-training improvement.

| | PARTIALLY-OBS. | | | FULLY-OBS. | | |
|---|---|---|---|---|---|---|
| MODEL | BEFORE | AFTER | DIFFERENCE | BEFORE | AFTER | DIFFERENCE |
| SF + RECONSTRUCTION | 0.23± 0.09 | 0.29± 0.08 | 0.05± 0.09 | 0.11± 0.07 | 0.10± 0.04 | -0.01± 0.06 |
| SF + RANDOM FEATURES | 0.25± 0.10 | 0.29± 0.10 | 0.03± 0.07 | 0.11± 0.07 | 0.06± 0.04 | -0.05± 0.07 |
| SF + ORTHOGONALITY | 0.24± 0.10 | 0.29± 0.09 | 0.04± 0.10 | 0.11± 0.1 | **0.34± 0.08** | **0.22± 0.14** |
| PRE-TRAIN (APS) | 0.21±0.07 | 0.17± 0.04 | -0.04± 0.06 | 0.04± 0.03 | 0.18± 0.04 | 0.13± 0.04 |
| SF + Q-TD + REWARD | 0.21±0.08 | **0.34± 0.08** | **0.13± 0.10** | 0.10± 0.07 | 0.26± 0.06 | 0.16± 0.11 |
| SIMPLE SF(OURS) | 0.24±0.09 | 0.32± 0.08 | 0.07± 0.10 | 0.11± 0.09 | 0.33± 0.07 | 0.22± 0.13 |

Table 7: Correlation Analysis against analytically computed Successor Representation in the Inverted-LWalls-Grid Environment with mean and standard deviation of the correlations. The data are categorized into three stages: before training, after training, and the observed differences post-training. Notably, our agent demonstrated the largest improvement in correlation as well as the highest resulting correlation after the training period in both Partially-observable scenarios (left) and Fully-observable scenarios (right).

| | PARTIALLY-OBS. | | | FULLY-OBS. | | |
|---|---|---|---|---|---|---|
| MODEL | BEFORE | AFTER | DIFFERENCE | BEFORE | AFTER | DIFFERENCE |
| SF + RECONSTRUCTION | 0.26 ±0.1 | 0.30 ± 0.07 | 0.04 ± 0.09 | 0.11 ± 0.08 | 0.10 ± 0.03 | -0.01 ± 0.07 |
| SF + ORTHOGONALITY | 0.25 ± 0.09 | 0.28 ± 0.07 | 0.03 ± 0.12 | 0.11 ± 0.09 | 0.32 ± 0.06 | 0.20 ± 0.13 |
| SF + RANDOM FEATURES | 0.25 ± 0.1 | 0.26 ± 0.1 | 0.01 ± 0.08 | 0.11 ± 0.09 | 0.09 ± 0.04 | -0.01 ± 0.08 |
| PRE-TRAIN (APS) | 0.19 ± 0.07 | 0.19 ± 0.04 | 0 ± 0.07 | 0.05 ± 0.03 | 0.20 ± 0.04 | 0.14 ± 0.06 |
| SF + Q-TD + REWARD | 0.22±0.09 | 0.29± 0.08 | 0.06± 0.11 | 0.12± 0.09 | 0.29± 0.07 | 0.17± 0.12 |
| SIMPLE SF(OURS) | 0.26 ± 0.1 | **0.34 ± 0.07** | **0.08 ± 0.11** | 0.11 ± 0.09 | **0.33 ± 0.06** | **0.22 ± 0.11** |

## N.6 Heatmap Visualization of SF Correlation in the Center-Wall Environment (Partially-Observable)

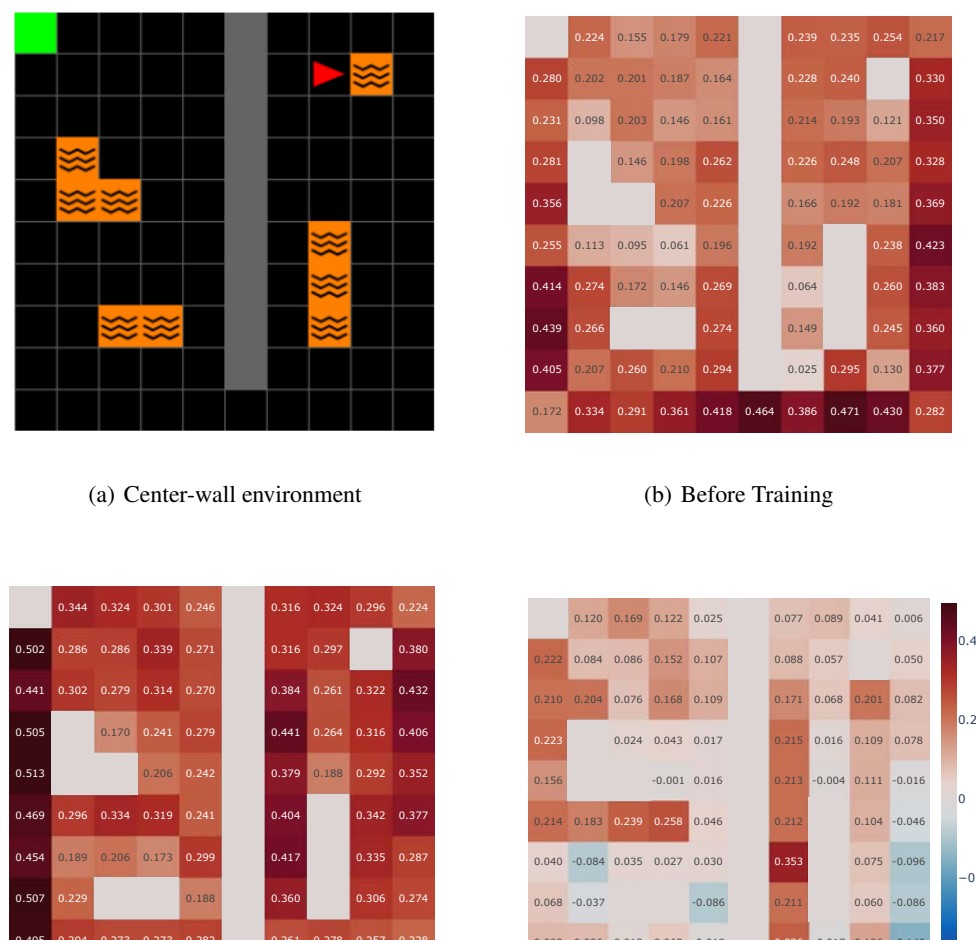

(a) Center-wall environment

(b) Before Training

(c) After Training

(d) Difference (Before vs After)

Figure 36: Correlation Analysis between Simple Successor Features (our model) and Successor Representation in the Center-Wall Environment (Partially-observable).

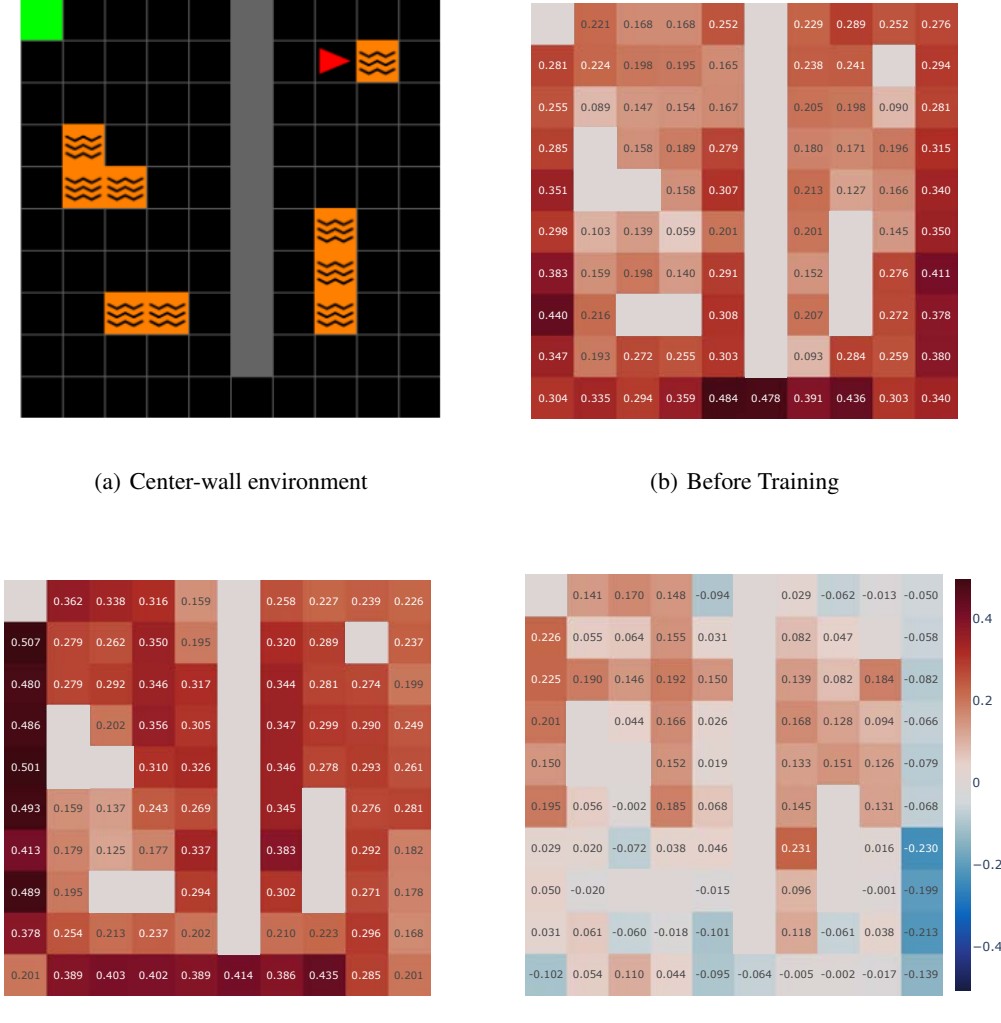

(a) Center-wall environment

(b) Before Training

(c) After Training

(d) Difference (Before vs After)

Figure 37: Correlation Analysis between Successor Features with orthogonality constraints (SF + Orthogonality) and Successor Representation in the Center-Wall Environment (Partially-observable)

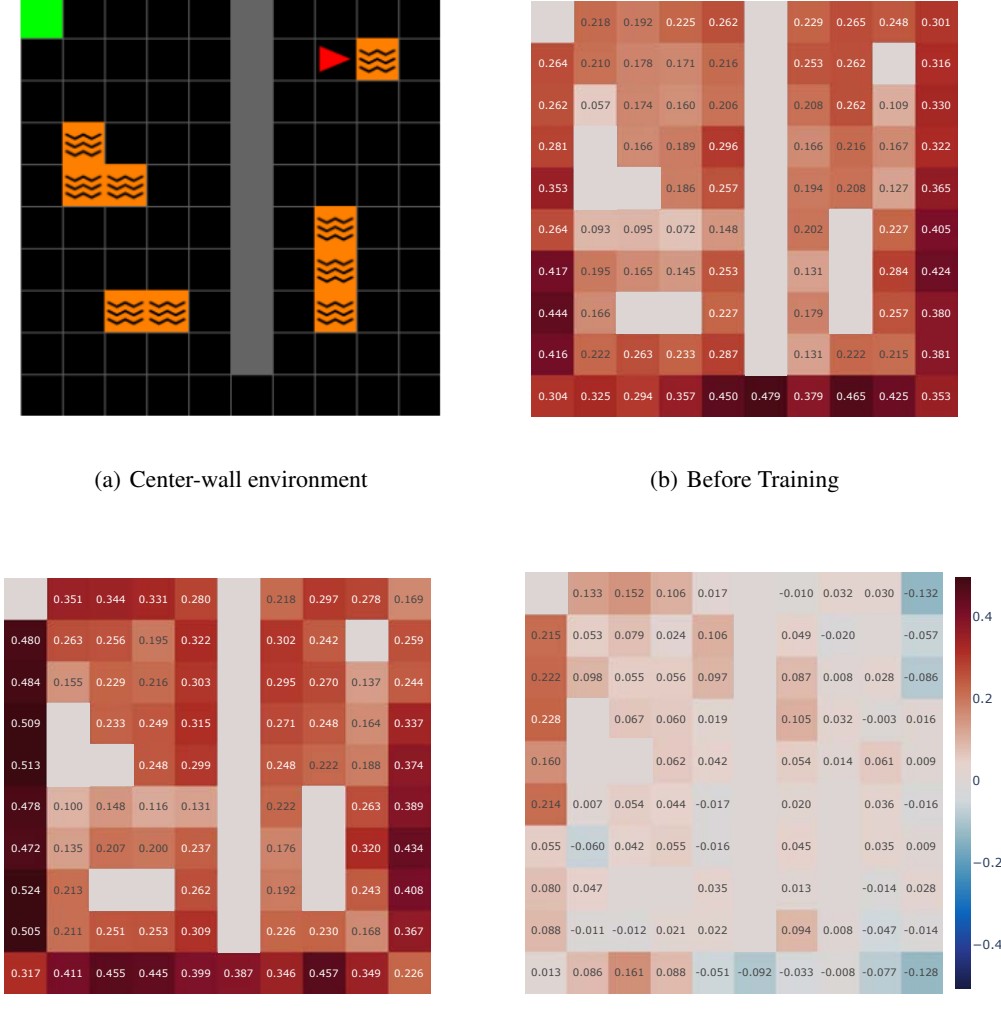

(a) Center-wall environment

(b) Before Training

(c) After Training

(d) Difference (Before vs After)

Figure 38: Correlation Analysis between Successor Features with Random un-learnable constraints (SF + Random) and Successor Representation in the Center-Wall Environment (Partially-observable)

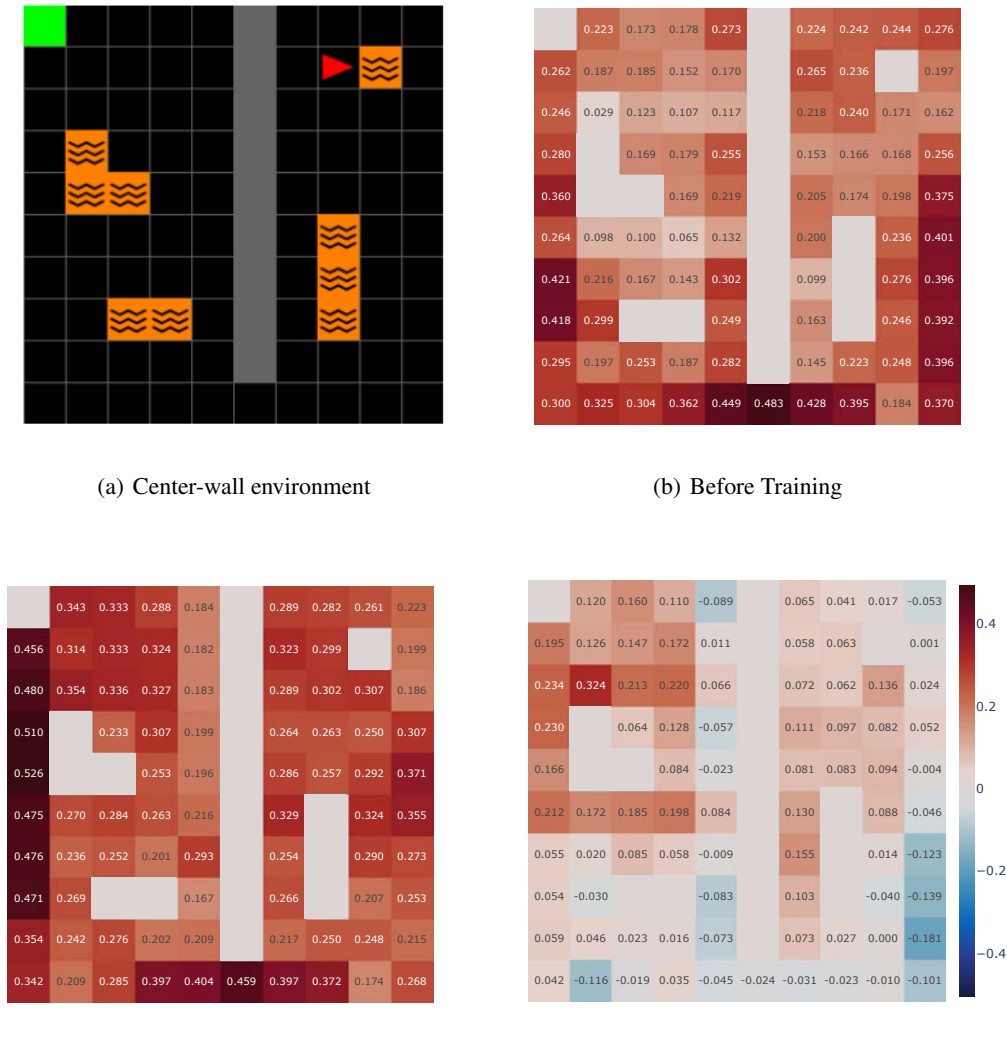

(a) Center-wall environment

(b) Before Training

(c) After Training

(d) Difference (Before vs After)

Figure 39: Correlation Analysis between Successor Features with reconstruction constraints (SF + Reconstruction) and Successor Representation in the Center-Wall Environment (Partially-observable)

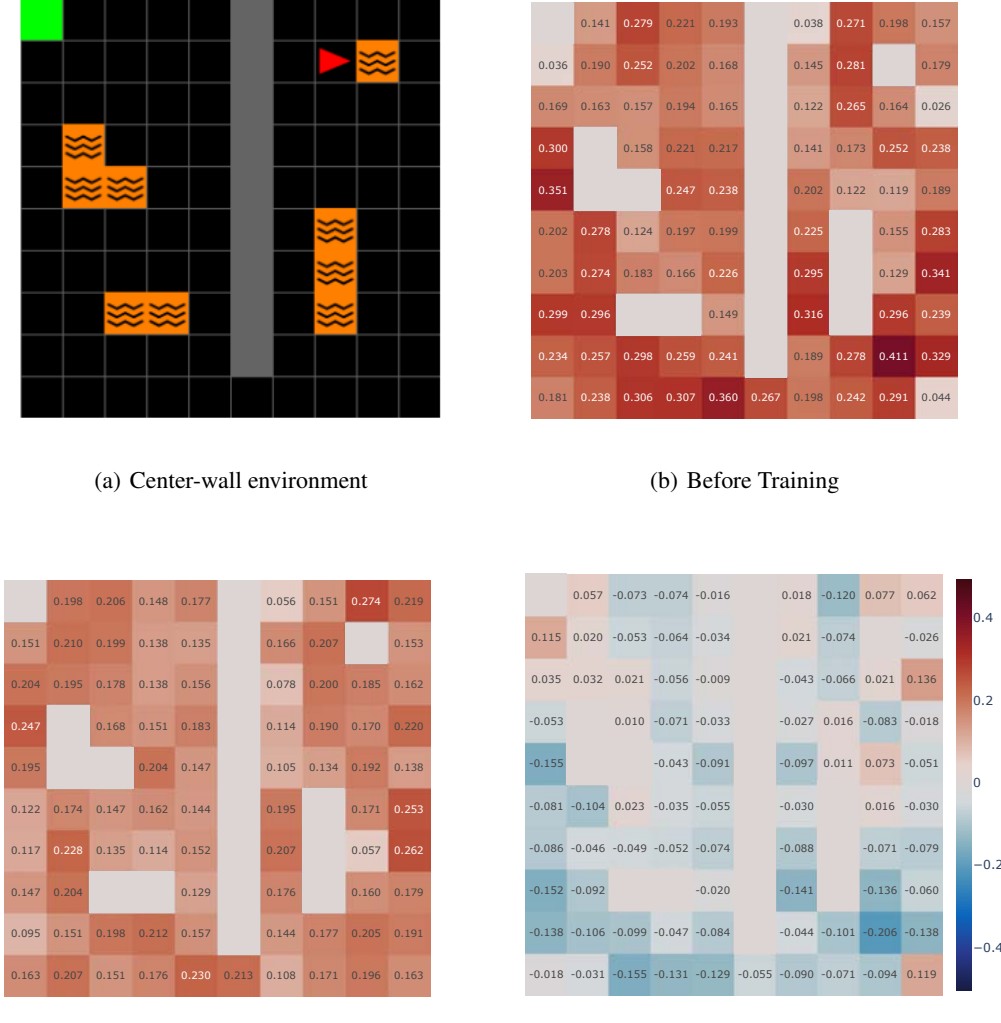

(a) Center-wall environment

(b) Before Training

(c) After Training

(d) Difference (Before vs After)

Figure 40: Correlation Analysis between APS Pre-train Successor Features [Liu and Abbeel, 2021] and Successor Representation in the Center-Wall Environment (Partially-observable)

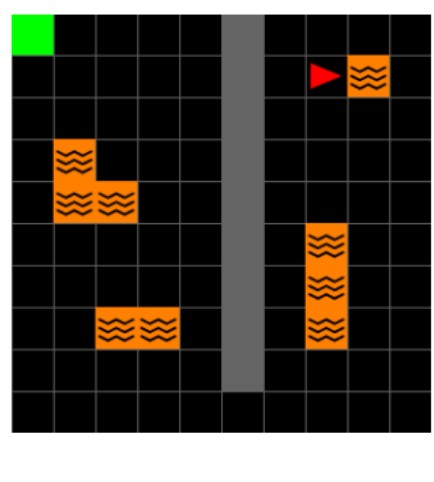

(a) Center-wall environment

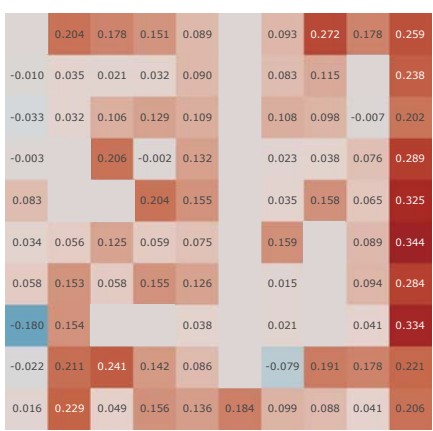

(b) Before Training

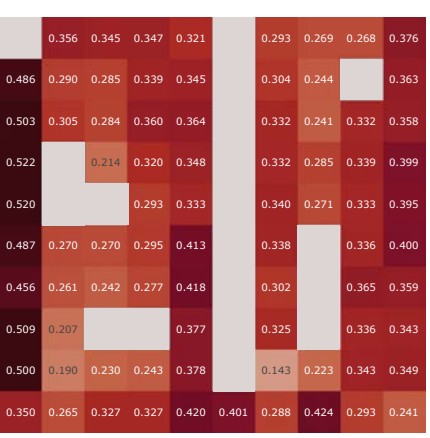

(c) After Training

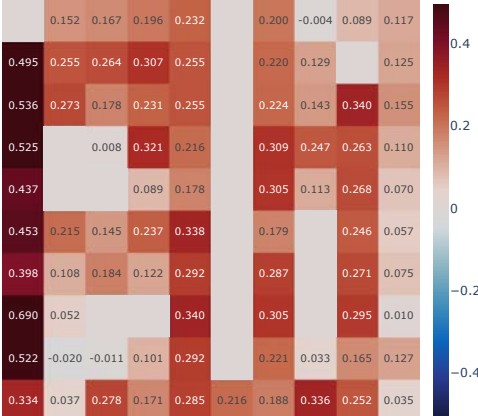

(d) Difference (Before vs After)

Figure 41: Correlation Analysis between Simple Successor Features (our model) and Successor Representation in the Center-Wall Environment (Fully-observable)

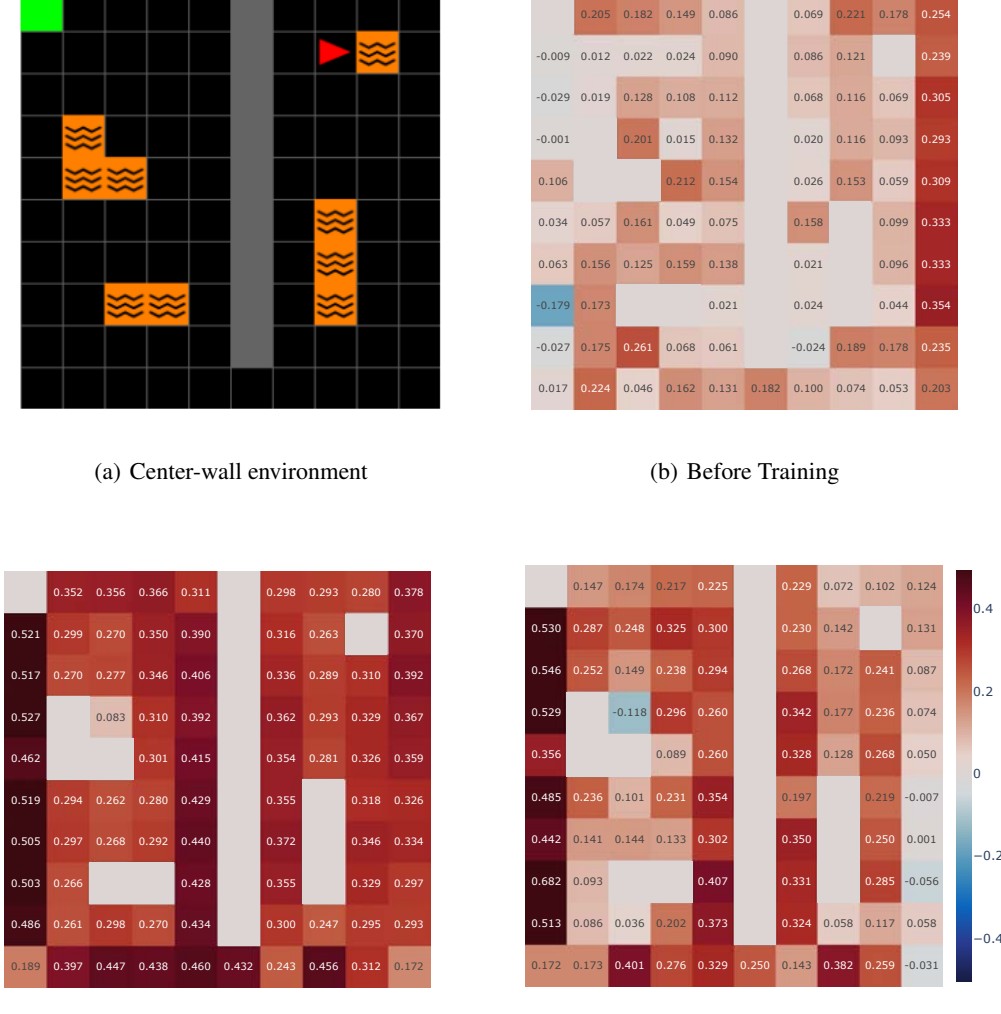

(a) Center-wall environment

(b) Before Training

(c) After Training

(d) Difference (Before vs After)

Figure 42: Correlation Analysis between Successor Features with orthogonality constraints (SF + Orthogonality) and Successor Representation in the Center-Wall Environment (Fully-observable)

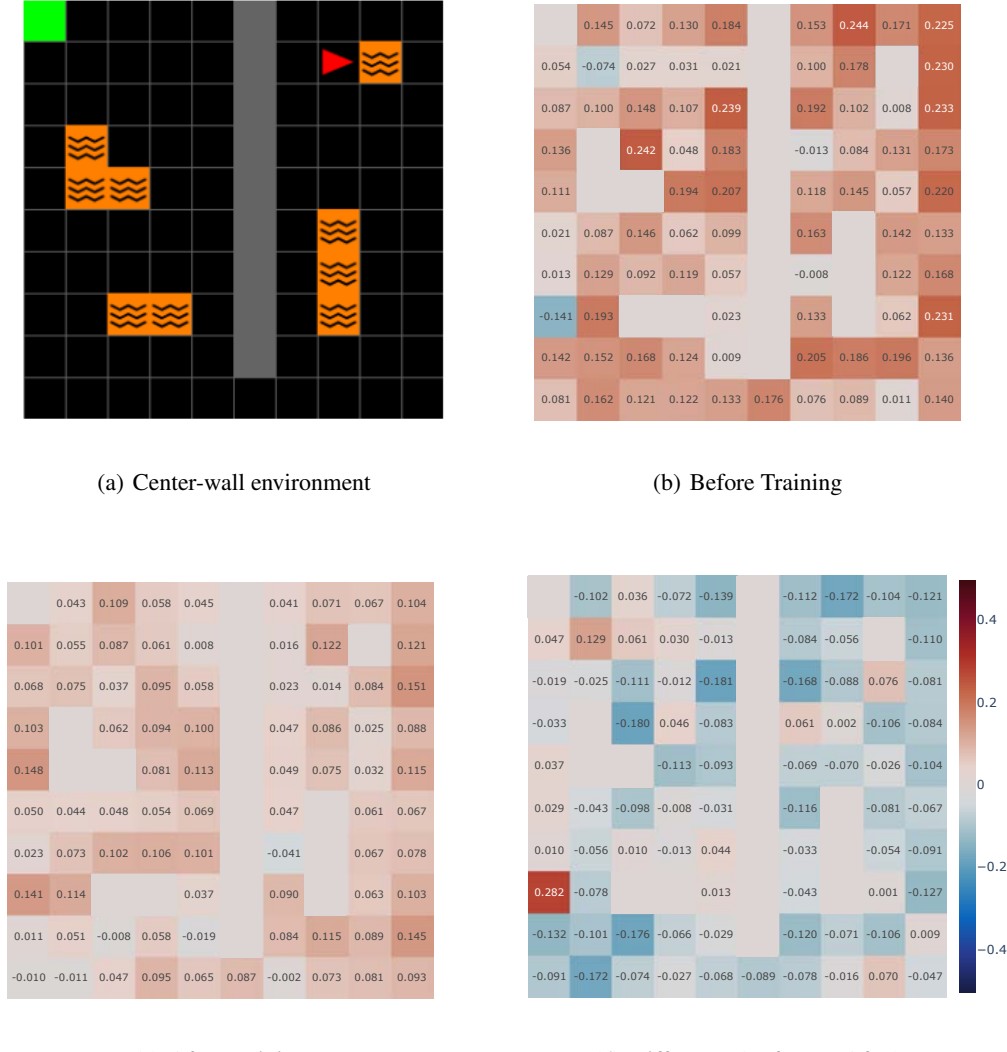

(a) Center-wall environment

(b) Before Training

(c) After Training

(d) Difference (Before vs After)

Figure 43: Correlation Analysis between Successor Features with Random un-learnable constraints (SF + Random) and Successor Representation in the Center-Wall Environment (Fully-observable)

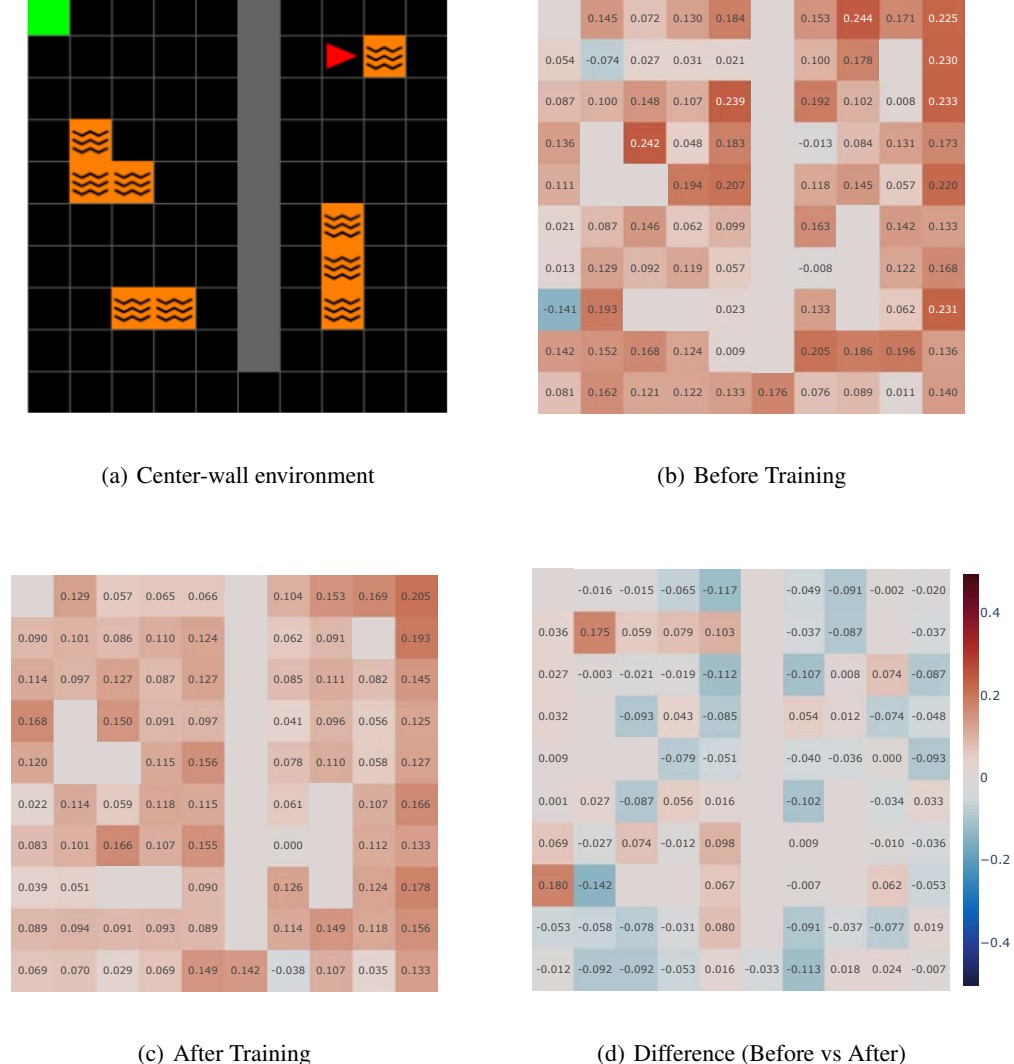

(a) Center-wall environment

(b) Before Training

(c) After Training

(d) Difference (Before vs After)

Figure 44: Correlation Analysis between Successor Features with reconstruction constraints (SF + Reconstruction) and Successor Representation in the Center-Wall Environment (Fully-observable)

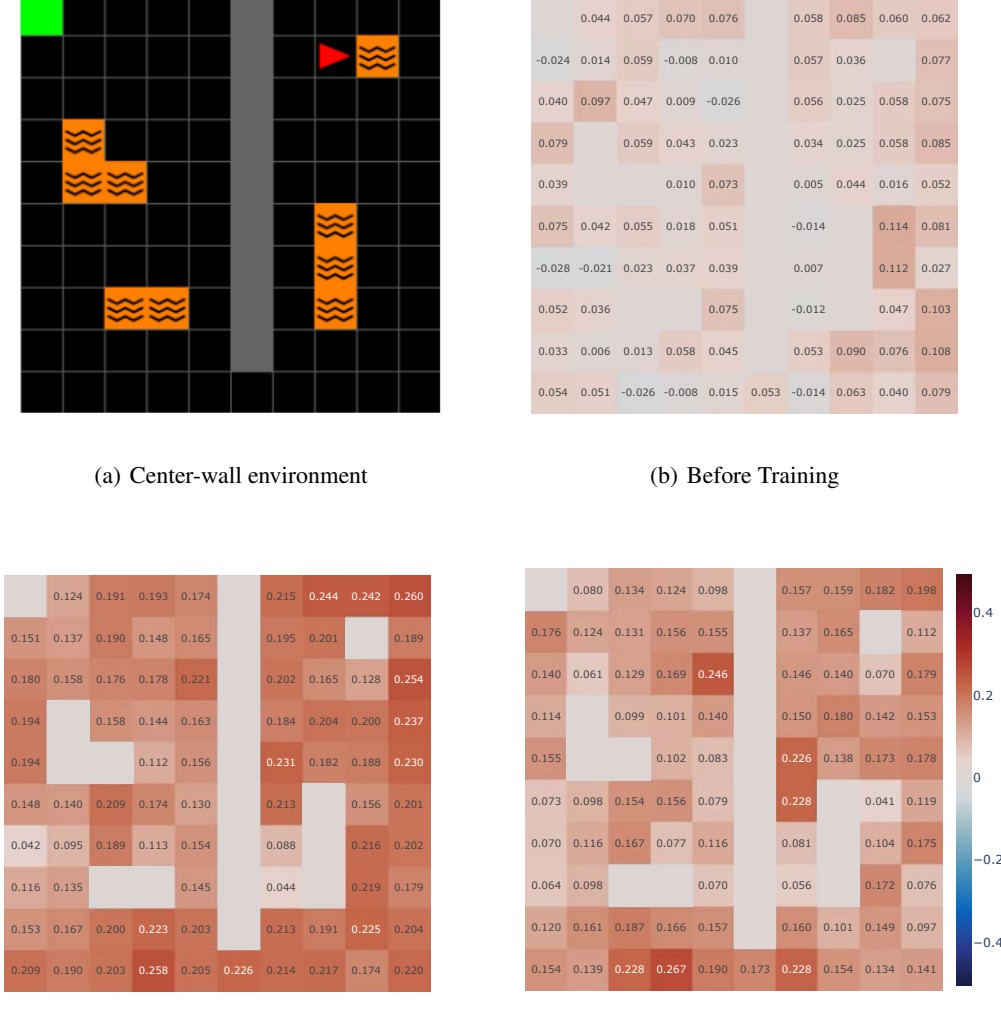

(a) Center-wall environment

(b) Before Training

(c) After Training

(d) Difference (Before vs After)

Figure 45: Correlation Analysis between APS Pre-train Successor Features [Liu and Abbeel, 2021] and Successor Representation in the Center-Wall Environment (Fully-observable)

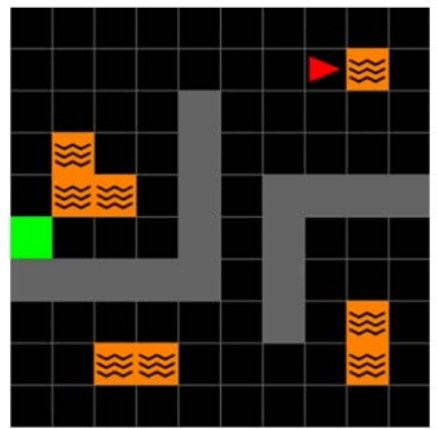

(a) Inverted-LWalls environment

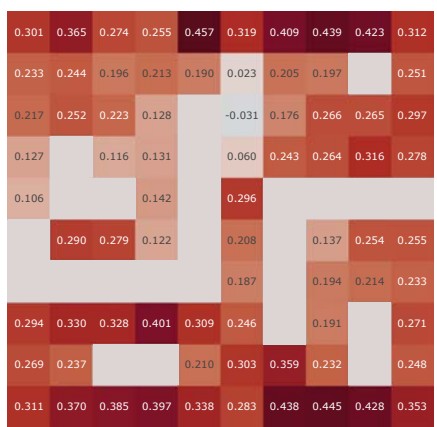

(b) Before Training

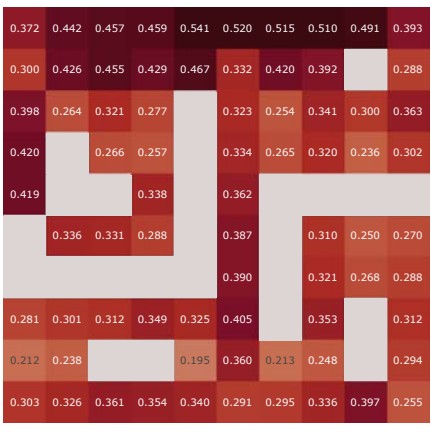

(c) After Training

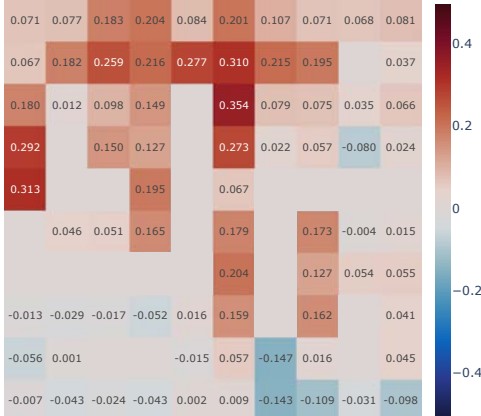

(d) Difference (Before vs After)

Figure 46: Correlation Analysis between Simple Successor Features (our model) and Successor Representation in the Inverted-LWalls-Grid Environment (Partially-observable).

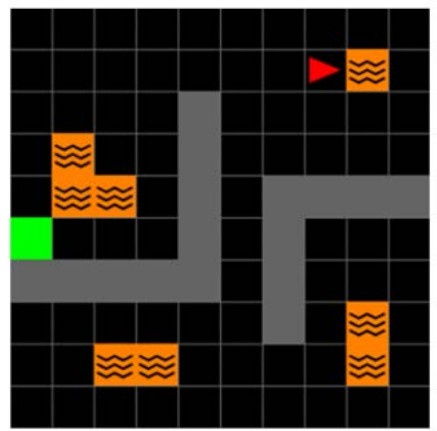

(a) Inverted-LWalls environment

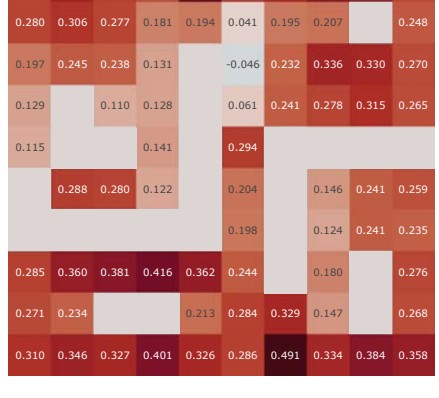

(b) Before Training

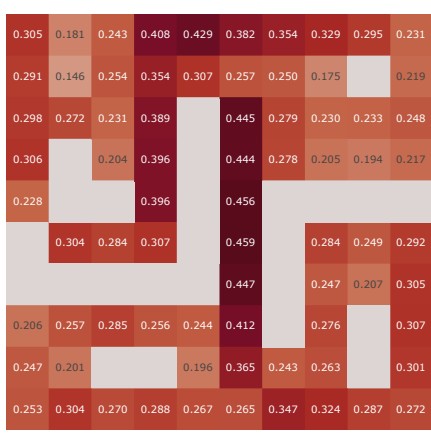

(c) After Training

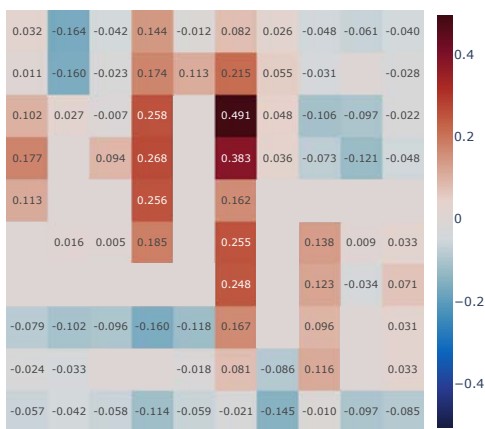

(d) Difference (Before vs After)

Figure 47: Correlation Analysis between Successor Features with orthogonality constraints (SF + Orthogonality) and Successor Representation in the Inverted-LWalls-Grid Environment (Partially-observable)

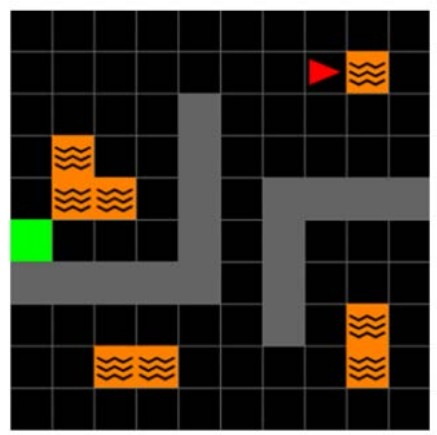

(a) Inverted-LWalls environment

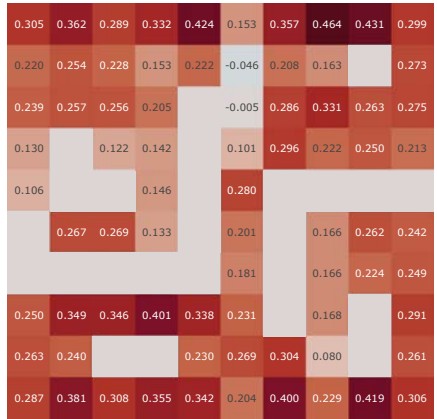

(b) Before Training

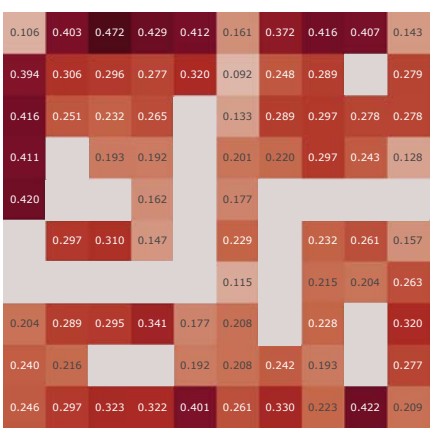

(c) After Training

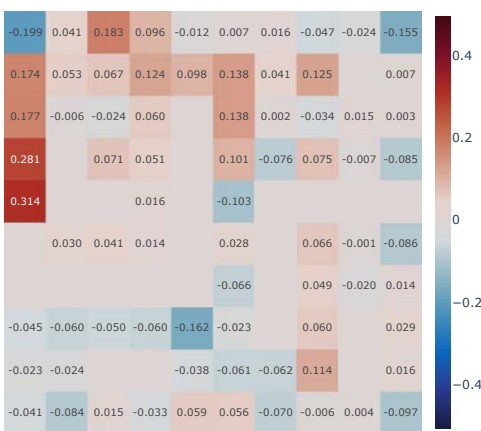

(d) Difference (Before vs After)

Figure 48: Correlation Analysis between Successor Features with Random un-learnable constraints (SF + Random) and Successor Representation in the Inverted-LWalls-Grid Environment (Partially-observable)

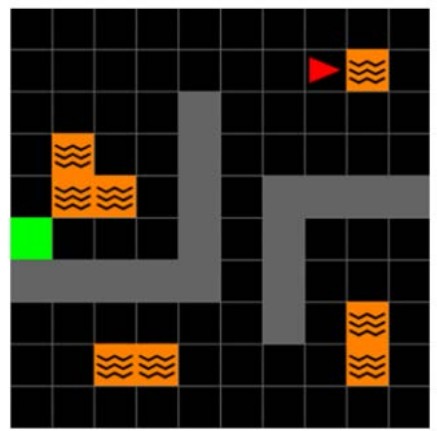

(a) Inverted-LWalls environment

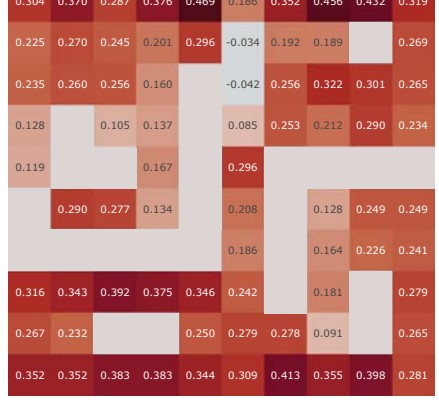

(b) Before Training

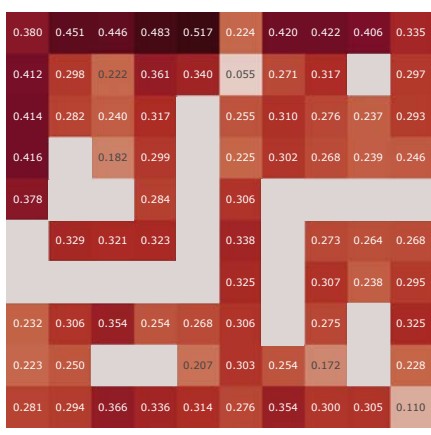

(c) After Training

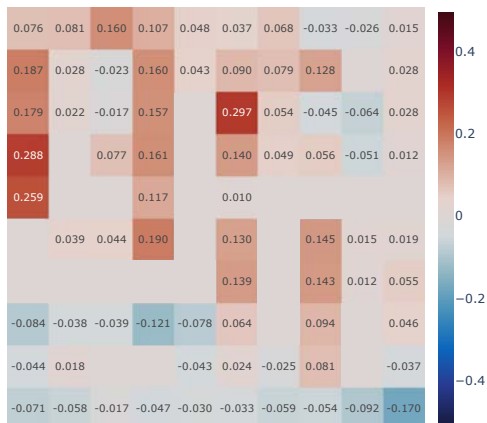

(d) Difference (Before vs After)

Figure 49: Correlation Analysis between Successor Features with reconstruction constraints (SF + Reconstruction) and Successor Representation in the Inverted-LWalls-Grid Environment (Partially-observable)

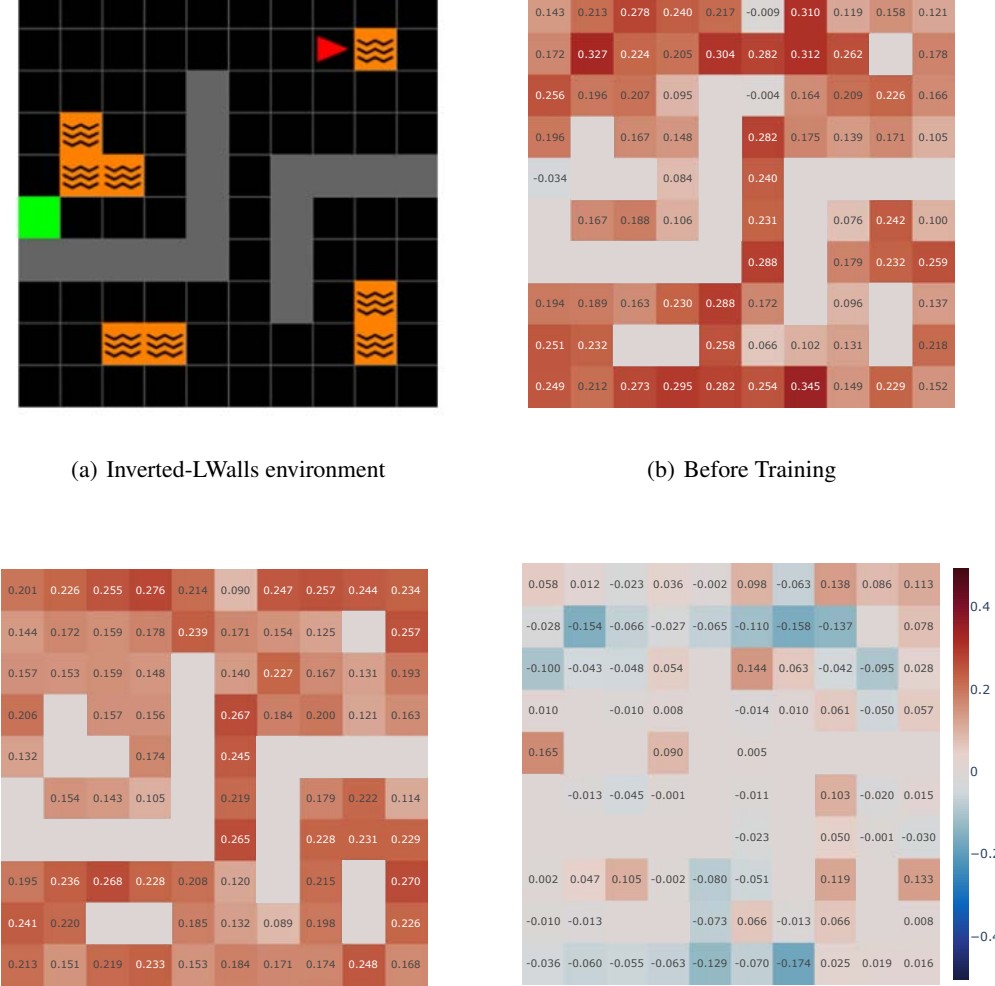

(a) Inverted-LWalls environment

(b) Before Training

(c) After Training

(d) Difference (Before vs After)

Figure 50: Correlation Analysis between APS Pre-train Successor Features [Liu and Abbeel, 2021] and Successor Representation in the Inverted-LWalls-Grid Environment (Partially-observable)

## N.9 Heatmap Visualization of SF Correlation in the Inverted-LWalls Environment (Partially-Observable)

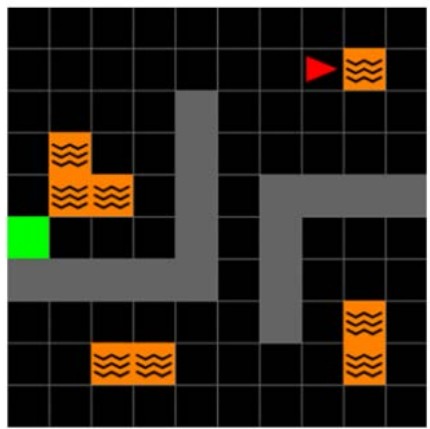

(a) Inverted-LWalls environment

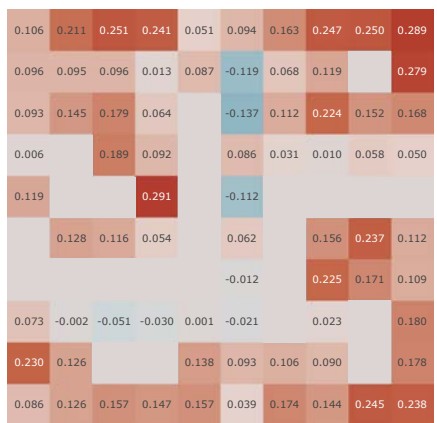

(b) Before Training

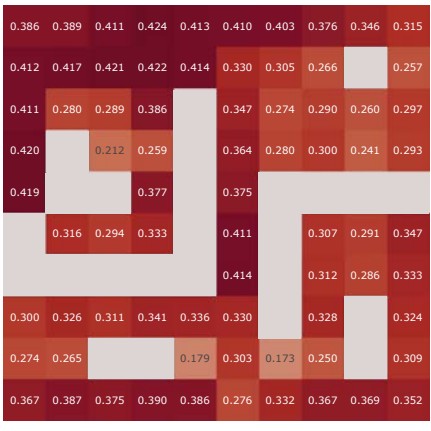

(c) After Training

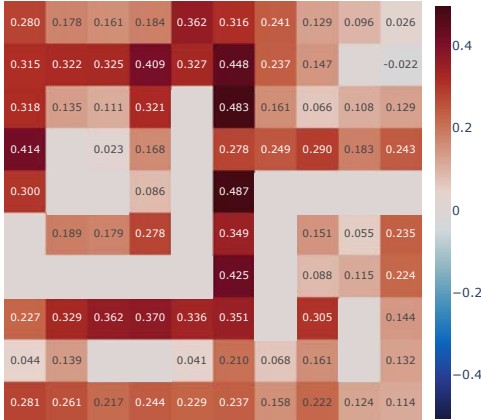

(d) Difference (Before vs After)

Figure 51: Correlation Analysis between Simple Successor Features (our model) and Successor Representation in the Inverted-LWalls-Grid Environment (Fully-observable)

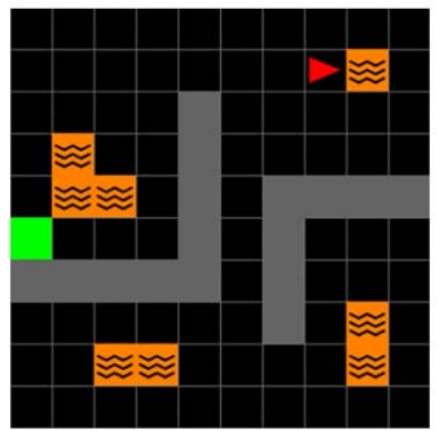

(a) Inverted-LWalls environment

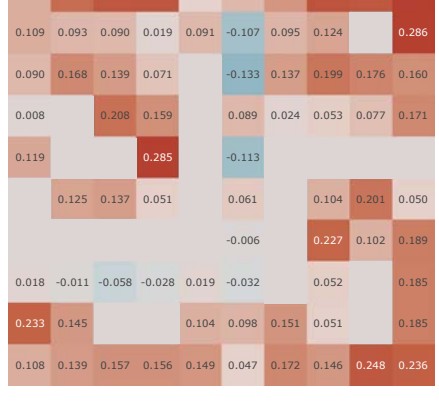
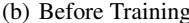

(b) Before Training

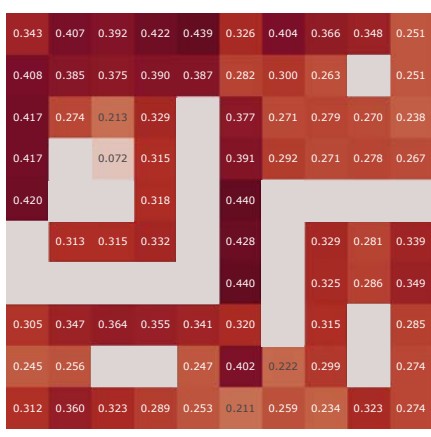

(c) After Training

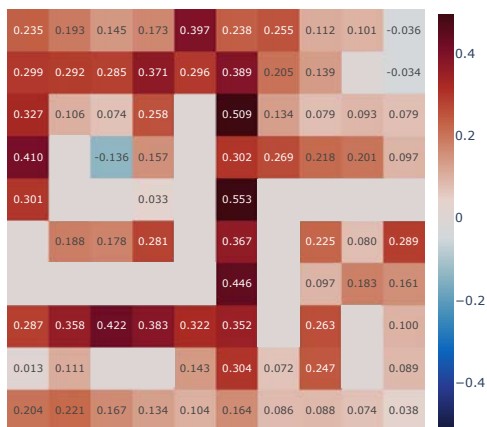

(d) Difference (Before vs After)

Figure 52: Correlation Analysis between Successor Features with orthogonality constraints (SF + Orthogonality) and Successor Representation in the Inverted-LWalls-Grid Environment (Fully-observable)

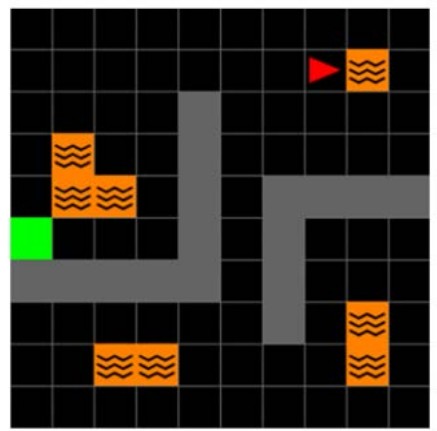

(a) Inverted-LWalls environment

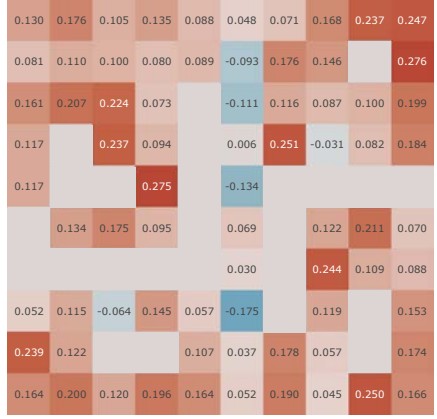

(b) Before Training

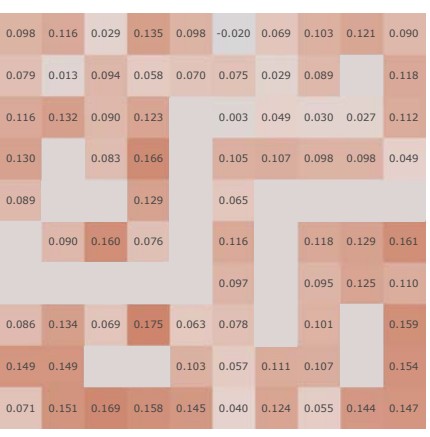

(c) After Training

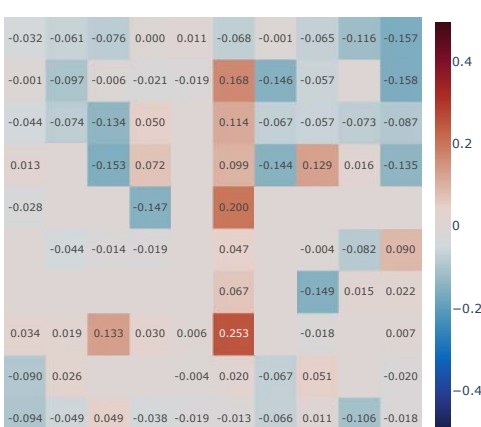

(d) Difference (Before vs After)

Figure 53: Correlation Analysis between Successor Features with Random un-learnable constraints (SF + Random) and Successor Representation in the Inverted-LWalls-Grid Environment (Fully-observable)

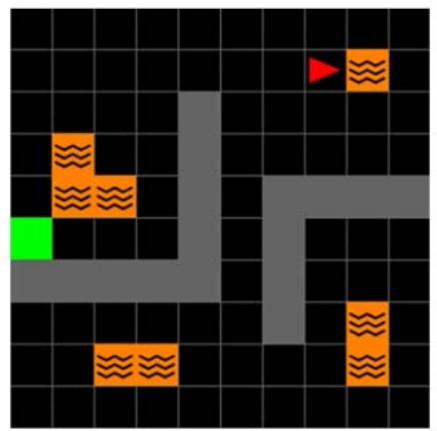

(a) Inverted-LWalls environment

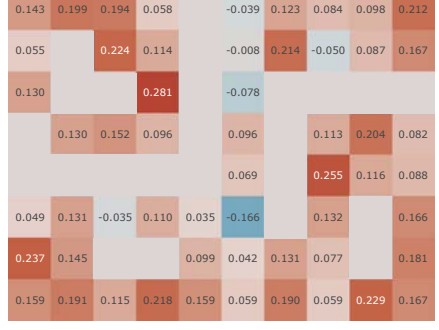

(b) Before Training

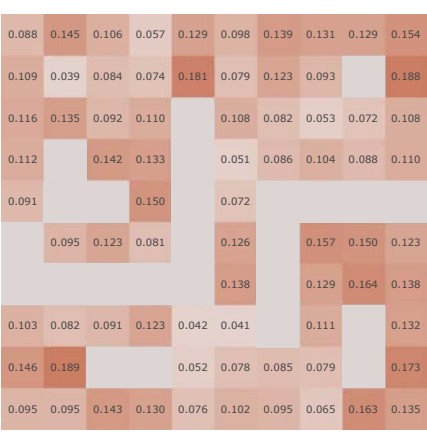

(c) After Training

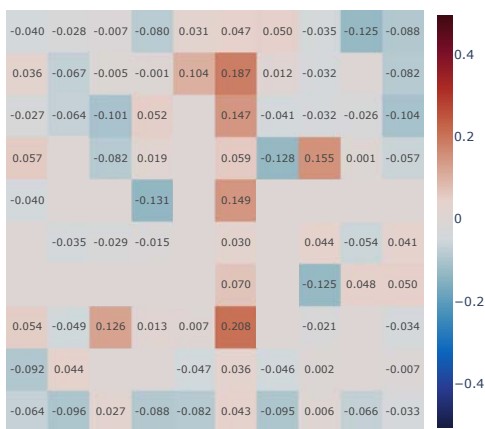

(d) Difference (Before vs After)

Figure 54: Correlation Analysis between Successor Features with reconstruction constraints (SF + Reconstruction) and Successor Representation in the Inverted-LWalls-Grid Environment (Fully-observable)

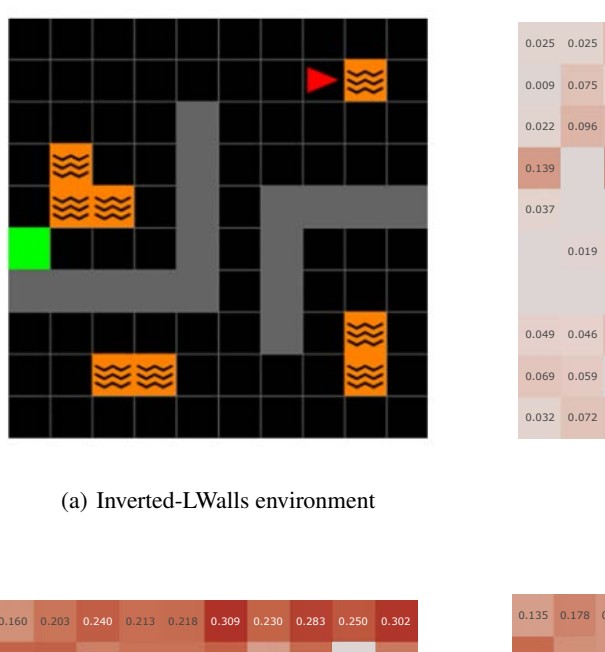

(a) Inverted-LWalls environment

(b) Before Training

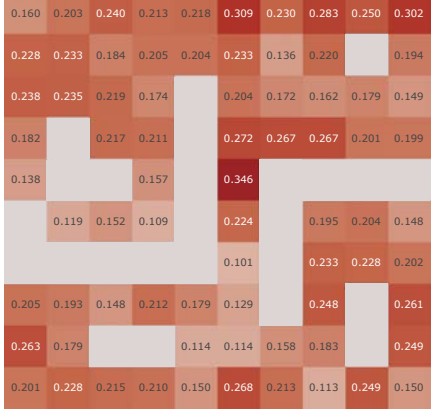

(c) After Training

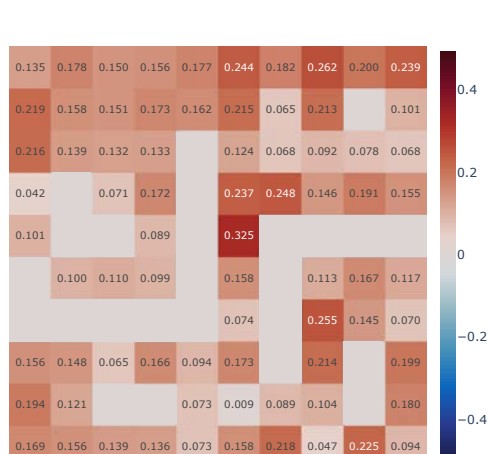

(d) Difference (Before vs After)

Figure 55: Correlation Analysis between APS Pre-train Successor Features [Liu and Abbeel, 2021] and Successor Representation in the Inverted-LWalls-Grid Environment (Fully-observable)

