# OpenReview forum: "Learning Successor Features the Simple Way"
_NeurIPS.cc/2024/Conference — NeurIPS 2024 poster_

### Official Review · Reviewer_nkgZ · 2024-06-21

**Soundness:** 3
**Presentation:** 2
**Contribution:** 3
**Rating:** 5
**Confidence:** 4

**Summary:**

This paper presents a method for learning Successor Features (SFs) from pixel-level observations in reinforcement learning (RL) by combining a Temporal-Difference (TD) loss with a reward prediction loss. This approach simplifies the learning process, improves performance, and speeds up learning compared to existing methods.

**Strengths:**

The proposed method is simple and easy to implement, which is a significant advantage in practical applications. The effectiveness of the approach is demonstrated in both simple 2D and 3D environments.

**Weaknesses:**

1. The writing quality of the paper could be improved. The Introduction section reads more like an extensive review of related work rather than setting the context for the proposed method.
2. The first three subplots in Figure 1 are difficult to understand without detailed background information. It is recommended to move these figures to the experimental section.
3. The statement in lines 61 to 62, "without any of the drawbacks," seems too absolute and should be toned down.
4. The experiments only demonstrate effectiveness in a few simple 2D and 3D environments. To further validate the proposed method, it is recommended to test in more complex environments, such as Atari games, similar to APS.
5. The experiments only tested the method with DQN, leaving its effectiveness with other RL methods unknown.

**Questions:**

Have the authors tested the proposed method with other RL algorithms and in more complex environments?

**Limitations:**

In addition to the limitations mentioned by the authors in the paper, please refer to my comments in the Weaknesses section for further details.

---

> ### Author Rebuttal · Authors · 2024-08-06
>
> Thank you for your feedback. We appreciate the opportunity to clarify and enhance our manuscript based on your observations. Please let us know if there is further clarification we can provide.
>
> # 1. Balancing Context and Review of Related Work
> Thank you for your feedback on the structure of our introduction. We will revise it to better balance the context setting with the review of related work, enhancing both readability and clarity. In response, we will incorporate the following to the introduction section of our manuscript:
>
> Successor Features (SFs) are crucial in continual RL for decoupling environmental dynamics from rewards. Yet, current SF implementations often face representation collapse when learning from pixels, due to reliance on predefined assumptions and extensive pre-training. Our approach addresses these limitations by integrating an innovative neural network architecture that enhances computational efficiency and scalability.
>
> We have validated our method with experiments in 2D and 3D mazes and the Mujoco environment (added during the rebuttal phase), detailed in **Figure 1 of the General Response (GR)**. Our findings show enhanced learning efficiency and adaptability, proving our model's broad applicability in RL scenarios.
>
> # 2. Placement and Purpose of Representation Collapse Analysis Plots in Figure 1.
> Thank you for your feedback on the placement of the representation collapse analysis plots in Figure 1. We positioned these plots early in the manuscript to establish the central motivation of our research. While representation collapse is a well-known issue in Machine Learning, its empirical analysis within the context of SFs is a novel aspect of our work, warranting prominent placement to set the stage for the discussions that follow.
>
> Introducing these plots at the beginning ensures that readers immediately understand the significance of the challenge we are addressing. This approach supports a cohesive narrative by linking the theoretical motivations directly with our proposed solutions and experimental validations.
>
> Moving these plots to a later section, such as the experimental results, could disconnect them from their theoretical context and reduce their impact on framing the research problem.
>
> # 3. The Exclusion of Certain Drawbacks in Our Method
> We appreciate the chance to clarify the use of “without any of the drawbacks” in our manuscript. To address your concern, we will amend the phrase to “without some of the drawbacks.”
>
> # 4. Further evaluation in complex environments
> Thank you for the suggestion to use Atari benchmarks. While APS's Atari setup involves pre-training and fine-tuning within environments that do not vary in features and reward functions, it is less suitable for assessing continual learning capabilities.
>
> Instead, we opted for a comprehensive evaluation in **Mujoco, utilizing pixel-based observations** [6], which further *demonstrates our model's capabilities with continuous actions*. We started in the half-cheetah domain, rewarding agents for running forward in Task 1. For Task 2, we introduced scenarios with running backwards, running faster, and switching to the walker domain. These are detailed in **Figure 1 in the GR**.
>
> Across all scenarios, our model not only maintained high performance but consistently outperformed all baselines in both Task 1 and Task 2, highlighting its superior adaptability and effectiveness in complex environments. This contrasted sharply with other SF-related baseline models, which struggled to adapt under these conditions.
>
> # 5. Exclusivity to DQN
> Thank you for your comment on the scope of our experiments. There are two reasons why we chose the baselines that we did.
>
> **First, we did not only compare to DQN, of course, but to several other techniques for learning SFs.** We did this because the focus of our work is learning SFs. Therefore, we selected other techniques for learning SFs as our primary baselines, such as with reconstruction or orthogonality constraints. Thus, we are comparing our approach to several other techniques, namely, those that are most relevant for the question of learning SFs.
>
> **Second, we chose DQN as a non-SF baseline because of its direct relation to the mathematical definition of SFs and Q-values, a common practice in SFs literature** [1,2,4,5]. This choice helps clarify the specific contributions of our approach in the context of well-understood benchmarks like DQN and DDPG [7].
>
> Moreover, **our primary goal was to develop a straightforward method for learning SFs, not to conduct a comprehensive benchmark across various RL algorithms.** More complex algorithms do not always lead to better performance, especially in settings with pixel-based observations, as shown in comparisons within the Mujoco environment where *simpler algorithms like DDPG often outperform more complex ones like SAC* [8] (see Figure 9a in [6]).
>
> While we acknowledge the value of broadening our evaluations to include a wider array of RL algorithms, our focus was on demonstrating the efficacy of our SF learning approach. Exploring performance with additional algorithms remains an important future research direction to enhance the generalizability of our findings.
>
> [1] Machado et al., 2020. Count-based exploration with the successor representation.
>
> [2] Ma, et al., 2020. Universal successor features for transfer reinforcement learning.
>
> [3] Touati et al., 2023. Does zero-shot reinforcement learning exist?
>
> [4] Janz et al., 2019. Successor uncertainties: exploration and uncertainty in temporal difference learning.
>
> [5] Barreto et al., 2017. Successor features for transfer in reinforcement learning.
>
> [6] Yarats et al., 2021. Mastering visual continuous control: Improved data-augmented reinforcement learning.
>
> [7] Lillicrap et al., 2015. Continuous control with deep reinforcement learning.
>
> [8] Haarnoja et al. 2018. Soft actor-critic algorithms and applications.

---

> > ### Comment · Reviewer_nkgZ · 2024-08-11
> >
> > Thank you to the authors for their answers to my questions and the additional experiments. This has helped me better understand the work. I am currently inclined to accept this paper and maintain my current scores.

---

> ### Author Response · Authors · 2024-08-12
> **Response to Reviewer nkgZ**
>
> We are very pleased to hear that our response helped to answer the reviewer’s questions, and that the reviewer is inclined to accept our paper. Given this, we wonder if the reviewer would be willing to raise their score to reflect the fact that we addressed their questions and put the score more clearly in the “accept” range.

---

> > ### Author Response · Authors · 2024-08-13
> > **Final Day Reminder: Clarifying Concerns and Updating Scores**
> >
> > Dear Reviewer nkgZ,
> >
> > We hope this message finds you well. As today is the final day for the review discussion, We would like to kindly check in to see if our latest response has addressed your concerns. If the clarifications provided have resolved your questions, we would greatly appreciate it if you could update your score accordingly.
> >
> > Thank you once again for your time and thoughtful feedback throughout this process. Your input has been invaluable, and we look forward to hearing from you soon.

---

### Official Review · Reviewer_Tczf · 2024-07-09

**Soundness:** 3
**Presentation:** 2
**Contribution:** 3
**Rating:** 5
**Confidence:** 4

**Summary:**

The paper proposes a simpler method to learn Successor Features that avoids representational collapse. For this, the authors decompose the loss function to learn the successor features and task encoding separately. This allows for keeping the basis features fixed while learning the successor features, thus avoiding representational collapse. The experiments involve a continual learning scenario where robustness to task changes is evaluated. The authors show that the method can better adapt to changing tasks.

**Strengths:**

- The problem is well motivated and the approach offers a simple solution to representational collapse when learning deep successor features.
- The authors provided a number of insightful ablations. Especially that reconstruction based SF methods have trouble learning a good representation for fully observed settings.
- The writing is clear and the method is presented in an understandable manner.

**Weaknesses:**

- The tested environments seem to be perhaps too simple for comparison both from the representational and task difficulty perspective, since DQN also has very good relearning capabilities in these environments. Why is it that DQN is better than the successor feature counterparts for the continual learning setting? This seems counterintuitive to me since successor features should be more robust than pure DQN.

- The presentation of the figures has issues. Some Figures are pixelated, i.e., not vector graphics. (e.g., Figure 4 or other environment Figures). Also I think Figure 1 could be split into 2 figures for better readability.

**Questions:**

- Should the method not be tested on environments where DQN itself cannot adapt to the new tasks at all? I wonder if the simple approach still holds when the transition dynamics become more complicated or the observations are more noisy.
- For the Minigrid environment: Do you learn the successor features from pixels or do you use the built-in symbolic state representations?
- Since DQN is also robust for the showed environments, I wonder how reward sparsity affects the performance of the different algorithms?

**Limitations:**

The authors have adequately addressed limitations.

---

> ### Author Rebuttal · Authors · 2024-08-06
>
> Thank you for your feedback. We appreciate the opportunity to clarify and enhance our manuscript based on your observations. Please let us know if there is further clarification we can provide.
>
> # 1. Is DQN better than the SFs in Continual RL setting?
> Thank you for your observations regarding the experimental results in Figures 2 and 3. While average episode returns offer quick performance insights, they do not fully capture the long-term benefits of our model. Thus, we also analyzed cumulative total returns across all tasks, as shown in **Figure 2 of the general response (GR)**.
>
> These results confirm that our model quickly learns and maintains effective policies, especially in complex 3D environments where tasks recur (**Figure 2c-d in GR**). Our model significantly outperformed the baseline in cumulative returns, demonstrating its robustness and superior transfer capabilities compared to DQN, which showed little to no transfer effects and needed to re-learn tasks.
>
> We will include these results in our manuscript to more comprehensively demonstrate our model's effectiveness in continual learning settings.
>
> # 2. Figure 4 Quality
> Thank you for your feedback on the graphics in our figures. While all our figures are created with vector-based graphics for high resolution and scalability, Figure 4 is an exception. It uses pixel-based graphics to **accurately reflect the native format of the RL environments** and the inputs our models process.
>
> # 3. Figure 1
> We appreciate the suggestion to split Figure 1 to enhance readability. Acknowledging the density of the current figure, we will implement several modifications:
> 1. **Simplification**: We'll remove the loss functions from Figures 1d and 1e, with detailed descriptions retained in Appendix E and the main text, respectively. This will help focus attention on the structural content.
>
> 2. **Reorganization**: Figure 1d will be moved to the Appendix as it primarily presents common approaches rather than our novel contributions, ensuring the main text remains focused on our work.
>
> 3. **Relabeling and Relocation**: Figure 1e will be renamed as Figure 2 and relocated closer to Sections 4 and 5 where it is first mentioned, aligning it more closely with its textual references and enhancing narrative coherence.
>
> 4. **Visual Guidance Enhancements**: We will replace terms like “Q-SF-TD loss” with “$L_\psi$: Q-SF-TD loss” and introduce color-coded information to improve figure-text integration such as, “Pixel-level observations, $S_t$​, are processed by a convolutional encoder to produce a latent representation $h(S_t)$, which is used to construct the basis features (*indicated by a yellow box in Figure 2*) and the SFs (*indicated by a green box in Figure 2*).”
>
> We hope these changes will streamline the presentation and ensure the figures more effectively complement the text.
>
> # 4. Complex and noise in environments
> Thank you for your comment on our model's effectiveness in complex, noisy environments.
>
> Firstly, our model's resilience to noise was proven in the “3D Slippery Four Rooms environment” (Section 6.1.3), where agents faced altered actions in Task 2. The results (Figure 3) demonstrate our model's superior robustness to induced stochasticity compared to baselines.
>
> Secondly, **we expanded our evaluation during the rebuttal phase to include the Mujoco environments**, using pixel-based observations and accounting for *continuous action spaces*. Following the setup in [1], we tested in scenarios like running backwards, running faster, and a major switch from the half-cheetah to the walker domain in Task 2. The outcomes (**Figure 1 in GR**) show our model consistently outperforming baselines across all scenarios, thereby showcasing its adaptability and effectiveness in more complex settings.
> These results affirm our model's advanced capability to robustly handle diverse and challenging environments, making it highly suitable for practical applications with complex dynamics and significant noise.
>
> # 5. Pixel or Symbolic States observations for SFs?
> Thank you for your question regarding the input modalities for Successor Features. In our work, **we exclusively use pixel observations across all experiments**. This choice is intentional, addressing a significant challenge in the field—the direct learning of Successor Features from high-dimensional sensory inputs such as pixels, which, as noted in [2], have historically posed difficulties for conventional methods and remain underexplored in the Successor Features literature [3, 4].
>
> # 6. Sparse Rewards
> Thank you for your question regarding sparse rewards. Like other DQN-based methods, our approach may face challenges in environments with sparse rewards, a recognized issue with bootstrapped learning methods. While our method is tailored for continual reinforcement learning, it is not specifically designed to address sparse rewards.
>
> We acknowledge the need for mechanisms to better manage sparse rewards. Recent findings suggest that reconstruction-based objectives do not always capture task-relevant features effectively in such settings [5]. Integrating techniques that generate intrinsic rewards could help by providing more frequent learning signals.
>
> However, exploring these techniques further is beyond the current scope of our work. Our primary focus remains on demonstrating the viability of our approach in typical continual learning environments, laying the groundwork for future research to more comprehensively tackle the challenges of sparse rewards.
>
> [1] Yarats et al., 2021. Mastering visual continuous control: Improved data-augmented reinforcement learning.
>
> [2] Machado, et al., 2020. Count-based exploration with the successor representation.
>
> [3] Ma, et al., 2001. Universal successor features for transfer reinforcement learning.
>
> [4] Touati et al., 2023. Does zero-shot reinforcement learning exist?
>
> [5] Balestriero., 2024. Learning by Reconstruction Produces Uninformative Features For Perception.

---

> > ### Comment · Reviewer_Tczf · 2024-08-12
> >
> > Thank you for your additional ablations and experiments!
> >
> > * I still feel it is somewhat strange that in Figure 2, DQN is still outperforming other SR methods.  At some point during task change it even outperform your proposed method. I feel the environments don't demonstrate precisely the effectiveness of your method, when DQN is outperforming other SR methods.
> >
> > * The results regarding continual RL are encouraging.
> >
> > I will keep my score, but increase my confidence to 4.

---

> ### Author Response · Authors · 2024-08-12
> **Response to Reviewer Tczf**
>
> Thank you for taking the time to review our rebuttal. We sincerely appreciate your thoughtful comments and are glad to have the opportunity to provide further clarifications. Please don’t hesitate to reach out if you have any additional questions or concerns.
>
> We appreciate your observation, but **we are unclear which specific plot in Figure 2 you are referring to**, as in all Continual RL plots (Figures 2e to 2g in our paper), our approach (orange) consistently outperforms DQN (blue). Additionally, if you refer to the plots generated using the *total cumulative return* in the same setup as Figures 2e to 2g, as shown in **Figures 2a to 2c in the general response**, it is clearly evident that our approach performed much better in the later tasks.
>
> **To emphasize why we presented (moving) average returns per episode instead of cumulative total return plots in our manuscript, it was to demonstrate that we allow learning for the first task to converge before introducing the second and subsequent tasks.**
>
> Furthermore, we acknowledge that the smaller size of these figures might make the trends less apparent. Therefore, we encourage you to refer to the larger illustrations in Appendix G (Figures 12 to 16), where the replay buffer is not reset to simulate conditions with less interference between task switches. Even under these conditions, our approach (orange) consistently demonstrates superior learning performance compared to DQN (blue).
>
> While the performance improvements in the simpler 2D minigrid environments (Center-Wall and Inverted-LWalls) are less pronounced, they remain significant. In contrast, the more complex 3D Four Rooms environment shows a clearer advantage of our method, as seen in Figures 12 and 13. This trend highlights the robustness of our approach, particularly as task complexity increases, further validating the effectiveness of our method across diverse environments.
>
> Moreover, the newly added results during the rebuttal phase, which utilize the more complex Mujoco environment, also show that our method (orange) outperforms DDPG (blue), a variant of DQN designed for continuous actions.
>
> All these results clearly demonstrate that our method, Simple SF (orange), learns more effectively than DQN and DDPG (blue). This superior performance is due to our method's ability to better generalize and transfer knowledge between tasks, as evidenced by the larger improvements in cumulative total returns when the agent re-encounters the tasks (Exposure 2 in Figure 2 in the General Response).

---

> > ### Author Response · Authors · 2024-08-13
> > **Final Day Reminder: Clarifying Concerns and Updating Scores**
> >
> > Dear Reviewer Tczf,
> >
> > We hope this message finds you well. As today is the final day for the review discussion, We would like to kindly check in to see if our latest response has addressed your concerns. If the clarifications provided have resolved your questions, we would greatly appreciate it if you could update your score accordingly.
> >
> > Thank you once again for your time and thoughtful feedback throughout this process. Your input has been invaluable, and we look forward to hearing from you soon.

---

> ### Author Response · Authors · 2024-08-13
> **Response to Reviewer Tczf**
>
> **This is a re-submit as it seems that our earlier previous response did not notify the reviewers via email.**
>
> Thank you for taking the time to review our rebuttal. We sincerely appreciate your thoughtful comments and are glad to have the opportunity to provide further clarifications. Please don’t hesitate to reach out if you have any additional questions or concerns.
>
> We appreciate your observation, but **we are unclear which specific plot in Figure 2 you are referring to**, as in all Continual RL plots (Figures 2e to 2g in our paper), our approach (orange) consistently outperforms DQN (blue). Additionally, if you refer to the plots generated using the *total cumulative return* in the same setup as Figures 2e to 2g, as shown in **Figures 2a to 2c in the general response**, it is clearly evident that our approach performed much better in the later tasks.
>
> **To emphasize why we presented (moving) average returns per episode instead of cumulative total return plots in our manuscript, it was to demonstrate that we allow learning for the first task to converge before introducing the second and subsequent tasks.**
>
> Furthermore, we acknowledge that the smaller size of these figures might make the trends less apparent. Therefore, we encourage you to refer to the larger illustrations in Appendix G (Figures 12 to 16), where the replay buffer is not reset to simulate conditions with less interference between task switches. Even under these conditions, our approach (orange) consistently demonstrates superior learning performance compared to DQN (blue).
>
> While the performance improvements in the simpler 2D minigrid environments (Center-Wall and Inverted-LWalls) are less pronounced, they remain significant. In contrast, the more complex 3D Four Rooms environment shows a clearer advantage of our method, as seen in Figures 12 and 13. This trend highlights the robustness of our approach, particularly as task complexity increases, further validating the effectiveness of our method across diverse environments.
>
> Moreover, the newly added results during the rebuttal phase, which utilize the more complex Mujoco environment, also show that our method (orange) outperforms DDPG (blue), a variant of DQN designed for continuous actions.
>
> All these results clearly demonstrate that our method, Simple SF (orange), learns more effectively than DQN and DDPG (blue). This superior performance is due to our method's ability to better generalize and transfer knowledge between tasks, as evidenced by the larger improvements in cumulative total returns when the agent re-encounters the tasks (Exposure 2 in Figure 2 in the General Response).

---

### Official Review · Reviewer_2gcv · 2024-07-13

**Soundness:** 2
**Presentation:** 2
**Contribution:** 1
**Rating:** 4
**Confidence:** 4

**Summary:**

This work presents a model architecture to learn successor features in reinforcement learning. It consists of optimizing Eqs. (5) and (6), i.e., a loss for learning the features and a loss for learning the task specific weights. It claims to avoid representation collapse. Experiments are conducted in common 2D and 3D tasks to show that the proposed method can achieve better performance and higher sample efficiency.

**Strengths:**

- A simple method that is easy to understand, and the presentation is easy to follow
- Reasonable performance in the experiments

**Weaknesses:**

- Even though the paper claims that using the reward to train the task weights $w$ is new, this approach has been discussed before (Ma et al., 2020). Specifically, it has shown that using the reward $r$ to train $w$ is inferior to using the $Q$ values (Appendix D, Ma et al., 2020). Of course, the two algorithms are not identical, but it remains unclear why using the reward to learn $w$ is the right choice in the current paper. It is necessary to discuss this prior work.
- Several other design choices need further explanation. L130-139 presents multiple design choices for the model architecture without proper discussion. For example, why the L2 normalization and the layer normalization are required? Why do we need to stop gradient at those specific places? It would be better to have ablation studies to show the importance of these choices.
- Experiment results require further analysis. There is barely any analysis or reasoning for the proposed method. Each subsection in Sec.6 ends with "our method is better" without properly addressing **why** it can achieve better performance. This question is not answered in Sec.7 either. Moreover, the improvement over existing methods is only marginal (see Figs.2&3). More importantly, The quantitive results in Table 1 only show marginal improvement over Orthogonality with a significant performance overlap.

Minor comment: Eqs.(5)&(6) are for scalars so a norm is unnecessary.

Reference:

- Ma, C., Ashley, D.R., Wen, J. and Bengio, Y., 2020. Universal successor features for transfer reinforcement learning. *arXiv preprint arXiv:2001.04025*.

**Questions:**

See the weakness above.

**Limitations:**

The authors adequately addressed the limitations

---

> ### Author Rebuttal · Authors · 2024-08-06
>
> Thank you for your feedback. We appreciate the opportunity to clarify and enhance our manuscript based on your observations. Please let us know if there is further clarification we can provide.
>
> # 1. Important differences between the Universal SFs and our approach
> While both our study and [1] utilize reward prediction loss, our approach integrates this with additional losses to directly learn SFs from pixel observations—a significant departure from [1].
>
> **Our model:**
> 1. **Learns basis features directly from pixels**, unlike [1] which uses pre-determined basis features.
>
> 2. **Does not assume prior knowledge of task specifics**, contrasting with [1] where this is required, making our approach more applicable in continually changing environments.
>
> 3. **Integrated SFs directly into the Q-value function**, simplifying and streamlining the learning without the need for redundant losses as seen in [1].
>
> 4. **Ensures SFs play a crucial role** in performance, contrary to [1] where SFs play a minimal role due to the low weighting of SF loss.
>
> # 2, Reward Integration & Stop-gradient operator
> The key distinction in our approach lies in the application of the stop-gradient operator during the learning of the task-encoding vector w with reward prediction loss (Eq. 6 in our manuscript). Unlike in Universal SFs [1]  where the basis features and the task-encoding vector w are learned concurrently, our approach prevents the updates to the basis features during this learning phase using a stop-gradient operator. This difference is crucial as we further demonstrated in an ablation study, **'Basis-Rewards' (Figure 4 in GR)**, since concurrent learning has shown to degrade learning efficiency (Figure 10 in Appendix D of [1]).
>
> # 3. Insufficient analysis
> We respectfully disagree with the assertion that our work contains “barely any analysis.” Our work goes beyond theoretical discussions of representation collapse by providing empirical evidence (Figures 1a-c) and a **clustering analysis** (Figure 1c) that validate our method's effectiveness. This is complemented by a **mathematical proof sketch in Appendix C**, which explicates the gradient projections in our model, enhancing its applicability in continual learning scenarios. Our comprehensive analysis also includes **computational overhead comparisons** (Figure 6) and ablation studies (**Figure 4 in GR**) that reinforce the efficiency and effectiveness of our approach. Additionally, we will include a proof sketch detailing conditions under which representation collapse can occur (see rebuttal to reviewer 7tgV).
>
> # 4. Improvements are marginal
> Thank you for your observations regarding the experimental results, specifically highlighted in Figures 2 and 3. Your feedback aligns with concerns previously noted by Reviewer Tczf regarding the apparent modest gains when measured using average episode returns.
>
> While average episode returns offer quick performance insights, they don't capture the full benefits of our approach. Hence, we've also evaluated cumulative total returns across tasks, which better reflect the agent’s ability to quickly learn and maintain effective policies over time
>
> Our analysis included in **Figure 2 in GR**, which consistently demonstrates significant improvements from our model compared to the baselines across various environments. Specifically, in the complex 3D environment, our model demonstrated significant improvement in cumulative returns, especially when the agent re-encountered previous tasks, *highlighting its enhanced transfer capabilities and effectiveness* in continual learning scenarios.
>
> # 5. L2 normalization and layer-norm
> L2-normalization is applied to both the basis features and the task-encoding vector w as commonly done [2,5]. We also normalize w before it enters the Features-Task network. These normalisations are to ensure consistent scale across inputs, enhancing optimization, training stability and preventsing any single feature from disproportionately influencing the learning process due to scale differences.
>
> Additionally, we use layer-normalization within the Features-Task network to address un-normalized outputs from the encoder, a practice well-established in deep reinforcement learning to improve model robustness by conditioning the gradients [3,4,6,7].
>
> # 6. Marginal correlation improvement over orthogonality
> Thank you for your observation regarding the correlation improvements of SFs learned using our model. It's true that models enforcing orthogonality on features might show high correlations with discrete one-hot SRs due to their structured nature.
>
> However, our empirical findings, presented in Figures 2 and 3 of the manuscript, highlight that despite possible high correlation, SFs learned with orthogonality constraints often suffer from significant learning deficiencies. This issue becomes even more evident in the challenging Mujoco environments, as detailed in **Figure 1 of GR**.
>
> Furthermore, maintaining orthogonality constraints demands considerably more computational resources (Figure 6). Thus, while improvements in correlation might seem modest, our model offers a more balanced approach in optimizing both performance and computational efficiency.
>
> # 7. Norm is unnecessary
> Thank you for the comment and we will make the revision in the final version.
>
> [1] Ma, et al., 2020. Universal successor features for transfer reinforcement learning
>
> [2] Machado et al., 2020. Count-based exploration with the successor representation.
>
> [3] Yarats et al., 2021. Improving sample efficiency in model-free reinforcement learning from images.
>
> [4] Yarats et al., 2021. Mastering visual continuous control: Improved data-augmented reinforcement learning.
>
> [5] Liu, et al., 2021. Aps: Active pretraining with successor features.
>
> [6] Ball et al., 2023. Efficient online reinforcement learning with offline data.
>
> [7] Lyle et al., 2024. Normalization and effective learning rates in reinforcement learning.

---

> > ### Comment · Reviewer_2gcv · 2024-08-11
> >
> > I thank the authors for the additional results and clarifications. Yet, there are some concerns:
> >
> > # 1. Differences
> >
> > 1\. [1] also uses a NN model to learn the basis features $\phi$ of a state (see Fig.1 of [1]). Whether the feature extractor is pixel-based or not depends on the task.
> >
> > 3\. [1] also integrated the SFs directly into the Q-value function (Eq.(3) of [1]).
> >
> > 4\. There is no connection between "crucial role" and large weighting. There is no guarantee that larger weighting indicates better performance either, as the scales of different losses can be vastly different across tasks. In fact, I found this argument contradicts point 3 above and also defeats the main point of the current paper. The current paper argues that the canonical SF loss is problematic (Fig.1), but now the rebuttal said that one needs to have a larger weighting for the canonical SF loss so that the SFs can play a "crucial role," whatever that means.
> >
> > # 4. Evaluation
> >
> > It is unclear why the average episode returns and the cumulative total returns can show different trends. Isn't the former equal to the latter divided by the number of test/evaluation runs?

---

> ### Author Response · Authors · 2024-08-12
> **Response to Reviewer 2gcv on differences with Ma et al. [1]**
>
> Thank you for taking the time to review our rebuttal. We sincerely appreciate your thoughtful comments and are glad to have the opportunity to provide further clarifications. Below, we respond to your excellent points. Please don’t hesitate to reach out if you have any additional questions or concerns.
>
> # 1. Differences
> ## 1.1 Basis features
>
> First, we see now that we made a mistake in our rebuttal, indeed, in [1] the basis features are learned. As well, the reviewer is correct that, in principle, there is no reason that the approach in [1] could not be applied to pixels. However, to the best of our knowledge, in the paper itself, the authors did not perform experiments and studies involving pixel-based observations. Instead, experiments in [1] were conducted using state inputs, which is likely why their architecture consisting of fully connected networks worked well (Appendix F in [1]). In addition, we ran experiments with the loss from [1] to make a more direct comparison. Please see below for the description of those experiments and the results.
>
> ## 1.3. Direct SF Integration in Q-Learning should eliminate the need for redundant Canonical SF Loss
>
> Thank you for your observation. Indeed, [1] does integrate SFs directly into the Q-value function, similar to our approach, but there is a key difference. **[1] relies on an additional SF loss (Eq. (4) in [1]), known as the Canonical SF-TD loss in our paper, which our method does not require.** Our main contribution lies in the simple (but we believe elegant) architectural design that allows the SFs to be learned directly through the Q-learning loss, eliminating the need for a separate SF loss.
>
> To highlight the impact of this difference, **we conducted experiments comparing our approach to an agent that combines the Q-learning loss, SF loss, and reward prediction loss, similar to the setup in [1].** Notably, we included the reward prediction loss because, unlike [1], our method does not assume prior knowledge of task specifics, such as goals, which aligns with the expectation in continual learning scenarios. We named this approach “SF + Q-TD + Reward.”
>
> Our results, presented in Figure 6a, demonstrate that the additional SF loss can impair learning efficiency, requiring more time steps to converge to a good policy in the complex 3D Four Rooms environment. Furthermore, this approach is significantly less computationally efficient, as shown by the slower computational speed and longer training duration in Figure 6b. For a detailed comparison of learning performance, **please refer to Figures 17 to 21 in Appendix H.**
>
> We believe these findings underscore the advantages of our method, particularly in terms of efficiency and practicality for continual learning. Moreover, they help to illustrate the key differences between the formulation of our approach from [1]. We agree with the reviewer that our method clearly builds on [1] (which was a seminal paper), but we do feel that what we build on this work represents a novel contribution that can help the field to learn SFs more efficiently, as evidenced by our data.
>
> ## 1.4. Avoiding arbitrary weighting adjustments of Separate SF loss
>
> Thank you for your feedback. We appreciate the opportunity to clarify our position.
>
> On reflection, any claim as to whether or not the SF loss plays a “crucial role” should not hinge on something as basic as the weighting term in the loss. Indeed, the lower weighting used in [1] ($\lambda > 0$) *may be a result of differing scales among the losses or potential conflicts between the SF loss and the Q-learning loss.*
>
> But, from a practical point of view, it is fair to say that [1] requires a careful selection of the weighting coefficient $\lambda$. In contrast, our proposal, through careful algorithmic and architectural design, eliminates the need for a separate SF loss and any concerns about its weighting. In doing so, we directly mitigate the potential problems associated with down-weighting the SF loss, ensuring that SFs meaningfully contribute to the agent's performance.
>
> We hope this addresses your concerns. After reading your reply, we feel that our initial rebuttal did not accurately capture the essence of why our contribution is unique and novel relative to [1]. But, as we described above, **the key advantages of our method are that: 1) We do not need to provide the goal to the agent (it is learned); (2) We provide direct evidence that we can learn off of pixel inputs; (3) We show that we do not need to include the SF loss; and (4) By eliminating the need for the SF loss, we reduce the number of hyperparameters required.**

---

> ### Author Response · Authors · 2024-08-12
> **Response to Reviewer 2gcv on Evaluation**
>
> # 4. Evaluation
> Thank you for your comments, and we apologize for any confusion. To clarify, the average episode return is calculated as a moving average over recent episodes—typically the last 100 episodes experienced by the agent in our case. This metric provides a more immediate snapshot of the agent’s recent performance.
>
> In contrast, the cumulative total return is the sum of all returns accumulated from the moment the agent is first exposed to the current task until the end of the evaluation for that task. This metric reflects the overall performance across the entire evaluation period.
>
> These two metrics can show different trends because the moving average episode return emphasizes recent performance, which may fluctuate, while the cumulative total return captures the long-term accumulation of rewards.
>
> **To emphasize why we presented moving average returns per episode instead of cumulative total return plots in our manuscript, it was to demonstrate that we allow learning for the first task to converge before introducing the second and subsequent tasks.**
>
> We hope this explanation resolves the confusion, and we will ensure that these differences are clearly explained in the manuscript.

---

> > ### Author Response · Authors · 2024-08-13
> > **Final Day Reminder: Clarifying Concerns and Updating Scores**
> >
> > Dear Reviewer 2gcv,
> >
> > We hope this message finds you well. As today is the final day for the review discussion, We would like to kindly check in to see if our latest response has addressed your concerns. If the clarifications provided have resolved your questions, we would greatly appreciate it if you could update your score accordingly.
> >
> > Thank you once again for your time and thoughtful feedback throughout this process. Your input has been invaluable, and we look forward to hearing from you soon.

---

> ### Author Response · Authors · 2024-08-13
> **Response to Reviewer 2gcv on differences with Ma et al. [1]**
>
> **This is a re-submit as it seems that our earlier previous response did not notify the reviewers via email.**
>
> Thank you for taking the time to review our rebuttal. We sincerely appreciate your thoughtful comments and are glad to have the opportunity to provide further clarifications. Below, we respond to your excellent points. Please don’t hesitate to reach out if you have any additional questions or concerns.
>
> # 1. Differences
> ## 1.1 Basis features
>
> First, we see now that we made a mistake in our rebuttal, indeed, in [1] the basis features are learned. As well, the reviewer is correct that, in principle, there is no reason that the approach in [1] could not be applied to pixels. However, to the best of our knowledge, in the paper itself, the authors did not perform experiments and studies involving pixel-based observations. Instead, experiments in [1] were conducted using state inputs, which is likely why their architecture consisting of fully connected networks worked well (Appendix F in [1]). In addition, we ran experiments with the loss from [1] to make a more direct comparison. Please see below for the description of those experiments and the results.
>
> ## 1.3. Direct SF Integration in Q-Learning should eliminate the need for redundant Canonical SF Loss
>
> Thank you for your observation. Indeed, [1] does integrate SFs directly into the Q-value function, similar to our approach, but there is a key difference. **[1] relies on an additional SF loss (Eq. (4) in [1]), known as the Canonical SF-TD loss in our paper, which our method does not require.** Our main contribution lies in the simple (but we believe elegant) architectural design that allows the SFs to be learned directly through the Q-learning loss, eliminating the need for a separate SF loss.
>
> To highlight the impact of this difference, **we conducted experiments comparing our approach to an agent that combines the Q-learning loss, SF loss, and reward prediction loss, similar to the setup in [1].** Notably, we included the reward prediction loss because, unlike [1], our method does not assume prior knowledge of task specifics, such as goals, which aligns with the expectation in continual learning scenarios. We named this approach “SF + Q-TD + Reward.”
>
> Our results, presented in Figure 6a, demonstrate that the additional SF loss can impair learning efficiency, requiring more time steps to converge to a good policy in the complex 3D Four Rooms environment. Furthermore, this approach is significantly less computationally efficient, as shown by the slower computational speed and longer training duration in Figure 6b. For a detailed comparison of learning performance, **please refer to Figures 17 to 21 in Appendix H.**
>
> We believe these findings underscore the advantages of our method, particularly in terms of efficiency and practicality for continual learning. Moreover, they help to illustrate the key differences between the formulation of our approach from [1]. We agree with the reviewer that our method clearly builds on [1] (which was a seminal paper), but we do feel that what we build on this work represents a novel contribution that can help the field to learn SFs more efficiently, as evidenced by our data.
>
> ## 1.4. Avoiding arbitrary weighting adjustments of Separate SF loss
>
> Thank you for your feedback. We appreciate the opportunity to clarify our position.
>
> On reflection, any claim as to whether or not the SF loss plays a “crucial role” should not hinge on something as basic as the weighting term in the loss. Indeed, the lower weighting used in [1] ($\lambda > 0$) *may be a result of differing scales among the losses or potential conflicts between the SF loss and the Q-learning loss.*
>
> But, from a practical point of view, it is fair to say that [1] requires a careful selection of the weighting coefficient $\lambda$. In contrast, our proposal, through careful algorithmic and architectural design, eliminates the need for a separate SF loss and any concerns about its weighting. In doing so, we directly mitigate the potential problems associated with down-weighting the SF loss, ensuring that SFs meaningfully contribute to the agent's performance.
>
> We hope this addresses your concerns. After reading your reply, we feel that our initial rebuttal did not accurately capture the essence of why our contribution is unique and novel relative to [1]. But, as we described above, **the key advantages of our method are that: 1) We do not need to provide the goal to the agent (it is learned); (2) We provide direct evidence that we can learn off of pixel inputs; (3) We show that we do not need to include the SF loss; and (4) By eliminating the need for the SF loss, we reduce the number of hyperparameters required.**

---

> ### Author Response · Authors · 2024-08-13
> **Response to Reviewer 2gcv on Evaluation**
>
> **This is a re-submit as it seems that our earlier previous response did not notify the reviewers via email.**
>
> # 4. Evaluation
> Thank you for your comments, and we apologize for any confusion. To clarify, the average episode return is calculated as a moving average over recent episodes—typically the last 100 episodes experienced by the agent in our case. This metric provides a more immediate snapshot of the agent’s recent performance.
>
> In contrast, the cumulative total return is the sum of all returns accumulated from the moment the agent is first exposed to the current task until the end of the evaluation for that task. This metric reflects the overall performance across the entire evaluation period.
>
> These two metrics can show different trends because the moving average episode return emphasizes recent performance, which may fluctuate, while the cumulative total return captures the long-term accumulation of rewards.
>
> **To emphasize why we presented moving average returns per episode instead of cumulative total return plots in our manuscript, it was to demonstrate that we allow learning for the first task to converge before introducing the second and subsequent tasks.**
>
> We hope this explanation resolves the confusion, and we will ensure that these differences are clearly explained in the manuscript.

---

### Official Review · Reviewer_7tgV · 2024-07-14

**Soundness:** 3
**Presentation:** 3
**Contribution:** 2
**Rating:** 6
**Confidence:** 5

**Summary:**

This work introduces a new algorithm for training successor features in deep reinforcement learning. This is achieved by optimizing two separate metrics. The first requires the model to predict the cumulative reward following a full trajectory and optimizes the successor features and the basis feature. The second optimized metric requires the model to predict the reward at the next step using the basis feature and task vector, but in this case only optimizes the task vector. Experiments are conducted in grid worlds and a 3D Four Rooms domain which demonstrates that the proposed algorithm learns more consistently than other SF baselines and also supports faster task switching. Finally, supporting experiments show that the proposed algorithm is faster algorithmically and in terms of wall clock time than SF alternatives and that it also results in more separable SFs which correlate better with successor representations.

**Strengths:**

# Originality
The decoupling of training into separate separate equations is new an intuitive idea. The authors note the inspiration from Liu et. al. (2021) however this algorithm is used as a baseline and clearly out-performed empirically.  Thus, it is clear that the changes made are material and have an impact on model performance.

# Clarity
The paper is well written and sections are structured appropriately. Notation is intuitive, consistent and aids understanding. Figure captions are detailed which also aids clarity.

# Quality
I particularly appreciate some of the additional experiments conducted in support of the algorithm, such as the correlation between the learned SFs and SRs. The core experiments appear sufficiently challenging to separate the proposed algorithm from the baselines, and the baselines which are used are appropriate to challenge the proposed algorithm, The results are interpreted fairly as all algorithms do struggle on at least one domain where the simple SFs do not and consistently perform well.

# Significance
I do think this work could lead to future work and provide a helpful step in improving SFs and making them more practical. The significance is aided by the originality and simplicity of the approach as it is likely to spur new ideas quickly as a result.

**Weaknesses:**

# Clarity
The figures in this work are laid out poorly and this significantly hinders the readability of the paper. At the least it would help if a reader does not have to look past unseen figures on their way to look at the one being referenced - such as when looking for Figure 6 which comes after Figure 5. Also having the architectures in Figure 1 far from where they are needed and completely out of context is jarring and unhelpful. These architecture diagrams are also very difficult to follow and there is not clear mapping from what is depicted for some of the pieces to any explanation in the text or caption. Most of what is depicted in Figure 1d is not in Section 3, and similarly Section 5 is not detailed enough for me to map onto Figure 1e. I think more detail could be added to the figure itself and to the caption here. Lastly, including the loss functions in the figure, especially ones which have not been explained in the text, like orthogonality loss, is confusing. If these losses are not necessary in the main text then I don't think it is necessary in the figure and so I would remove them. With respect to Proposition 1, it would be better if this was at least in the main text - some kind of proof sketch would be better - but in the interest of space I see why it was omitted. Once again I do then just point to this as a part which could do with more explanation of why this matters and intuition on what it is true. If I am correct, Proposition 1 is the reason why Equations 5 and 6 cannot be optimized with the representation collapse strategy?

Secondly, the Preliminaries section could have more elaboration. Equation 2 in particular is presented without any discussion and $\gamma$ is not introduced at all. A reader with experience on RL and SFs will be fine but less experienced readers will likely be alienated. It would be ideal if this section could set up the ideas to come, and this is done to a degree with it being noted that representation collapse can still optimize Equation 4. More of this insight would just be helpful. Similarly, I would appreciate more discussion on why representation collapse is now not able to optimize Equation 5 and 6. This is merely stated without reason on lines 113 to 115.

# Quality
I am not certain I agree with the assessment from Section 7.2. While it is a worth experiment, the result of simple SFs being more correlated just appears to be due to the fact that it has a more linear latent embedding. This can be seen in Figure 5. SF+Reconstruction has very separable and clear clusters but they are just not organised in a straight line. So correlations - a linear metric - will not work. It seems to me that it would be more appropriate to try decode SRs from the SFs using a simple but nonlinear model and report the final accuracy.

**Questions:**

I have asked some questions in the my review above and would appreciate those be answered. I do not have any other questions at this time.

If my question on Proposition 1 is answers and makes sense to me, and the concerns in Quality address I would be likely to advocate for acceptance, with agreement that the clarity would also be improved and figures restructured.

**Limitations:**

The limitations are stated in their own section and a good of consideration given towards the broader impact of this work.

---

> ### Author Rebuttal · Authors · 2024-08-06
>
> Thank you for your feedback. We appreciate the opportunity to clarify and enhance our manuscript based on your observations. Please let us know if there is further clarification we can provide.
>
> # 1. Layout of Figure 5 and Figure 6
> Thank you for your feedback on the order of Figures 5 and 6 in our manuscript. We recognize that aligning figure placement with their respective sections will enhance the manuscript's readability and coherence.
>
> Currently, Figure 6 is introduced in Section 7.1's “Efficiency Analysis,” and Figure 5 appears later in Section 7.2's “Comparison to Successor Representations.” To improve narrative flow, we propose to swap these sections. This rearrangement will position 'Comparison to Successor Representations' before 'Efficiency Analysis,' ensuring that the figures align more logically with their related discussions.
>
> # 2. Modifications of Figure 1
> Thank you for your valuable feedback regarding Figure 1. Due to space constraints in this rebuttal response, we invite you to refer to our detailed response provided to Reviewer Tczf under the section 'Splitting Figure 1' and in the general response above.
>
> # 3. Motivations behind Proposition 1
> Thank you for the suggestion. Proposition 1 aims to mathematically demonstrate that the gradients from optimizing the Q-SF-TD loss (Eq. 5) effectively project the gradients from the canonical SF-TD loss (Eq. 4) along the task-encoding vector w. This projection is crucial in Continual RL as it aligns the SFs with different tasks, enabling the agent to adapt more rapidly to varying tasks.
>
> However, your comment on Proposition 1 being the reason why representation collapse is being mitigated is incorrect. For clarity on why representation collapse can occur, we have included an additional proof sketch below.
>
> # 4. Proof Sketch for Representation Collapse in Basis Features
>
> Consider the basis features function $\phi(\cdot) \in \mathbb{R}^n$ and the SFs $\psi(\cdot) \in \mathbb{R}^n$, omitting the inputs for clarity. The canonical SF-TD loss (Eq. 4) is defined as:
>
> \begin{align}
>     L_{\phi, \psi} = \frac{1}{2} \left \| \| \phi(\cdot) + \gamma \psi (\cdot) - \psi(\cdot) \right \| \|^2
> \end{align}
>
> Assume both $\phi(\cdot)$ and $\psi(\cdot)$ are constants across all states $S$, such that $\phi(\cdot) = c_1$ and $\psi(\cdot) = c_2$.
> If $c_1 = (1-\gamma)c_2$, then:
>
> $L_{\phi, \psi} = \frac{1}{2} \left \| \| (1 - \gamma)c_2 + \gamma c_2 - c_2 \right \| \|^2 = 0$
>
> This scenario illustrates that if both the basis features and SFs become constants, particularly with $c_1 = (1-\gamma)c_2$, the system will satisfy the zero-loss conditions, resulting in representation collapse. In this state, $\phi(\cdot)$ loses its ability to distinguish between different states effectively, causing the model to lose critical discriminative information and thus impair its generalization capabilities.
>
> # 5. Introduction to RL in Preliminaries
> Thank you for the comment. We will add the following text to section 3 to aid readers who may not be familiar with RL.
>
> The RL setting is formalized as a Markov Decision Process defined by a tuple $(S, A, p, r, \gamma)$, where $\mathcal{S}$ is the set of states, $\mathcal{A}$ is the set of actions, $r: S \rightarrow \mathbb{R}$ is the reward function, $p: \mathcal{S} \times \mathcal{A} \rightarrow [0,1]$  is the transition probability function and $\gamma \in [0,1)$ is the discount factor which is being to used to balance the importance of immediate and future rewards.
>
> At each time step $t$, the agent observes state $S_t \in \mathcal{S}$ and takes an action $A_t\in \mathcal{A}$ sampled from a policy $\pi: \mathcal{S} \times \mathcal{A} \rightarrow [0,1]$, resulting in to a transition of next state $S_{t+1}$ with probability $p(S_{t+1} \mid S_t, A_t)$ and the reward $R_{t+1}$.
>
> # 6. Linear Latent Embeddings and Correlation Analysis
> Thank you for your comment. We welcome the chance to clarify our use of UMAP for embeddings (Figure 5) and Spearman's rank correlation for analysis (Table 1).
>
> First, we use UMAP [1], a non-linear dimension reduction technique, for its effectiveness in visualizing complex relationships within Successor Features (SFs) in 2D space. It's crucial to note that UMAP does not imply linearity; the spatial arrangement of clusters should not be interpreted as linear relationships among features.
>
> Second, our correlation analysis employs Spearman's rank correlation coefficient [2], outlined in Appendix K. This method assesses monotonic, non-linear relationships, suitable for our data's characteristics. Contrary to any suggestions of linearity, Spearman's correlation is non-parametric and does not assume linear relationships.
>
> We will clarify these points in the manuscript to eliminate any ambiguity about our methods and to underscore the appropriateness and robustness of our analysis.
>
> # 7. Decoding SRs from SFs
> Thank you for suggesting we use a simple nonlinear model to decode SRs from SFs. We implemented a single-layer perceptron with ReLU activation, training it for 4000 iterations using a 0.001 learning rate and Adam optimizer to ensure convergence within the center-wall environment.
>
> In **Figure 3 of the General Response**, we present the mean squared error (MSE) results for both fully-observable (allocentric) and partially-observable (egocentric) settings. Our model achieved notably low errors, outperforming baselines in both contexts, highlighting its robustness and the effectiveness of our successor features in varying observational settings. This consistency was not observed in baseline models, such as SF + Random (green) and SF + Reconstruction (red), which showed variable performance.
>
> These results confirm the strength and reliability of our decoded successor representations across diverse settings. We will incorporate this analysis into the manuscript.
>
> [1]  Leland et al., 2018. Umap: Uniform manifold approximation and projection.
>
> [2] Zar et al., 2005. Spearman rank correlation.

---

> > ### Comment · Reviewer_7tgV · 2024-08-12
> > **Response to Rebuttal by Reviewer 7tgV**
> >
> > I thank the authors for their response. I make note of the agreed changes to the figures and their layout in the general comment, this rebuttal and in the discussion with Reviewer Tczf. These seem like appropriate changes and address my concerns with the figures.
> >
> > For the points on the motivation of Proposition 1 and the proof sketch for representation collapse. I understood how if the $\phi$ and $\psi$ become constant and the loss is minimized. My point is that the insight of the proposition is missing. For example, the sentence in this rebuttal: "This projection is crucial in Continual RL as it aligns the SFs with different tasks, enabling the agent to adapt more rapidly to varying tasks." is key but even then the insight would come from showing how the projection aligns with the task encoding vector and showing or at least arguing why this helps with downstream tasks. I appreciate that the proof and some insight of this sort is in the appendix, but in reality this is the primary contribution and deeper insight here, and is comes in as a passing comment. Perhaps I am missing the point of the work, but the fact that the proposed method behaves in this way seems crucial. In addition, since the representation collapse strategy to the standard SR is noted as being the primary problem, showing how projecting along the task vector fixes this is key. Relying in the main text only on empirics denies the reader this deeper insight, or at least expects them to go and read the proof and obtain the insight themselves from it.
> >
> > With respect to the RL preliminaries. I appreciate the authors agreeing to this, as it must feel like a nuisance. However, I think it is in the best interest of the broader readership of NeurIPS.
> >
> > For the correlation analysis. I thank the authors for correcting me, and agree that mention of this would be helpful in the main text. As I mention this was one of my main concerns and I will be raising my score as a result. I also appreciate the new experiments added to the pdf draft and find this to be compelling evidence. I thank the authors for including the decoding experiment as well and find it convincing.
> >
> > Ultimately, my lingering concern remains with Proposition 1. Essentially lines 113 to 115 explain to us the outcome of the proposed setup, lines 116 to 118 then state literally the Proposition 1 (given it's proximity to lines 113 to 115 leading me to assume it was more related than it is) and then there is a throw forward to the empirical results. So from lines 113 to 119 where the main purpose of the proposed method is being summarize a reader is told what to think, but never shown it. This makes the entire section fall flat, and since this is the technical punchline it makes the paper fall slightly flat. I believe the experimental results (adjusting for my own misunderstandings and the new results) show the intended meaning and I am confident in the correctness of the claims of the work. But a deeper insight into how the proposed method really results in learned SRs and avoid representation collapse still seems missing. Proposition 1, or something of this nature would likely address this. I would raise my score further if this was addressed in the coming days.
> >
> > I once again thank the authors for their thoughtful response and new experiments. As my quality concerns were due to an error in my understanding, and this has now been corrected, I will raise my score to a 5 and also increase my confidence. I am also raising the soundness and clarity scores in light of the correction and improved figures. I look forward to further discussion on Proposition 1 if the authors are able.

---

> ### Author Response · Authors · 2024-08-12
> **Response to Reviewer 7tgV**
>
> Thank you for taking the time to review our rebuttal. We sincerely appreciate your thoughtful feedback and the subsequent adjustment in your evaluation.
>
> In addition, we would like to take this opportunity to further clarify certain points, specifically regarding how our proposed method results in learned Successor Features (SFs) and avoids representation collapse as well as Proposition 1.
>
> # Improving Clarification on Overcoming Representation Collapse
> We would like to thank the reviewer for engaging constructively with us, their input has been extremely helpful for improving our paper. Below, we propose two additional modifications to the manuscript to address the points raised by the reviewer in their response to our rebuttal.
>
> First, we will incorporate the proof sketch regarding representation collapse into the main text near line 100, where we initially mention the scenario where the basis features $\phi$ may become a constant vector when the loss is minimized.
>
> Second, at the beginning of Section 4, “Proposed Method,” we will emphasize that the key insight from the proof sketch is that preventing representation collapse requires avoiding the scenario where the basis features $\phi$ become a constant vector for all states, which would minimize the loss without contributing to meaningful learning.
>
> Our approach addresses these constraints by not optimizing the basis features $\phi$ within any loss functions used. Instead, we treat the basis features $\phi$ as the normalized output from the encoder, which is learned using the Q-SF-TD loss (Eq. 5). When the basis features $\phi$ are needed to learn the task encoding vector $w$ through the reward prediction loss (Eq. 6), we apply a stop-gradient operator to treat the basis features $\phi$ as a constant. As we will demonstrate in section 7 “Analysis of Efficiency and Efficacy”, this inclusion of a stop-gradient operator is crucial. Without it, learning both the basis features $\phi$ and the task encoding vector $w$ concurrently can lead to learning instability (as we explained to Reviewer 2gcv).
>
> # Improving Clarification for Proposition 1
>
> Regarding Proposition 1, in re-reading our own text, we must admit that we agree completely with the reviewer. The way that the text jumps from the discussion of representational collapse in lines 113-115, then brings up Proposition 1, it is natural for a reader to assume that Proposition 1 will deal with representational collapse, and yet it doesn’t. We can see now that this would have caused confusion for readers, potentially suggesting a misleading connection between Proposition 1 and representation collapse.
>
> To improve clarity, we propose the following amendments:
>
> 1. Add a concluding sentence after line 115 where we will state, “Next, we will clarify how our approach relates to learning SFs, as they are defined mathematically.”
>
> 2. Create a new subsection titled “4.1 Bridging Simple SFs and Universal Successor Features,” where we will expand on the insights related to Proposition 1 (expanding the text currently in lines 116-119).
>
> In terms of expanding on the insights related to Proposition 1, we will highlight the fact that Proposition 1 explains why our approach ultimately produces true SFs. Proposition 1 does this by proving that minimizing our losses (Eqs. 5 & 6) also minimizes the canonical SF loss used in Universal Successor Features (Eq. 4). In order to tie this to the previous section, we will also note that our approach minimizes these losses in a manner such that setting the basis features $\phi$ to a constant is not a solution.
> Specifically, we will note in the text that if one sets $\psi = c_2 $ and $\phi = c_1 = (1 - \gamma) c_2$,  then Eqs. 5 & 6 are not minimized, due to the fact that $\hat{y}$ and $R_{t+1}$ are not constants.
>
> We believe these revisions will significantly enhance the clarity and robustness of our manuscript. Again, we thank you for your insightful feedback, and please feel free to reach out if any further clarification is needed. We hope you will consider raising your score once more based on these responses.

---

> > ### Author Response · Authors · 2024-08-13
> > **Final Day Reminder: Clarifying Concerns and Updating Scores**
> >
> > Dear Reviewer 7tgV,
> >
> > We hope this message finds you well. As today is the final day for the review discussion, We would like to kindly check in to see if our latest response has addressed your concerns. If the clarifications provided have resolved your questions, we would greatly appreciate it if you could update your score accordingly.
> >
> > Thank you once again for your time and thoughtful feedback throughout this process. Your input has been invaluable, and we look forward to hearing from you soon.

---

> ### Author Response · Authors · 2024-08-13
> **Response to Reviewer 7tgV**
>
> **This is a re-submit as it seems that our earlier previous response did not notify the reviewers via email.**
>
> Thank you for taking the time to review our rebuttal. We sincerely appreciate your thoughtful feedback and the subsequent adjustment in your evaluation.
>
> In addition, we would like to take this opportunity to further clarify certain points, specifically regarding how our proposed method results in learned Successor Features (SFs) and avoids representation collapse as well as Proposition 1.
>
> # Improving Clarification on Overcoming Representation Collapse
> We would like to thank the reviewer for engaging constructively with us, their input has been extremely helpful for improving our paper. Below, we propose two additional modifications to the manuscript to address the points raised by the reviewer in their response to our rebuttal.
>
> First, we will incorporate the proof sketch regarding representation collapse into the main text near line 100, where we initially mention the scenario where the basis features $\phi$ may become a constant vector when the loss is minimized.
>
> Second, at the beginning of Section 4, “Proposed Method,” we will emphasize that the key insight from the proof sketch is that preventing representation collapse requires avoiding the scenario where the basis features $\phi$ become a constant vector for all states, which would minimize the loss without contributing to meaningful learning.
>
> Our approach addresses these constraints by not optimizing the basis features $\phi$ within any loss functions used. Instead, we treat the basis features $\phi$ as the normalized output from the encoder, which is learned using the Q-SF-TD loss (Eq. 5). When the basis features $\phi$ are needed to learn the task encoding vector $w$ through the reward prediction loss (Eq. 6), we apply a stop-gradient operator to treat the basis features $\phi$ as a constant. As we will demonstrate in section 7 “Analysis of Efficiency and Efficacy”, this inclusion of a stop-gradient operator is crucial. Without it, learning both the basis features $\phi$ and the task encoding vector $w$ concurrently can lead to learning instability (as we explained to Reviewer 2gcv).
>
> # Improving Clarification for Proposition 1
>
> Regarding Proposition 1, in re-reading our own text, we must admit that we agree completely with the reviewer. The way that the text jumps from the discussion of representational collapse in lines 113-115, then brings up Proposition 1, it is natural for a reader to assume that Proposition 1 will deal with representational collapse, and yet it doesn’t. We can see now that this would have caused confusion for readers, potentially suggesting a misleading connection between Proposition 1 and representation collapse.
>
> To improve clarity, we propose the following amendments:
>
> 1. Add a concluding sentence after line 115 where we will state, “Next, we will clarify how our approach relates to learning SFs, as they are defined mathematically.”
>
> 2. Create a new subsection titled “4.1 Bridging Simple SFs and Universal Successor Features,” where we will expand on the insights related to Proposition 1 (expanding the text currently in lines 116-119).
>
> In terms of expanding on the insights related to Proposition 1, we will highlight the fact that Proposition 1 explains why our approach ultimately produces true SFs. Proposition 1 does this by proving that minimizing our losses (Eqs. 5 & 6) also minimizes the canonical SF loss used in Universal Successor Features (Eq. 4). In order to tie this to the previous section, we will also note that our approach minimizes these losses in a manner such that setting the basis features $\phi$ to a constant is not a solution.
> Specifically, we will note in the text that if one sets $\psi = c_2 $ and $\phi = c_1 = (1 - \gamma) c_2$,  then Eqs. 5 & 6 are not minimized, due to the fact that $\hat{y}$ and $R_{t+1}$ are not constants.
>
> We believe these revisions will significantly enhance the clarity and robustness of our manuscript. Again, we thank you for your insightful feedback, and please feel free to reach out if any further clarification is needed. We hope you will consider raising your score once more based on these responses.

---

### Author Rebuttal · Authors · 2024-08-06

We would like to thank the reviewers once again for their valuable feedback, which has guided clarifications and improvements that we will include in the final revision of our manuscript. **We have attached a set of figures in this Author Rebuttal, which we denote as General Response (GR)**, to address the main concerns from the reviewers. The concerns fall broadly under the following themes:

# 1. Complexity of the environments

During the rebuttal phase, we further evaluated our model in more complex settings using the Mujoco environments with pixel-based observations. We consider this benchmark to show the potential of our model in continuous action spaces.

Following the established protocol in [1], we started with the half-cheetah domain in Task 1 where agents were rewarded for running forward. We then introduced three different scenarios in Task 2: agents were rewarded for running backwards (Figure 1a in GR), running faster (Figure 1b in GR), and, in the most drastic change, switching from the half-cheetah to the walker domain (same num of actions) with a forward running task (Figure 1c in GR). **To ensure comparability across these diverse scenarios, we normalized the returns, considering that each task has different maximum attainable returns per episode.**

In all tested scenarios, our model consistently outperformed all baselines in Task 1 and particularly, Task 2, highlighting its superior adaptability and effectiveness in complex environments. This performance sharply contrasts with other SF-related baseline models, which struggled to adapt under similar conditions.

# 2. Marginal improvements

We initially used average episode returns to provide quick insights into short-term performance, but recognize that this metric may not fully capture the long-term benefits of our model. To address this, we also evaluated cumulative total returns across all tasks, which are illustrated in Figure 2 in GR.

These results demonstrate that our model not only learns effective policies more rapidly but also sustains these improvements, particularly in complex 3D environments where tasks are re-encountered (Figure 2c-d in GR).

Overall, our model showed significant improvement in cumulative returns over the baseline models, highlighting its robustness and ability to transfer learning effectively across tasks. This contrasts with DQN, which exhibited little to no transfer effects and required re-learning from scratch, as evidenced by its performance in these scenarios.

# 3. Simple nonlinear decoder
Reviewer 7tgV recommended a simple non-linear decoder to assess which model’s SFs most effectively decode into Successor Representations (SRs). We conducted this evaluation using both allocentric (fully-observable) and egocentric (partially-observable) pixel observations within the center-wall environment. The results, depicted in Figure 3 in the GR, demonstrate consistently high accuracy across both settings. This contrasts sharply with SFs developed using reconstruction constraints or random basis features, which, while effective in egocentric settings, perform poorly in allocentric settings where feature sparsity is greater. This analysis highlights the robustness and versatility of our model's SFs in varied observational contexts.

# 4. Stop Gradient Operator
The comments from Reviewer 2gcv prompted us to conduct an additional ablation study to elucidate the effectiveness of the reward prediction loss (Eq. 6) in our approach, compared to prior work [2] that faced challenges with similar methods. A key differentiator in our model is the application of a stop gradient operator on the basis features during the learning process with reward prediction loss.
We designed this study to specifically assess whether the stop gradient operator is essential for successful learning using reward prediction loss. The findings, presented in Figure 4a in GR, conclusively show that omitting the stop gradient operator leads to significantly reduced learning efficiency and policy effectiveness.

Additionally, visual analysis of the SFs in Figure 4b in GR further demonstrates that concurrently learned basis features and task-encoding vectors without a stop gradient operator result in SFs with poor discriminative capabilities, undermining effective policy learning. These results underscore the critical role of the stop gradient operator in maintaining the integrity and effectiveness of our learning process, confirming its necessity for achieving the robust performance we report.

# 5. Modifications to Figure 1
Lastly, there were additional concerns regarding the configuration and density of Figure 1. As previously detailed in individual rebuttal responses to Reviewer 7tgV and Tczf, and for broader awareness, we will implement the following modifications:
Simplification: We'll remove the loss functions from Figures 1d and 1e, with detailed descriptions retained in Appendix E and the main text, respectively. This will help focus attention on the structural content.

- **Reorganization**: Figure 1d will be moved to the Appendix as it primarily presents common approaches rather than our novel contributions, ensuring the main text remains focused on our work.

- **Relabeling and Relocation**: Figure 1e will be renamed as Figure 2 and relocated closer to Sections 4 and 5 where it is first mentioned, aligning it more closely with its textual references and enhancing narrative coherence.

- **Visual Guidance Enhancements**: We will replace terms like “Q-SF-TD loss” with “$L_\psi$: Q-SF-TD loss” and introduce color-coded symbols to improve figure-text integration. For example, pixel-level observations, $S_t$, will be described in the text with direct references to their visual representation in the newly labeled Figure 2.


[1] Yarats et al., 2021. Mastering visual continuous control: Improved data-augmented reinforcement learning.

[2] Ma, et al., 2020. Universal successor features for transfer reinforcement learning

---

### Author Response · Authors · 2024-08-14
**Concluding Remarks and Acknowledgments**

Dear all,

As the discussion period comes to a close, we would like to express our sincere gratitude to the reviewers for their valuable feedback, which has greatly guided the clarifications and improvements we plan to include in the final revision of our manuscript.

Among the key updates is the inclusion of results from the complex Mujoco environment using pixel observations, which strongly validate the robustness of our approach. Furthermore, we have clarified that the average episode returns were computed using moving averages and have included cumulative total return figures to demonstrate that our approach exhibits superior transfer across different environments. These results provide strong evidence that our method achieves significant improvements, beyond marginal gains.

Once again, we would like to emphasize that we presented moving average returns per episode instead of cumulative total return plots in our manuscript to demonstrate that *we allow learning for the first task to converge before introducing the second and subsequent tasks.*

In response to Reviewer 7tgV's suggestion, we have extended our analysis by decoding the Successor Features (SFs) into Successor Representations (SRs) using a simple non-linear decoder. This additional analysis highlights the robustness and versatility of our model's SFs in both egocentric and allocentric observations, which other baseline models did not achieve.

Regarding the concerns raised by Reviewer 2gcv about the similarity between our work and that of Ma et al., 2020 [1], we would like to **reaffirm the novel contributions our work makes: (1) Our approach does not require the agent to be provided with a goal—it is learned; (2) We provide direct evidence that our method works with pixel inputs; (3) We demonstrate that our approach eliminates the need for an SF loss; and (4) By removing the SF loss, we reduce the number of hyperparameters required, thereby simplifying the model.**

In the manuscript (Figures 17 to 21 in Appendix H), we included results comparing our approach to an agent (named "SF + Q-TD + Reward") that combines the Q-learning loss, SF loss, and reward prediction loss, similar to the setup in [1], with the key difference being the inclusion of reward prediction loss, as we do not assume prior knowledge of task specifics. The results showed that our model achieved better learning performance, especially in the continual learning setting (Figure 6a)., and was more computationally efficient (Figure 6b).

Another analysis we included in the general response was to *verify the importance of using a stop-gradient operator on the basis features* when learning the task-encoding vector with the reward prediction loss. We believe that incorporating the stop-gradient operator in our approach enables more effective learning. This is consistent with the findings in [1], which suggest that without the stop-gradient operator on the basis features, concurrently learning both the basis features and the task-encoding vector using the reward prediction loss can lead to learning difficulties.

Although Reviewer Tczf raised concerns about DQN outperforming SFs in the continual RL setting in Figure 2e to 2g, we were unable to directly address this as the specific results being referred to were not clear. However, we invite the reviewers to **refer to the plots generated using the total cumulative return in the same setup as Figures 2e to 2g (in our work) in Figure 2a-2c in general response, where it is clearly evident that our approach performed much better in later tasks.**

More concretely, across all our experimental results in the manuscript, including Figures 2, 3, 6, and 12 to 16, as well as in the complex Mujoco environment, our model, Simple SF, consistently outperforms DQN/DDPG, particularly after the first task.

Finally, we acknowledge the concerns regarding the configuration and density of Figure 1, as well as the need for better insights into Proposition 1. We agree with these observations and, as noted in the comments and general response, we will be making modifications to further improve the clarity and presentation of the manuscript.

The positive feedback on our manuscript's clarity and presentation has been very encouraging. We will incorporate the reviewers' suggestions into the final version of our manuscript. We believe these changes, arising from the discussions, will not only refine our manuscript but also emphasize the simplicity and practicality of our framework, paving the way for future research in the field of Successor Features.

We hope that these revisions address the raised concerns and further solidify the contributions of our manuscript.

[1] Ma, et al., 2020. Universal successor features for transfer reinforcement learning

---

### Decision · Program_Chairs · 2024-09-25

**Decision:**

Accept (poster)

**Comment:**

Scores 5 5 4 6

This is a solid paper, that is right on the threshold.

Criticism included that the tested environments are too limited and that the paper failed to properly discuss previous work from Ma et al 2020.  The rebuttal added further ablations and experiments, which is good and helps reviewers to increase their scores.  However, reviewer 2gcv wasn't completely convinced about the rebuttal against the concern that a key part of the paper is not completely new, but has been done by Ma et al. 2020 already.  However, 2gcv didn't answer the final rebuttal comment.

Overall the paper is good and the rebuttal did address most concerns, but at the end didn't convince all reviewers in all points.  The resulting score is just at the boundary.  I really appreciate papers trying SIMPLE approaches, so I tend to accept the paper.